# Non-antibiotics disrupt colonization resistance against enteropathogens

Anne Grießhammer[1,2,3,11], Jacobo de la Cuesta-Zuluaga[1,2,3,11], Patrick Müller[1,2,3], Cordula Gekeler[1,2,3], Jan Homolak[1,2,3,4,5], Hsuan Chang[6], Katharina Schmitt[1,2], Chiara Planker[1,2], Verena Schmidtchen[1,2], Suchira Gallage[3], Erwin Bohn[1,2], Taylor H. Nguyen[7], Jenny Hetzer[3], Mathias Heikenwälder[3,8], Kerwyn Casey Huang[7,9,10], Taiyeb Zahir[1,2] & Lisa Maier[1,2,3✉]

Non-antibiotic drugs can alter the composition of the gut microbiome[1], but they have largely unknown implications for human health[2]. Here we examined how non-antibiotics affect the ability of gut commensals to resist colonization by enteropathogens[3]. We also developed an in vitro assay to assess enteropathogen growth in drug-perturbed microbial communities. Pathogenic Gammaproteobacteria were more resistant to non-antibiotics than commensals and their post-treatment expansion was potentiated. For 28% of the 53 drugs tested, the growth of *Salmonella enterica* subsp. *enterica* serovar Typhimurium. (*S*. Tm) in synthetic and human stool-derived communities was increased, and similar effects were observed for other enteropathogens. Non-antibiotics promoted pathogen proliferation by inhibiting the growth of commensals, altering microbial interactions and enhancing the ability of *S*. Tm to exploit metabolic niches. Drugs that promoted pathogen expansion in vitro increased the intestinal *S*. Tm load in mice. For the antihistamine terfenadine, drug-induced disruption of colonization resistance accelerated disease onset and increased inflammation caused by *S*. Tm. Our findings identify non-antibiotics as previously overlooked risk factors that may contribute to the development of enteric infections.

The gut microbiome provides protection against intestinal infections by preventing pathogen colonization and the overgrowth of indigenous pathobionts. This resistance to colonization arises from antagonistic microbe–microbe interactions driven by competition for nutritional resources[3–6] and the induction of host immune responses[7]. Therefore, perturbations to the microbial community, such as those caused by antibiotic therapy, can lead to increased infection risk[8–11]. A common model organism used to study these processes is *S*. Tm, an invasive foodborne pathogen that causes inflammatory diarrhoea in immunocompetent individuals[6,12–14].

Many non-antibiotic drugs from diverse therapeutic classes can also collaterally alter the composition and function of the human gut microbiome[2], often by directly inhibiting the growth of commensal bacteria[1]. These perturbations are typically dose-dependent[15], can synergize in multimedicated patients and can accumulate with repeated exposure[16–20].

Similar to the effects of antibiotic treatment, alterations to the composition of the gut microbiome caused by non-antibiotic treatment could lead to a loss of colonization resistance. In support of this hypothesis, population-based metagenomic analyses have shown that intake of several non-antibiotics is associated with increased intestinal loads of pathobionts[19]. However, it remains unclear whether loss of colonization resistance occurs generally. Moreover, it is unknown whether pathogen levels increase owing to direct interactions of drugs with the gut microbiome or from disrupted host responses caused by drug use or disease. In situations when the association is mediated by the microbiome, identification of the specific effects of the drug on the microbiome that promote pathogen expansion will be important to ameliorate infection risk.

Here we develop a high-throughput in vitro assay to identify non-antibiotic medications that interfere with the ability of gut commensal communities to resist invaders. We also examine how drug-induced changes in microbiome composition and function lead to pathogen expansion. We mainly focus on the growth of *S*. Tm in defined microbial communities treated with non-antibiotics. We show that *S*. Tm growth is modulated by drug-induced changes in community biomass, community taxonomic composition, the presence of nutritional competitors or a combination thereof. Similar effects were observed for other pathogenic Gammaproteobacteria species and in complex microbial communities derived from human donors. Selected drugs that enhanced pathogen invasion in vitro also disrupted colonization resistance in mouse models and led to a more severe course of infection. Our results highlight the increased sensitivity of gut commensals to non-antibiotic drugs and reveal that

[1]Interfaculty Institute for Microbiology and Infection Medicine Tübingen, University of Tübingen, Tübingen, Germany. [2]Cluster of Excellence EXC 2124 Controlling Microbes to Fight Infections, University of Tübingen, Tübingen, Germany. [3]M3 Research Center for Malignome, Metabolome and Microbiome, University Hospital Tübingen, Tübingen, Germany. [4]Department of Pharmacology, University of Zagreb School of Medicine, Zagreb, Croatia. [5]Croatian Institute for Brain Research, University of Zagreb School of Medicine, Zagreb, Croatia. [6]European Molecular Biology Laboratory, Genome Biology, Heidelberg, Germany. [7]Department of Bioengineering, Stanford University, Stanford, CA, USA. [8]German Cancer Research Center (DKFZ), Division of Chronic Inflammation and Cancer, Heidelberg, Germany. [9]Department of Microbiology and Immunology, Stanford University School of Medicine, Stanford, CA, USA. [10]Chan Zuckerberg Biohub, San Francisco, CA, USA. [11]These authors contributed equally: Anne Grießhammer, Jacobo de la Cuesta-Zuluaga. ✉e-mail: l.maier@uni-tuebingen.de

such drugs are neglected risk factors for the development of enteric infections.

## Pathogens resist non-antibiotics

The growth of human gut commensal bacteria is directly inhibited by diverse non-antibiotic drugs[1]. Here we aimed to determine whether inhibition patterns differ between gut commensals and pathogens from the class Gammaproteobacteria. We investigated the direct effects of 1,197 drugs (used at 20 μM) approved by the US Food and Drug Administration (Extended Data Fig. 1a) on five Gammaproteobacteria species: *S*. Tm, *Haemophilus parainfluenzae*, *Shigella flexneri*, *Vibrio cholerae* and *Yersinia pseudotuberculosis*. We compared the responses of these pathogens to 43 commensal bacteria reported in our previous study[1]. Both groups were inhibited by a similar number of antibiotics (median ± interquartile range (IQR) of 80 ± 16 for commensals and 78 ± 4 for pathogens, adjusted $P = 0.55$, two-tailed $t$-test with Benjamini–Hochberg correction). However, commensals were affected by more non-antibiotics than pathogens (53 ± 37 for commensals and 17 ± 7 for pathogens, adjusted $P < 0.01$, two-tailed $t$-test with Benjamini–Hochberg correction) (Fig. 1a and Supplementary Table 1).

To identify dose–response relationships, we selected from the screen 65 antibiotic and non-antibiotic drugs with a wide range of inhibitory effects (Methods and Extended Data Fig. 1b). Using a subset of 20 gut commensals and 5 pathogens, we determined the concentration for 25% growth inhibition ($IC_{25}$). Non-antibiotics inhibited gut commensals at lower concentrations than pathogens (Extended Data Fig. 2a and Supplementary Table 2), which confirmed that commensals have increased drug sensitivity. Compounds that affected a higher number of commensals tended to be hydrophobic and have high molecular mass and large three-dimensional volume (Extended Data Fig. 2b,c). The set of drugs, both antibiotic and non-antibiotic, that inhibited a given species varied widely in each phylum (Extended Data Fig. 3a,b). Consequently, there was a weak association between drug sensitivity profiles and phylogenetic relatedness (Mantel's correlation: antibiotics = 0.08, $P = 0.04$; non-antibiotics = 0.03, $P = 0.18$) (Extended Data Fig. 3c,d).

As Gram-negative pathogens, Gammaproteobacteria species are protected from many drugs by their selective outer membrane. Moreover, compared with other commensals, their genomes have a higher proportion of genes linked to efflux processes ($P = 0.008$, one-tailed $t$-test) and to antibiotic resistance and stress responses ($P = 0.06$, one-tailed $t$-test) (Extended Data Fig. 3e,f). The importance of drug efflux for Gammaproteobacteria pathogens compared with other Gram-negative commensals became evident when efflux pumps were removed. In *S*. Tm, the deletion of *tolC*, which encodes a key component of the resistance nodulation cell-division multidrug efflux pump, induced sensitivity to an additional 35 drugs out of the 1,471 tested. By contrast, deletion of a homologous pump in the commensal *Phocaeicola vulgatus* induced sensitivity to only four drugs (Extended Data Fig. 3g and Supplementary Table 3).

These results suggest that stress and detoxification responses in commensals are less effective at withstanding non-antibiotics. By contrast, Gammaproteobacteria species may be more resistant to these compounds owing to their adaptations to hostile environments, such as those created by the host immune system during infection—conditions that commensals are less likely to face.

## Non-antibiotics drive pathogen expansion

Selective disruptive effects of drugs on gut commensals could alter the ability of microbial communities to resist the growth of pathogenic Gammaproteobacteria. To test this hypothesis in vitro, we developed a high-throughput challenge assay using a model synthetic community composed of 20 gut commensals (Com20). Com20 is phylogenetically

and functionally diverse and encodes 246 of the metabolic pathways in the MetaCyc database, which represents 61.3% of the 372 pathways detected with a prevalence of >20% in individuals from the Human Microbiome Project (Fig. 1b and Extended Data Fig. 4a). Com20 grew stably and reproducibly in vitro in the gut-mimetic medium mGAM[21] and readily colonized the gastrointestinal tract of germ-free mice for at least 57 days after an initial 7-day adaptation phase (Fig. 1b).

To investigate the effect of drug exposure on the synthetic community and *S*. Tm proliferation, Com20 was first treated with drugs for 24 h. After drug treatment, the community was challenged with *S*. Tm at 1:500 of its biomass to mimic the predominance of gut commensal bacteria in the initial stage of community invasion by *S*. Tm (Fig. 1c). The untreated community restricted pathogen growth, as quantified through *S*. Tm-specific luminescence (the median relative luminescence unit (RLU) of *S*. Tm in untreated Com20 was about 25 times lower than in pure culture) (Extended Data Fig. 4b–d). Using this assay, we tested 53 out of the 65 drugs evaluated in monoculture at 5 concentrations (note that we excluded drugs that directly inhibited *S*. Tm; Extended Data Fig. 1a). The invasion assay was robust, reproducible across replicates and mostly unaffected by washing of Com20 before pathogen challenge (Extended Data Fig. 4e,f). Different drugs led to distinct community compositions (Extended Data Fig. 4g) that were often predictable from $IC_{25}$ data for species tested in isolation, with emergent behaviours resulting in a minority of drug–microbe interactions (cross-protection, 19.0% of drug–microbe interactions; cross-sensitization, 4.1%) (Extended Data Fig. 4h). Of the 53 drugs tested, 15 promoted *S*. Tm expansion, often in a concentration-dependent manner, whereas 2 drugs inhibited *S*. Tm expansion in Com20, even though they had no inhibitory effects on *S*. Tm growth in monoculture (Fig. 1d, Extended Data Fig. 5a and Supplementary Table 4). Community biomass (measured via an optical density of 578 nm ($OD_{578}$)) was strongly negatively correlated with *S*. Tm growth (as assessed by luminescence) across drug treatments and with serial dilutions of untreated Com20 (Spearman's $\rho = -0.98$, $P < 0.01$; Fig. 1d, red dots and line). Non-antibiotic drugs had a smaller effect on community biomass than antibiotics; however, the effects were sufficient to alter the ability of the community to resist colonization by *S*. Tm.

Given the similar response of other pathogenic Gammaproteobacteria species to non-antibiotics in monoculture (Fig. 1a), we investigated whether the drugs that influenced *S*. Tm expansion would also affect the expansion of other pathogens. We performed challenge assays for six additional enteric pathogens that invade Com20 (Extended Data Fig. 5b,c) with a subset of ten drugs at five concentrations (Extended Data Figs. 1a and 5d). The invasion patterns of other Gammaproteobacteria species in drug-treated Com20 were significantly and positively correlated with that of *S*. Tm and with each other (Spearman's $\rho > 0.48$, $P < 0.05$ in all cases) (Fig. 1e and Supplementary Table 5).

Our results suggest that the effects of non-antibiotic drugs on gut commensals generally lead to the disruption of colonization resistance against pathogenic Gammaproteobacteria. However, the variation in pathogen-specific growth in drug-perturbed communities underscores the importance of pathogen-specific elements, such as their repertoire of virulence factors, their metabolic capabilities or their interactions with commensal bacteria.

## Drug-driven community shifts favour pathogens

A reduction in Com20 biomass was not always necessary for *S*. Tm levels to increase. Therefore, we analysed how drug-induced changes in Com20 diversity, with and without alterations in biomass, were associated with pathogen growth. To do so, we quantified the composition of 53 drug-treated communities using 16S rRNA gene amplicon sequencing (Extended Data Fig. 4g). On the basis of these taxonomic profiles, we used isolates to construct communities for which composition resembled that of four drug-treated Com20 communities (Extended Data Fig. 6a). In the absence of drugs, communities that mimicked

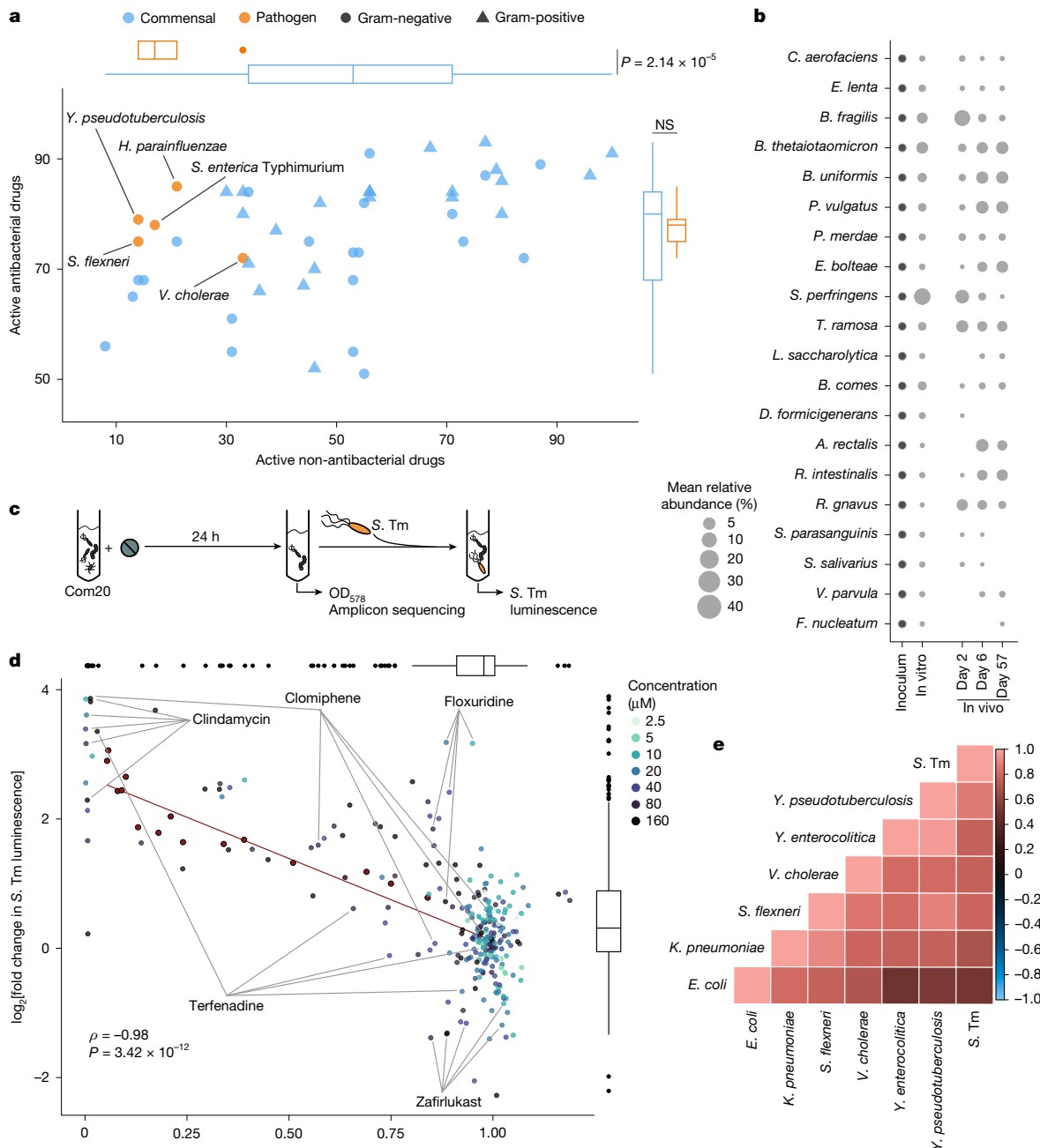

**Fig. 1 | Pathogenic Gammaproteobacteria are more resistant to non-antibiotics than commensal gut bacteria and proliferate in drug-perturbed microbial communities. a**, Association between the number of active (that is, inhibitory) antibiotic and non-antibiotic drugs across 43 gut commensals and 5 pathogens. Boxplots on the top and right show the distribution of the number of active antibiotic and non-antibiotic drugs for all gut commensals or pathogens. *P* values are from two-tailed *t*-tests. Boxplots show the median, the IQR, whiskers to the minimum and maximum within 1.5× the IQR, and outliers as individual points. **b**, Relative abundance of each Com20 member in the initial inoculum after 24 h of in vitro growth and in gnotobiotic mice 2, 6 and 57 days after colonization. The circle size is proportional to the relative abundance. The mean of 3–5 biological replicates is shown. *A. rectalis, Agathobacter rectalis; B. comes, Bariatricus comes* (also known as *Coprococcus comes*); *B. fragilis, Bacteroides fragilis; B. thetaiotaomicron, Bacteroides thetaiotaomicron; B. uniformis, Bacteroides uniformis; C. aerofaciens, Collinsella aerofaciens; D. formicigenerans, Dorea formicigenerans; E. bolteae, Enterocloster bolteae; L. saccharolytica, Lacrimispora saccharolytica; P. merdae, Parabacteroides merdae; R. gnavus, Ruminococcus gnavus; R. intestinalis, Roseburia intestinalis;*

*S. parasanguinis, Streptococcus parasanguinis; T. ramosa, Thomasclavelia ramosa.* **c**, Schematic of the in vitro *S*. Tm challenge assay. Com20 was exposed to various concentrations of drugs for 24 h. Next, the $OD_{578}$ of the community was measured as a proxy for biomass. The drug-treated community was then challenged with *S*. Tm in fresh medium. *S*. Tm was quantified on the basis of luminescence after 4.5 h. **d**, The association between community biomass and *S*. Tm growth. Each point corresponds to the $OD_{578}$ of Com20 and the luminescence of *S*. Tm after treatment with one of the 53 drugs tested at a given concentration denoted by the colour. The mean of three biological replicates is shown. $OD_{578}$ and luminescence measurements were normalized to the value of untreated Com20. The red points, linear regression line and statistics show the values for serially diluted untreated Com20. Highlighted drugs, among others, were selected for downstream experiments. Spearman's correlation coefficient between the relative optical density ($OD_{578}$) and $\log_2[S.$ Tm luminescence], $P = 3.42 \times 10^{-12}$. Boxplots are as for **a**. **e**, Spearman's correlation coefficients between the growth of Gammaproteobacteria pathogens with Com20 across nine drugs. Comparisons were performed where possible. NS, not significant ($P > 0.05$).

biomass-reducing drugs (erythromycin and sertindole) led to increased pathogen levels only after dilution. Conversely, *S*. Tm levels in communities that mimicked drug treatments and did not alter biomass (zafirlukast and floxuridine) phenocopied drug-treated Com20 even at high $OD_{578}$ values (Extended Data Fig. 6b).

We next asked how the diversity of the community was linked to pathogen growth in the community. For this, we looked into the association between alpha diversity, as measured by the species richness and the Shannon's index, and *S*. Tm luminescence. We observed a negative correlation between both diversity measures and *S*. Tm luminescence (Spearman's $\rho$ richness = −0.37, Shannon index = −0.39, adjusted $P < 0.001$ in both cases). However, both measures were also significantly and positively correlated with community biomass ($OD_{578}$ Spearman's $\rho$ richness = 0.40, Shannon index = 0.47, Benjamini–Hochberg-adjusted $P < 0.001$ in both cases). Given this positive correlation, we asked whether microbial diversity retained explanatory power after accounting for community biomass. For this analysis, we compared the following five linear models of *S*. Tm luminescence: (1) species richness; (2) Shannon index; (3) $OD_{578}$; (4) a combination of $OD_{578}$ and Shannon index; and (5) a combination of $OD_{578}$ and species richness. The model of *S*. Tm luminescence that incorporated both $OD_{578}$ and species richness provided the best fit (adjusted $R^2 = 0.26$). This finding indicates that when community biomass was accounted for, the number of species present explains the growth of the pathogen better than how evenly distributed the species are.

We then assessed drug effects on Com20 composition and their links to *S*. Tm expansion. After removing low-biomass treatments, we classified the remaining treatments into three groups: *S*. Tm favouring (9 drugs and 9 treatments); *S*. Tm restricting (1 drug and 1 treatment); and *S*. Tm neutral (33 drugs and 37 treatments) (Fig. 2a,b, Extended Data Fig. 5a and Supplementary Table 4). We then examined differences in beta diversity among the groups. Community composition in all three groups was significantly different from untreated controls while accounting for $OD_{578}$ (*S*. Tm neutral versus controls, permutational multivariate analysis of variance (PERMANOVA) adjusted $R^2 = 0.04$; *S*. Tm favouring, adjusted $R^2 = 0.09$; *S*. Tm restricting, adjusted $R^2 = 0.31$; Benjamini–Hochberg-adjusted $P < 0.001$ in all cases) (Fig. 2c). Drug treatment resulted in changes in the composition of the community profiles, regardless of the colonization outcome (Fig. 2c and Extended Data Fig. 6c). However, *S*. Tm-restricting community compositions clustered together and were characterized by a depletion in *Sarcina perfringens* (as per the Genome Taxonomy Database; also known as *Clostridium perfringens*), *Veillonella parvula* and *Fusobacterium nucleatum* (Benjamini–Hochberg-adjusted $P < 0.1$ in all cases) (Fig. 2c and Extended Data Fig. 6d). Consistently, in the absence of any treatment, the removal of *S*. perfringens from Com20 substantially changed the community structure and significantly restricted *S*. Tm expansion (Fig. 2d and Extended Data Fig. 6e–g). We observed similar pathogen-restricting community properties in the absence of *S*. perfringens for other metabolically related pathogenic Gammaproteobacteria species, including *Klebsiella pneumoniae*, *S. flexneri* and *Yersinia enterocolitica* (Extended Data Fig. 7a–c). This observation highlights that specific changes in community structure are consistently linked to colonization outcomes across different drugs and pathogens. However, other associations between individual species and *S*. Tm levels were only observed in the context of drug treatment of Com20. In the absence of treatment, direct pathogen–commensal interactions were poor predictors of *S*. Tm growth. That is, the expansion of a pathogen in pairwise co-cultures or in dropout communities (19 members) did not follow the patterns deduced from the drug treatments (Extended Data Fig. 6e–h and Supplementary Note).

In summary, non-antibiotics can promote pathogen invasion by changing the community biomass or by altering the diversity and composition of the community. These effects can occur concomitantly, which emphasizes the importance of species richness in a community for protection against a pathogen. Moreover, the discordance between pathogen–commensal interactions and invasion of drug-treated communities underscores that colonization after drug treatment is a complex, context-dependent phenomenon.

## Com20 treatment shifts *S*. Tm gene expression

As drug treatment altered the taxonomic composition of the community, we next evaluated gene expression patterns of the pathogen and commensals in drug-treated communities. For this, we used Transwell plates in which *S*. Tm and drug-exposed Com20 were separated by a membrane but shared the same culture medium to ensure an adequate quantity of *S*. Tm cells under all treatments. Four drugs that promoted *S*. Tm expansion were assessed: clomiphene and terfenadine, which decreased community biomass, and simvastatin and floxuridine, which did not.

Although the distributions of expression of *S*. Tm genes were similar between treatments (Fig. 2e), it was possible to distinguish the response of the pathogen growing in an untreated community from treated communities (Fig. 2f). Biomass-depleting drugs (low-biomass drugs) produced transcriptional profiles similar to those of *S*. Tm grown in the absence of a community and were distinct from those in biomass-preserving treatments (high-biomass drugs). Notably, the two high-biomass treatments led to distinct transcriptional profiles. *S*. Tm genes involved in carbon metabolism and the transport of simple sugars were more frequently upregulated in treated compared with untreated communities (DMSO only). By contrast, genes involved in chemotaxis, toxin production, the flagellar apparatus and ribosome assembly were downregulated (Extended Data Fig. 8a and Supplementary Table 6).

In the community, treatment with high-biomass drugs resulted in large changes in the distribution of transcript levels compared with untreated Com20 and between both treatments (Fig. 2g), which precluded the identification of differentially expressed genes. Therefore, we examined the top 20% highest expressed genes of each species under each condition and determined how many of the highest expressed genes were previously identified as markers of stress responses in bacteria[22]. After exposure, the fraction of stress response markers increased compared with untreated Com20 (mean ± s.d., control = 0.40 ± 0.06; floxuridine = 0.42 ± 0.08, adjusted $P = 0.21$; simvastatin = 0.46 ± 0.09, adjusted $P = 0.01$; one-tailed $t$-test with Benjamini–Hochberg correction) (Extended Data Fig. 8b,c). In treated communities, pathways involved in translation, protein folding, ribosome function and biofilm formation were frequently represented among the most expressed genes. By contrast, pathways for the synthesis of vitamins and other secondary metabolites were less frequently represented among this set (Supplementary Table 7).

These results suggest that the expansion of *S*. Tm in Com20 after biomass-reducing treatments largely results from decreased competition with commensal bacteria for the available resources. Conversely, the invasion of high-biomass communities is facilitated by alterations in the function of the community, which are drug specific and involves an active response from the pathogen. In these cases, the response of *S*. Tm varies, which further highlights the role of context in colonization resistance.

## Niche competitor limits *S*. Tm growth post-treatment

A niche competitor can help a microbial community resist a pathogen. To study this scenario in the context of drug-induced perturbations, we generated a new community (Com21) by adding a species with metabolic characteristics similar to *S*. Tm to Com20: the commensal strain *Escherichia coli* ED1α (Extended Data Fig. 7d,e). This addition increased the metabolic diversity encoded by Com21 compared with Com20 (Extended Data Fig. 4a) and reduced *S*. Tm levels in the absence of drug treatment (Extended Data Fig. 4d). We observed a positive correlation

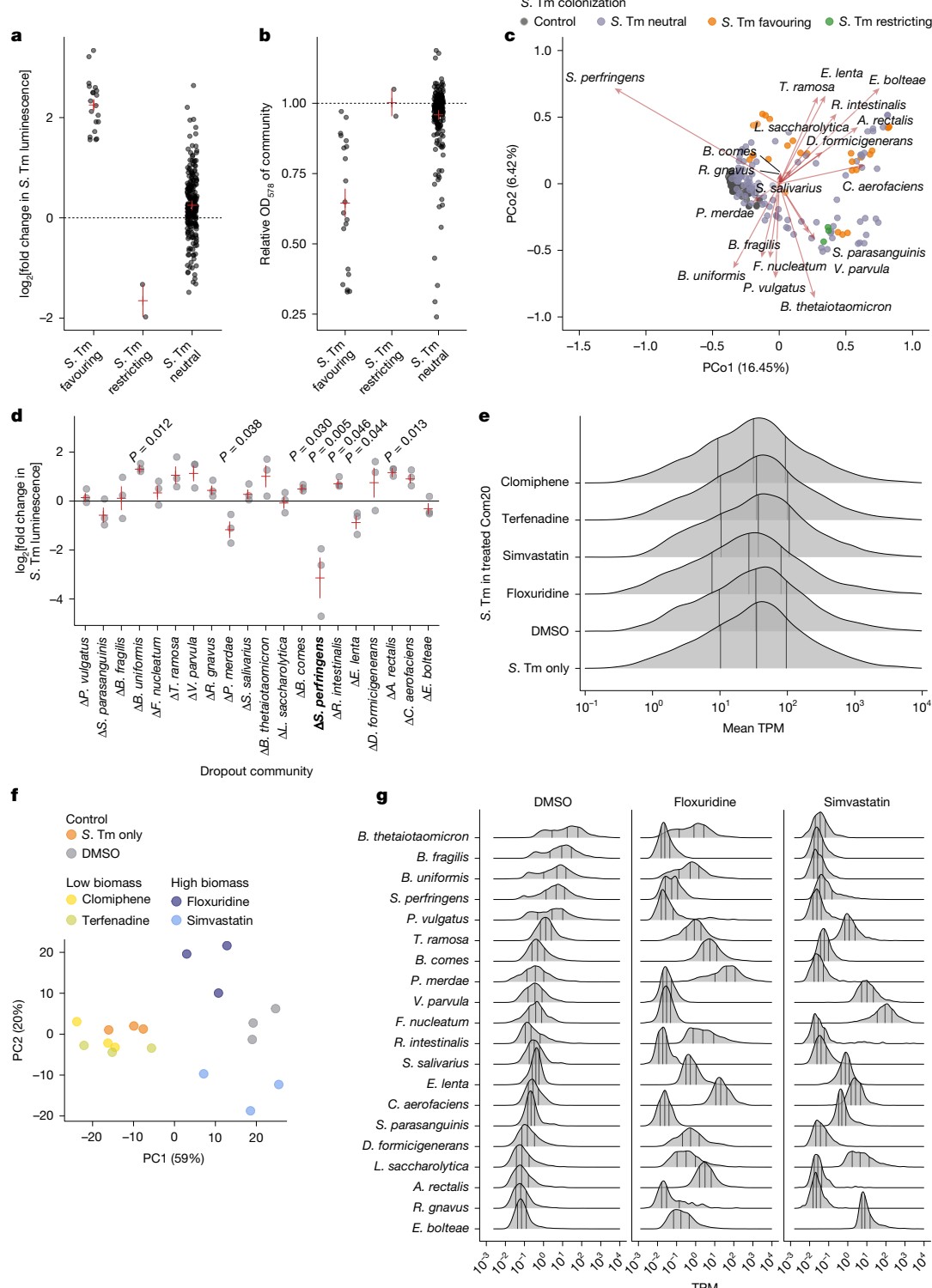

**Fig. 2 | Drug treatment of Com20 facilitates the growth of *S*. Tm through alterations in community biomass, composition and function. a**,**b**, *S*. Tm growth as measured by luminescence (**a**) and the $OD_{578}$ of drug-treated Com20 (**b**) in the challenge assay separated by colonization groups. Values were normalized to the value of untreated Com20. Each point represents the mean across three biological replicates. Red lines represent the mean ± 1 s.e.m. **c**, Biplot and principal coordinate analysis of drug-treated Com20 based on Bray–Curtis distances. Each point represents a drug-treated community coloured by colonization group (untreated controls are shown in grey). The direction of the arrows indicates the correlation of the abundance of a species with each principal coordinate (PCo1 and PCo2), and their length reflects the strength of the association. **d**, Growth of *S*. Tm in 19-member communities that lack one

member of the Com20 community compared with its growth in the full Com20 community. *S*. *perfringens* is highlighted in bold (Extended Data Fig. 6f–h). Red lines represent the mean ± 1 s.e.m. across three biological replicates. *P* values are from two-sided *t*-tests (only values ≤0.05 are shown). **e**, Distribution of mean *S*. Tm gene expression levels in co-culture with Com20 or in monoculture in three biological replicates per treatment. TPM, transcripts per million. Black vertical lines indicate quartiles. **f**, Principal component analysis (PC1 and PC2) of *S*. Tm gene expression based on the 500 genes with the highest variance across treatments. **g**, Distribution of mean gene expression levels of each Com20 member across treatments in three biological replicates. Black vertical lines indicate quartiles.

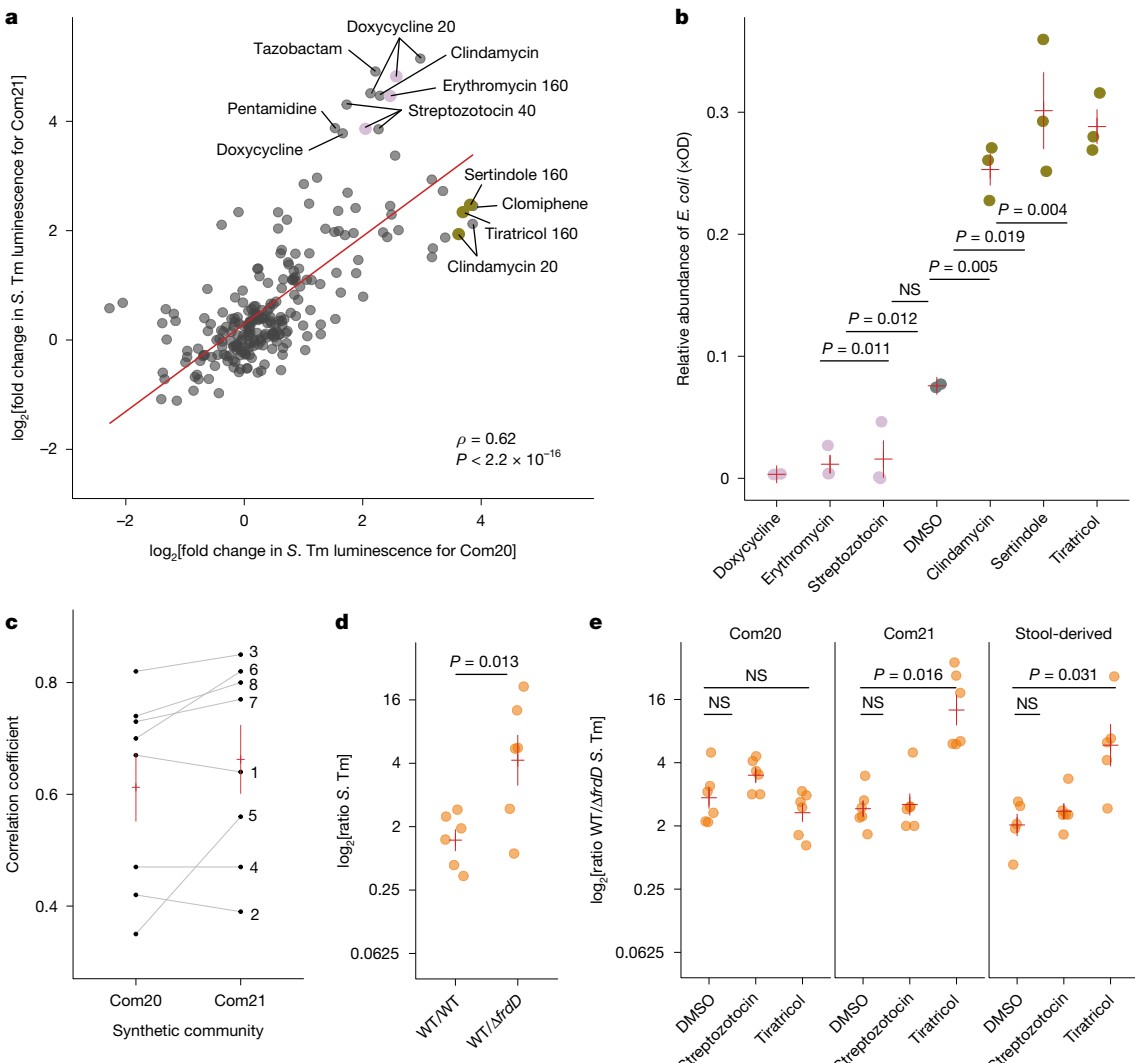

**Fig. 3 | S. Tm growth in drug-disrupted communities is modulated by the effect of treatment on the close niche competitor E. coli ED1α. a**, Growth of *S.* Tm in Com20 compared with *E. coli*-containing Com21 across 240 drug–concentration combinations. The regression line is indicated in red. Conditions with a large difference in the growth of *S.* Tm between communities ($\log_2$[fold change] of ≥3.5 or ≤−3.5) are highlighted. Conditions followed up by 16S rRNA sequencing are shown in gold and pink. Numbers following the text labels indicate drug concentrations in μM. The Spearman's correlation coefficient is between the $\log_2$[fold change] in *S.* Tm luminescence of Com20 and Com21. **b**, Relative abundance of *E. coli* ED1α in Com21 scaled by the total community biomass (OD). Pink points correspond to treatments with lower *S.* Tm growth in Com20 than in Com21 and gold points indicate treatments with higher *S.* Tm growth in Com20 than in Com21 from **a**; grey points correspond to untreated

Com21. Red lines represent the mean ± 1 s.e.m. *P* values from two-sided *t*-tests. **c**, Spearman's correlation coefficient of the growth of *S.* Tm in stool-derived communities from eight donors (1–8) compared with Com20 or Com21 across multiple treatments. Red lines represent the mean ± 1 s.e.m. across all stool-derived communities. **d**, Growth of a fumarate respiration-impaired Δ*frdD S.* Tm mutant relative to WT *S.* Tm in an in vitro challenge assay, involving exposure to only *E. coli* ED1α. Red lines represent the mean ± 1 s.e.m. *P* values are from one-tailed Wilcoxon test. **e**, Growth of *S.* Tm Δ*frdD* relative to WT in Com20, Com21 or a human donor-derived community (stool-derived) after treatment with *E. coli*-targeting streptozotocin and *E. coli*-sparing tiratricol. Red lines represent the mean ± 1 s.e.m. Adjusted *P* values with Benjamini–Hochberg correction from one-tailed Wilcoxon tests.

between drug effects on *S.* Tm expansion in Com20 and Com21 (Spearman's $\rho = 0.62$, $P < 0.01$; Fig. 3a and Supplementary Table 8). Expansion of the pathogen on treated communities was facilitated when *E. coli* was targeted by the drug treatment. For treatments that included *E. coli* inhibitors, we observed a greater *S.* Tm luminescence in treated compared with untreated Com21 and between treated and untreated Com20. Conversely, drugs that increased the relative abundance of *E. coli* resulted in decreased pathogen levels in Com21 compared with the untreated community (Fig. 3b and Extended Data Fig. 9a–c).

Com20 and Com21 are simplified models of a typical human gut microbial community, which contains many more species. To assess whether our findings were generalizable to more diverse communities, we derived stable microbial communities from stool samples

of eight healthy adults. The sensitivity of the stool-derived communities to drugs varied across donors (Extended Data Fig. 9d). After treatment with ten drugs at various concentrations (Extended Data Fig. 1a), *S.* Tm growth in the stool-derived communities was positively correlated with growth in Com20 (mean Spearman's correlation across all stool samples = 0.61 ± 0.17) and Com21 (mean correlation = 0.66 ± 0.17). This result held for all individual stool-derived communities (Spearman's $\rho > 0.35$, Benjamini–Hochberg-adjusted $P < 0.1$ in all cases) (Extended Data Fig. 9e and Supplementary Table 9). The increased correlation between stool-derived communities and Com21 compared with Com20 (Fig. 3c) may be explained by the presence of *Escherichia* species in the stool-derived communities.

Next, we interrogated candidate biochemical pathways that could drive nutritional competition between S. Tm and E. coli in the presence of drugs that promote E. coli growth. We proposed that competition for fumarate as an electron acceptor in anaerobic respiration might have a role, as fumarate respiration is key during various stages of gut colonization[23,24]. In co-culture with E. coli ED1α, a S. Tm mutant lacking the frdD gene (ΔfrdD), which encodes the D subunit of fumarate reductase, was outcompeted by wild-type (WT) S. Tm (P = 0.013, two-tailed t-test) (Fig. 3d). Correspondingly, drug treatment with tiratricol, which increased E. coli counts in Com21 and stool-derived communities, resulted in a competitive disadvantage for S. Tm ΔfrdD compared with WT S. Tm (P = 0.016 and P = 0.031, respectively; adjusted P values with Benjamini–Hochberg correction are from one-tailed Wilcoxon tests) (Fig. 3e). This finding indicates that fumarate respiration is an important driver of S. Tm expansion during certain treatments.

These results suggest that drug-induced alterations in the abundance of a niche competitor can influence the ability of a pathogen to expand in a microbial community. Differences in the drug sensitivity of niche competitors will lead to different community compositions and alter the ecological dynamics, which in turn influence the outcome of invasion depending on the fitness of the pathogen.

## Drugs impair S. Tm resistance in mice

We assessed whether the modulation of S. Tm growth by non-antibiotics observed in vitro would translate into disruption of colonization resistance in vivo. For this, we used three animal models (Fig. 4a): gnotobiotic mice colonized with Com20; specific-pathogen-free (SPF) mice; and gnotobiotic mice colonized with a stool-derived community from a human donor (hereafter referred to as humanized mice). We selected five drugs on the basis of their effects in vitro (Extended Data Fig. 5a,d): four that promoted pathogen growth (clotrimazole, chlorpromazine, terfenadine and clomiphene) and one that restricted pathogen growth (zafirlukast). Drugs were administered at concentrations equivalent to their human dose for chronic treatment (3–60 mg kg⁻¹).

In Com20-colonized mice, the pathogen-favouring drugs led to significantly higher S. Tm levels in faeces and the caecum 1 day post-infection (d.p.i.) compared with controls (adjusted P values for faeces: clotrimazole = 0.009, chlorpromazine = 0.012 and terfenadine = 0.002; adjusted P values for caecum: clotrimazole = 0.004, chlorpromazine = 0.019 and terfenadine = 0.009; Wilcoxon test with Benjamini–Hochberg correction), whereas zafirlukast did not (Fig. 4b,c). Similar results were observed in SPF mice, although in this model, zafirlukast treatment led to increased S. Tm levels (adjusted P values for faeces: clotrimazole = 0.033, zafirlukast = 0.017, chlorpromazine = 0.011, terfenadine = 0.004 and clomiphene = 0.005; adjusted P values for caecum: zafirlukast = 0.017, chlorpromazine = 0.017, terfenadine = 0.004 and clomiphene = 0.005; Wilcoxon test with Benjamini–Hochberg correction) (Fig. 4d,e). Increased S. Tm loads did not lead to host symptoms, signs of intestinal inflammation or systemic infection 24 h after S. Tm challenge in either mouse model (Extended Data Fig. 10a,b). For the humanized mouse model, we focused on terfenadine as it had the strongest effects in Com20-colonized mice and SPF mice. Terfenadine-treated animals exhibited higher S. Tm loads in faeces at 1 and 4 d.p.i. (P = 0.036 for day 1 and P = 0.040 for day 4) (Fig. 4f) and in the caecum at 4 d.p.i. (P = 0.041) (Fig. 4g) but not at systemic sites (Extended Data Fig. 10c). A more rapid increase in faecal lipocalin-2 (also known as NGAL) levels (P = 0.040 for day 2 and P = 0.021 for day 3) (Fig. 4h) and a higher S. Tm pathoscore at 4 d.p.i. (P = 0.0022) (Fig. 4i) indicated earlier disease onset and increased severity of inflammation compared with controls.

Pathogen levels changed without large rearrangements in microbiome composition in treated mice (Extended Data Fig. 10d–f and Supplementary Tables 10–12) or significant changes in E. coli counts in SPF and humanized mice after drug treatment (adjusted P > 0.05

(not significant) in all cases, Wilcoxon test with Benjamini–Hochberg correction) (Extended Data Fig. 10g,h). This result was in line with our in vitro work for biomass-lowering, non-E. coli-targeting drugs such as terfenadine. Moreover, terfenadine treatment alone did not result in proinflammatory effects, pathological changes or alterations in epithelial hypoxia levels in caecal tissue, which are known to support the growth of enteric pathogens[14,25,26]. These observations indicate that increased S. Tm levels are probably not caused by a host physiological response to drug treatment (Supplementary Fig. 1).

Overall, these results confirm our in vitro findings by demonstrating that non-antibiotic drugs across therapeutic classes disrupt colonization resistance against S. Tm in mice with defined and complex microbiotas. For certain drugs (for example, zafirlukast), the interference with colonization resistance depended on the microbiome composition of the host. For the antihistamine terfenadine, higher pathogen load led to more rapid disease progression in humanized mice.

## Discussion

Resistance against invasion of pathogenic bacteria is a key ecosystem service provided by the microbiome to the host[3,27]. Although it is well established that antibiotics can disrupt this community property[11,18,20], the effects of non-antibiotics on colonization resistance were largely unknown. Population-level metagenomic studies and epidemiological analyses of large cohorts have identified a link between drug consumption, higher pathobiont load[19] and symptoms consistent with gastrointestinal infections[28]. We directly addressed this gap in knowledge by systematically investigating how drug-induced disruption of communities of gut commensals affects pathogen invasion in vitro and in vivo. Starting from a large number of compounds, our approach led to the identification of non-antibiotic drugs that increased S. Tm load in mice. The effect of drug exposure on microbial communities was generally disruptive, with more compounds promoting S. Tm expansion than restricting it. These drugs belonged to a wide range of therapeutic classes, including antiasthmatic, antipsychotic, antifungal, antihistaminic and selective oestrogen receptor modulator agents. Notably, unlike antibiotics, these drugs do not have broad antimicrobial effects. Instead, they cause more subtle changes in microbial communities, which highlights new ways in which drugs can impair colonization resistance.

Drug-induced changes in the microbial community increased pathogen expansion in several ways. Non-antibiotics reduced the total biomass of the microbial community by either killing or inhibiting the growth of a subset of commensals, which therefore enabled the pathogen to proliferate in a similar way to post-antibiotic expansion (for example, clomiphene; Fig. 1d). Alternatively, non-antibiotics shifted the diversity or composition of the microbial community—without necessarily affecting the biomass—towards a community state that is unable to resist the growth of the pathogen (for example, floxuridine; Fig. 2f,g). Moreover, drugs selectively targeted species with high nutritional overlap with the pathogen, so that competition for resources was reduced (for example, streptozotocin; Fig. 3a,b). The level of colonization resistance will depend on the degree to which each factor is affected, which is a function of the compound and the baseline state of the community. As a metabolic generalist, S. Tm can adapt to diverse post-drug landscapes; however, the ultimate success of the pathogen will depend on its fitness relative to the community and community members, especially niche competitors such as E. coli[29]. Therefore, differences in drug sensitivities between resident microorganisms and invading pathogens (Fig. 1a), combined with their differential ability to use limited substrates (Fig. 3d,e), will influence infection outcomes.

Community changes may occur through direct interference of a drug with bacterial structures or processes[30–32], sequestration of nutrients[33] or the induction of physiological changes in the host[28,34]. Consequently, nutrient competition between commensals and pathogens

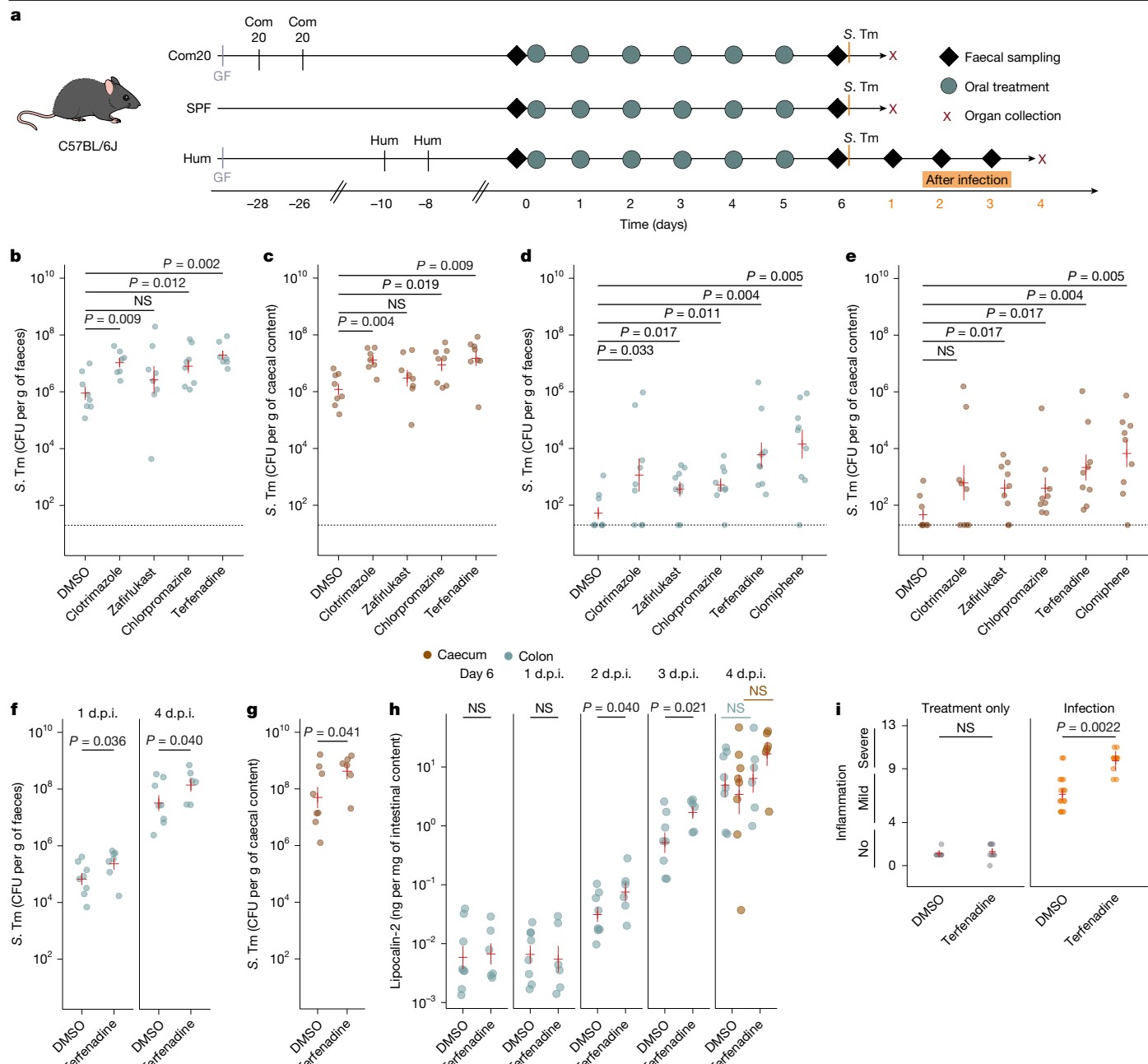

**Fig. 4 | Non-antibiotics from diverse therapeutic classes disrupt colonization resistance in mice. a**, Drug treatment and sampling scheme for three mouse models: Com20-colonized mice (defined microbiome); SPF mice with a complex mouse microbiome; and humanized (Hum) mice with a complex human microbiome. Oral gavage of drugs was performed daily for 6 days at the following doses: 38 mg kg⁻¹ for clotrimazole; 20 mg kg⁻¹ for zafirlukast; 3 mg kg⁻¹ for chlorpromazine; 25 mg kg⁻¹ for terfenadine; and 60 mg kg⁻¹ for clomiphene. **b**,**c**, *S.* Tm load in faeces (**b**) and caecum (**c**) of drug-treated Com20-colonized mice 1 d.p.i. with *S.* Tm. Number of mice per treatment: 8 (DMSO), 7 (clotrimazole), 8 (zafirlukast), 8 (chlorpromazine) and 7 (terfenadine). CFU, colony-forming unit. **d**,**e**, *S.* Tm load in faeces (**d**) and caecum (**e**) at 1 d.p.i. in SPF mice. Number of mice per treatment: 9 (DMSO), 9 (clotrimazole), 9 (zafirlukast), 9 (chlorpromazine), 9 (terfenadine) and 9 (clomiphene). **f**,**g**, *S.* Tm load in faeces (**f**) at 1 and 4 d.p.i. and in the caecum (**g**) at 4 d.p.i. in humanized mice. Number of mice per

treatment: 8 (DMSO) and 7 (terfenadine). **h**, Lipocalin-2 levels were measured by ELISA in faecal samples from humanized mice after treatment and before infection (day 6) and at 1 and 4 d.p.i. Number of mice used are as in **g**. Red lines indicate the mean ±1 s.e.m. of the above number of biological replicates. Adjusted *P* values with Benjamini–Hochberg correction from one-tailed Wilcoxon tests for comparisons of drug-treated versus DMSO-treated mice. **i**, Histopathological evaluation of caecal sections after haematoxylin and eosin staining (Methods). Left, mice were either treated with DMSO (8 mice) or terfenadine (6 mice). Right, mice were infected after treatment with DMSO (5 mice) or terfenadine (5 mice). Each point represents a pathoscore assigned to one animal by an independent evaluator. Generalized linear mixed models were used with the animal identifier as a random effect. Two-sided Wald *z*-tests assessed fixed effects and post hoc comparisons.

will be altered, which affects the ability of the community to effectively respond to the pathogen[6]. Furthermore, drugs can cause lysis of commensal microorganisms to alter the pool of available substrates such as microbiota-derived fumarate[24], which is used by *S.* Tm for anaerobic

respiration. Other factors, such as the production of inhibitory compounds such as short-chain fatty acids[35] and bacteriocins[36] or the inhibition of the expression of virulence factors[37], can also modulate pathogen growth. Beyond altered ecological dynamics as a general

consequence of drug treatments, no individual molecular interaction or cellular process explained the loss of colonization resistance across all drugs. Different compounds led to distinct community compositions (Fig. 2c) and gene expression patterns (Fig. 2e–g), yet exhibited similar reductions in resistance to *S*. Tm. Given the large chemical diversity of non-antibiotic drugs, along with factors such as dosage, treatment duration and inter-individual variation in the microbiome, future studies should evaluate drug–microbe–host interactions in a context-dependent basis once a relevant phenotype has been identified.

Our work is not without limitations. We propose a framework to explain how non-antibiotics can alter the ecological properties of the microbiome that results in a loss of colonization resistance. However, this framework currently lacks insight into the underlying molecular and cellular mechanisms. Because different drugs may affect different microorganisms in distinct ways, future research will need to take a comprehensive, case-by-case approach to uncover these processes. Moreover, our in vitro analyses represent a conservative estimate of the number of non-antibiotics with the potential to increase pathogen load. As our high-throughput method cannot account for host-mediated aspects of colonization resistance, we may have overlooked non-antibiotic drugs that promote pathogen growth due to microbiome-independent factors. Nonetheless, we demonstrated that drug-induced loss of colonization resistance can occur in vivo. Using models of varying microbial complexity, we showed that non-antibiotic drugs can increase *S*. Tm levels in the mouse gut and exacerbate inflammation after—but not before—infection compared with untreated animals.

In summary, the current work provides a basis for understanding non-antibiotic-mediated microbiome disruption and the expansion of pathogenic bacteria. Our results suggest that non-antibiotic drugs can compromise colonization resistance through means similar to classical antibiotics—primarily by reducing commensal biomass and diversity—thereby weakening nutritional competition against pathogens. However, the impact of non-antibiotics on commensal bacteria is generally milder (Fig. 1d), which means that higher doses or prolonged exposure may be required to disrupt the microbial community. This disruption may enable opportunistic colonization by enteric pathogens and poses a risk that healthcare professionals may underestimate. Thus, future studies should examine the effects of non-antibiotic drugs across a wide range of microbiome compositions, drug dosages and treatment regimens. The outcomes of these studies will be pivotal in the development of strategies to predict, minimize and mitigate microbiome-mediated disruptions.

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

## Methods

### Bacterial cultivation of monocultures, Com20, Com21 and stool-derived communities

The species used in this study are listed in Supplementary Table 13. They were purchased from the Leibniz-Institut DSMZ-Deutsche Sammlung von Mikroorganismen und Zellkulturen GmbH, BEI Resources, the American Type Culture Collection or Dupont Health & Nutrition, or were provided as gifts from the Denamur Laboratory (INSERM), the Blokesch Laboratory (EPFL), the Andrews–Polymenis Laboratory (Texas A&M University), the Darby Laboratory (UCSF) or the Wagner Laboratory (University of Tübingen). All gut commensal species, whether grown individually or as a community, were cultivated in mGAM (Nisui Pharma Solutions) at 37 °C, with the exception of *Veillonella parvula* and *Bilophila wadsworthia* monocultures. We cultured *V. parvula* in Todd–Hewitt broth supplemented with 0.6% sodium lactate and *B. wadsworthia* in mGAM supplemented with 60 mM sodium formate and 10 mM taurine. We pre-reduced the medium for a minimum of 24 h under anoxic conditions (2% $H_2$, 12% $CO_2$ and 86% $N_2$) in an anaerobic chamber (Coy Laboratory Products). The species were inoculated from frozen stocks into liquid culture medium and passaged twice (1:100) overnight to ensure robust growth. We periodically verified the purity and identity of the species through sequencing of the 16S rRNA gene and/or MALDI–TOF mass spectrometry[38].

We selected a set of 31 prevalent and abundant species from the human gut microbiome, which differed by >3% in their 16S rRNA gene sequences in the V4 region. When monocultures of all species were mixed in equal OD ratios, 20 of these species were consistently detectable and their levels were stable after several passages[15].

For experiments involving human stool-derived material, informed consent was obtained from all eight donors (approved by the Ethics Committee of the University Hospital Tübingen, project ID 314/2022B02). We generated stool-derived communities from fresh human faecal samples as previously described[39,40] by inoculating from frozen glycerol stocks into 3 ml BHI and serial dilution of 1:200 for three 48-h passages. This process ensured that the composition reached a steady state before measurements. We performed the experiments in clear, flat-bottomed 96-well plates (Greiner Bio-One). We sealed the plates with breathable AeraSeals (Excel Scientific). Communities were stored with glycerol as frozen stocks at −80 °C, inoculated from frozen stocks into fresh mGAM and grown overnight.

We carried out selective plating of pathogens under aerobic conditions. For animal experiments, we cultured *S.* Tm in LB broth supplemented with 0.3 M NaCl and determined *S.* Tm loads in intestinal content and organs on MacConkey agar supplemented with 50 μg ml⁻¹ streptomycin.

### Prestwick library screening for pathogens

We carried out the Prestwick library screening as previously described[1] on five pathogenic bacterial species in mGAM under anaerobic conditions. In brief, the library, which consists of 1,197 drugs approved by the Food and Drug Administration, was diluted to 100-fold the working concentration in DMSO (2 mM) in V-bottom polypropylene plates (Greiner Bio-One, 651261). For the screening experiments, we diluted drug master plates to 2-fold the working concentration in mGAM (40 μM) in U-bottom plates (Thermo Scientific, 168136), aliquoted them (50 μl per plate) and stored them at −20 °C for a maximum of 1 month. The DMSO control wells in each 96-well plate served as controls. For experiments with *H. parainfluenzae*, we supplemented mGAM with 0.5 mg l⁻¹ hemin and 2 mg l⁻¹ NAD. Before inoculation, we pre-reduced the drug plates overnight in an anaerobic chamber.

Before the screening experiments, we passaged bacterial strains twice overnight (1:100) anaerobically and adjusted the $OD_{578}$ to 0.02. After inoculation, the starting $OD_{578}$ for all bacterial species was 0.01 and the drug concentration in the plate was 20 μM with 1% DMSO. We sealed all plates with breathable membranes (Breathe-Easy, Sigma-Aldrich, Z380059). Bacterial growth was tracked by measuring the $OD_{578}$ every hour for 24 h using a microplate spectrophotometer (EON, Biotek) coupled with a Biostack 4 microplate stacker (Biotek), both housed inside an incubator (EMBL workshop). All screening experiments were performed in three biological replicates. For analysis, we truncated growth curves at the transition from the exponential to the stationary phase for analysis. We then calculated the area under the curve (AUC) using the trapezoidal rule and normalized it to the solvent or DMSO controls in the same plate. We identified hits from normalized AUC measurements by fitting heavy-tailed distributions, specifically the scaled Student's *t*-distribution[41], to the wells containing controls. We combined *P* values for each drug and strain across replicates using Fisher's method, and calculated the false-discovery rate using the Benjamini–Hochberg method over the entire matrix.

To compare drug effects between *P. vulgatus* WT and *P. vulgatus* ΔBVU_1672-1675, we used an updated version of the Prestwick library, which contains 1,520 drugs. We performed the screen as described for the pathogens and calculated the normalized median AUC per drug–strain combination[1,21]. We defined compounds that reduced the median AUC below 0.1 as hit compounds and compared total hit counts and counts by class (antibiotic or non-antibiotic drug).

### Targeted gene deletions in *P. vulgatus*

We generated genomic knockouts of target genes in *P. vulgatus* as described elsewhere[42]. In brief, the method we used relies on a two-step allelic exchange by homologous recombination. For this, the regions flanking the gene of interest 1,500 bp upstream and downstream were amplified from genomic DNA and cloned into the linearized anhydrotetracycline (aTc)-inducible suicide vector pLGB13 using HiFi–Gibson assembly. This plasmid contains an ampicillin-resistance cassette for maintenance in *E. coli*, an erythromycin-resistance cassette as a selection marker in Bacteroidota species and the aTc-inducible ssBfe1 counter selection cassette, which will express the highly toxic effector Bfe1 from the type VI secretion system of *Bacteroides fragilis*[43]. We transformed the vector that contained the flanking regions into *E. coli* DATC[44] using heat-shock transformation. We then conjugated the plasmid into *P. vulgatus*, where it integrates into the chromosome at the site of the gene of interest through homologous recombination under erythromycin selection. Using aTc counterselection, which induces ssBfe1-mediated rapid cell death, colonies that underwent a second homologous recombination, thereby losing the integrated plasmid again, were selected. We analysed colonies by PCR and Sanger sequencing to verify whether they were WT revertants or knockouts.

### Targeted gene deletions in *S.* Tm

We introduced Δ*frdD*::*aphT* into *S.* Tm SB300 from *S.* Tm 14028 Δ*frdD*::*aphT*[45] through P22 phage transduction and subsequent selection on kanamycin. Successful phage transduction was confirmed by PCR using tag-specific primers (aphT_fwd: 5′-CTGGCTGCTATTGGGCGAAG-3′; frdD_rev: 5′-GATTCACATCTTGGACCGCC-3′).

### Drug selection

We selected drugs for the in vitro challenge assay on the basis of their direct inhibitory effect on members of Com20 in monocultures[1]. We aimed to identify drugs with different inhibition profiles across the 20 species so that we could generate communities with sufficient compositional variation. We performed hierarchical clustering (Euclidean distance metric and complete linkage method) using the normalized AUC values of the 172 drugs that showed significant inhibition (adjusted *P* < 0.01) against at least 5 of the 20 species in Com20. From these clusters, we selected 63 drugs that represented diverse inhibition spectra across Com20 members (Extended Data Fig. 1a). We also included 3 drugs not part of the Prestwick library, which resulted in a panel of 65 drugs for the initial assays in Com20 members. This set was further

streamlined as we progressed through the experiments, as illustrated in Extended Data Fig. 1a. None of the drugs interfered with the luminescence readout of the assay. We excluded drugs that directly inhibited *S*. Tm growth with $IC_{25}$ values < 5 µM (nalidixic acid and norfloxacin; Supplementary Table 2). We further excluded β-lactam antibiotics owing to the presence of ampicillin resistance on the pilux luminescence plasmid used in the *S*. Tm invasion assay.

## $IC_{25}$ determination

We dissolved all drugs in DMSO, except for clomipramin, doxorubicin and tobramycin, which were dissolved in water. We prepared drug master plates at a concentration 100 times the working concentration by serially diluting the stock solutions 2-fold in DMSO or water. We diluted column-wise in V-bottom 96-well plates (Greiner Bio-One, 651261), starting from 160 mM. Each column in the plate contained eight twofold dilutions of a drug, except for column 7, which contained DMSO or water as a control. This strategy resulted in 11 drugs screened per plate. We diluted the plates to 2 times the assay concentration in 50 µl mGAM in U-bottom 96-well plates (Thermo Scientific, 168136) and stored them at −20 °C for a maximum of 1 month. Before the assay, we thawed and pre-reduced the plates overnight in an anaerobic chamber.

Monocultures or stool-derived communities[21] were grown overnight in 5 ml mGAM. The next day, we diluted the communities to an $OD_{578}$ of 0.02. We then added 50 µl of the suspension to the drug plates to result in a starting $OD_{578}$ of 0.01 and a DMSO concentration of 1% in all wells. We sealed plates with Breathe-Easy membranes (Sigma-Aldrich, Z380059). Growth curves at $OD_{578}$ were monitored every hour after 1 min of linear shaking under anaerobic conditions using an Epoch2 microplate reader coupled with a Biostack 4 microplate stacker (both Agilent) housed in a custom-made incubator (EMBL workshop[21]). We analysed at least three biological replicates for each species.

To calculate the AUC, we used the R package neckaR (https://github.com/Lisa-Maier-Lab/neckaR), using control wells in the plate that did not contain any drugs to define normal growth. We calculated the median AUC for each concentration across the three replicates. To conservatively remove the effects of noise, we enforced monotonicity. If the AUC decreased at lower concentrations, it was set to the highest AUC measured at higher concentrations. The $IC_{25}$ was defined as the lowest concentration at which a median AUC < 0.75 was observed.

## Assessment of correlations between phylogenetic relatedness and responses to bacterial and non-antibiotic drugs

To evaluate the association between the response of microorganisms to antibiotics and non-antibiotics in an evolutionary context, we used the newly obtained AUCs from the Prestwick library screen, together with those of previously reported commensal bacteria[1], to determine the similarity of the response between bacterial species, as measured by the Euclidean distance. Only drugs that inhibited the growth of >5 species were included in the analysis. We used the R package ape (v.5.8)[46] to calculate the cophenetic distances between species; the phylogeny was reconstructed using a multilocus alignment obtained from whole bacterial genomes using phylophlan (v.3.0)[47]. Principal coordinate plots were generated from Euclidean distance matrices. We tested the global association between AUC and phylogenetic distances using the Mantel test as implemented in the R package Ade4 (v.1.7-22)[48]. We further assessed the change in the strength of the association across evolutionary distances using a phylocorrelogram with the R package phylosignal (v.1.3.1)[49], which shows the correlation between the overall response to the drugs and the phylogenetic distance between species as a function of the phylogenetic distance.

## Prediction of antimicrobial resistance and stress-related genes in pathogens and gut commensals

To assess the genetic repertoire of antimicrobial resistance and stress response genes in gut commensals and pathogens, we first performed gene calling from whole genome sequences using FragGeneScan (v.1.31)[50]. We identified genes involved in antimicrobial resistance, efflux and stress responses using AMRFinderPlus (v.3.11)[51] and the database (v.2023-09-26.1) with amino acid sequences as input. We then used argNorm (v.0.2)[52] to normalize the antibiotic-resistance gene annotations.

## Preparation of drug master plates for in vitro invasion assays for *S*. Tm

For each drug, we tested five concentrations. Drugs with reported intestinal concentrations exceeding 20 µM (ref. 1) were screened at concentrations from 10 to 160 µM, and the remaining drugs were screened at concentrations from 2.5 to 40 µM.

We prepared master plates in V-bottom 96-well plates at 100-fold the drug working concentration in DMSO as described above. Concentration gradients of the drugs (16 mM, 8 mM, 4 mM, 2 mM and 1 mM or 4 mM, 2 mM, 1 mM, 0.5 mM and 0.25 mM) were represented by each column, with rows B and G having the highest and lowest concentration, respectively. In each deep-well plate, row E served as solvent controls that contained only DMSO or water. To prepare the 96-deep-well plates (Thermo Fisher Scientific, AB-0564) for the *S*. Tm challenge assay, we transferred 5 µl of the drug master plate to the deep-well plate, which already contained 95 µl mGAM. Subsequently, we pre-reduced the deep-well plates overnight (5 times the drug working concentration in 5% DMSO) in an anaerobic chamber. Wells on the border contained only mGAM (sterile controls).

## Assembly of Com20 and Com21 for in vitro invasion assays for *S*. Tm

For Com20 and Com21 assembly, we inoculated each member from frozen stocks and cultured them anaerobically in 5 ml mGAM over two overnight passages (1:100) as monocultures. We measured the $OD_{578}$ individually for each species. We mixed together the cultures in the volume required to achieve a total $OD_{578}$ of 0.0125 (for example, in Com20, each species contributed an equal $OD_{578}$ of 0.000625) and 400 µl of this suspension was added to wells of 96-deep-well plates that contained drugs as described above to achieve a starting $OD_{578}$ of 0.01 (total volume of 500 µl).

We sealed the deep-well plate with the drugs and communities with a Breathe-Easier membrane (Sigma-Aldrich, Z763624) and incubated them anaerobically at 37 °C for 24 h. The 24 h of incubation with drugs disrupted the composition of the communities, and we used the disrupted communities for the in vitro *S*. Tm challenge assay. We obtained pellets from 300 µl of the cultures, which we then froze for 16S rRNA gene analysis.

## In vitro *S*. Tm challenge assay

For the luminescence-based invasion assay, we used the human gut pathogen *S*. Tm strain SB300 (ref. 53) with the plasmid pIJ11282 *ilux* (pRS16591, *S*. Tm pilux; a gift from the Foster Laboratory, University of Oxford) for constitutive expression of the *ilux* operon under the *nptII* promoter[54]. *S*. Tm pilux was grown anaerobically at 37 °C overnight in mGAM supplemented with 100 µg ml$^{-1}$ ampicillin and then subcultured by diluting 1:100 in the same medium. The next day, we measured the $OD_{578}$ of 100 µl of all drug-perturbed communities in a 96-well clear, flat-bottom plate. To assess the growth potential of *S*. Tm pilux in the drug-perturbed communities, we transferred 50 µl from each well of the drug-perturbed communities into new pre-reduced, deep-well plates. We diluted *S*. Tm pilux to an $OD_{578}$ of 0.0025 and added 200 µl of this suspension to the assay deep-well plate. We added 250 µl mGAM so that the total volume was 500 µl, which resulted in a starting $OD_{578}$ for *S*. Tm of 0.001 and for the untreated community of 0.5. Of note, this protocol resulted in the transfer of residual amounts of the drug, up to 10% of the original concentration. We sealed the assay plate with a Breathe-Easier membrane (Sigma-Aldrich, Z763624) and incubated

it anaerobically at 37 °C for 4.5 h. Thereafter, the plate was taken out of the anaerobic chamber. We thoroughly mixed the contents of the wells and added 25 μl of 2 mg ml$^{-1}$ chloramphenicol to each well to halt *S*. Tm growth and to stabilize the luminescence signal. The cell suspension (100 μl) was transferred to a white 96-well plate (Thermo Fisher, 236105). Approximately 10 min later, the plate was incubated for 10 min at 37 °C in a Tecan Infinite 200 PRO microplate reader and luminescence was measured.

We obtained two measurements: the OD$_{578}$ of communities after overnight incubation with the drugs, and the luminescence emitted by *S*. Tm as a proxy for pathogen growth in the drug-perturbed communities. For data analysis, OD$_{578}$ values were first corrected by subtracting the baseline OD$_{578}$ from mGAM. Then, we normalized the luminescence and OD$_{578}$ values to the control column in row E, which contained the unperturbed community (solvent controls) by dividing values of perturbed communities by values of untreated communities. Both *S*. Tm luminescence and Com20 OD$_{578}$ were highly correlated among the three replicates ($R^2 = 0.56-0.74$ and $R^2 = 0.85-0.9$, respectively; Extended Data Fig. 4e).

## Post-wash *S*. Tm challenge assay
We evaluated whether washing drug residue from the community after treatment affected *S*. Tm growth. The post-wash *S*. Tm challenge assay was conducted following the same protocol as the *S*. Tm challenge assay, with the modification that the procedure was carried out in 1.5 ml Eppendorf tubes. We tested the following drugs: 20 μM clotrimazole, 80 μM zafirlukast, 160 μM chlorpromazine and 80 μM terfenadine from the colonization group *S*. Tm neutral; 80 μM clomiphene, 20 μM floxuridine, 20 μM erythromycin and 80 μM sertindole from the colonization group *S*. Tm favouring; and 1% DMSO as a control. Each condition was carried out in two Eppendorf tubes. After 24 h of drug treatment, we centrifuged one tube of each condition for 5 min at 3,000 *g* at room temperature under anaerobic conditions. The supernatant was removed and the pellet was resuspended in 500 μl mGAM. Next, we transferred 50 μl of each tube of each condition in triplicate to a deep-well plate containing 250 μl mGAM to compare the washed and the unwashed culture. Finally, *S*. Tm was added and the luminescence was measured after 4.5 h. Signals were normalized to *S*. Tm growing in the DMSO-treated community.

## Pairwise co-culture and single-species dropout assays
We conducted pairwise co-culture assays to measure the contribution of each member of Com20 to *S*. Tm growth individually. The commensal species and *S*. Tm pilux were grown anaerobically overnight in mGAM and subcultured once before the experiment. On the following day, we mixed the commensal and *S*. Tm pilux in 96-deep-well plates with a total volume of 500 μl mGAM. We set the initial OD$_{578}$ of the commensal to 0.1, whereas *S*. Tm had an initial OD$_{578}$ of 0.0002 (commensal to pathogen ratio of 500:1), as described above for the *S*. Tm challenge assay in Com20. Control wells contained only *S*. Tm in monoculture. After a growth period of 4.5 h at 37 °C under anaerobic conditions, we calculated *S*. Tm levels as described for the in vitro *S*. Tm challenge assay.

The single-species dropout assay was performed in a similar way as for the in vitro *S*. Tm challenge assays. We assembled 19-member communities by omitting one strain at a time in the volume required to achieve a total OD$_{578}$ of 0.5. Then, we mixed 800 μl of this suspension with 200 μl 50% glycerol (with a few crystals of palladium black (Sigma-Aldrich)). Communities preserved in 1.8 ml cryovials (Thermo Scientific NUNC, 10674511) were frozen at −80 °C and grown overnight twice anaerobically at 37 °C before conducting the pathogen challenge assay. We normalized the growth of *S*. Tm on dropout communities to the growth of the pathogen in Com20.

## In vitro Transwell *S*. Tm challenge assay
To analyse the transcriptional profile of both the community and *S*. Tm after drug treatment, we performed the *S*. Tm challenge assay in a Transwell format. For this, we selected treatments that led to a low biomass of Com20 (160 μM terfenadine and 80 μM clomiphene) and drugs that affected community composition but not biomass (20 μM floxuridine and 80 μM simvastatin). Com20 and *S*. Tm were inoculated from a cryostock in 5 ml mGAM and incubated anaerobically overnight at 37 °C. We diluted the drugs and the control solvent (DMSO) in mGAM in glass tubes in a total volume of 4.5 ml. After overnight incubation, we measured the OD$_{578}$ of Com20. Then, 500 μl Com20 at an OD$_{578}$ of 0.05 was added to tubes containing mGAM and drug or solvent, which resulted in a total volume of 5 ml. This setup ensured a final drug concentration of 1× and an initial Com20 OD$_{578}$ of 0.005. We incubated drug-treated Com20 anaerobically for 24 h at 37 °C; *S*. Tm was subcultured. The next day, we transferred Com20 and *S*. Tm to 6-well cell culture plates (Greiner, 657160) in a total volume of 7 ml per well. We added 2.3 ml mGAM to all wells and transferred 700 μl of each drug-treated community in triplicate to the 6-well cell culture plate. We placed the 6-well cell culture insert (CellQART, 0.4 μm, 9300402) in each well and filled all inserts with 4 ml *S*. Tm with an initial OD$_{578}$ of 0.001. Plates were anaerobically incubated at 37 °C. After 4.5 h, we combined the 3 technical replicates in one 15 ml Falcon tube of either *S*. Tm or the drug-treated community to reach a minimum of 10$^9$ cells per tube. We took out the tubes from the anaerobic chamber to centrifuge them at 4,300 *g* for 20 min at 4 °C. After centrifugation, we placed the tubes immediately on ice, removed the supernatant and added 1 ml TRIzol (Invitrogen by Thermo Fisher Scientific, 15596026). After vortexing the tubes, we transferred their content to a 2 ml Eppendorf tube before leaving the pellet for 10 min at room temperature and freezing it at −80 °C. We repeated the experiment three times across three different weeks.

Samples were sent to Novogene for RNA isolation and sequencing. In brief, total RNA was extracted using an in-house RNA purification kit and ribosomal RNA was removed using a Ribo-Zero Plus rRNA depletion kit (Illumina) followed by ethanol precipitation. After fragmentation, the first-strand cDNA was synthesized using random hexamer primers. During the second-strand cDNA synthesis, dUTPs were replaced with dTTPs in the reaction buffer. Directional libraries were generated using a Novogene NGS Stranded RNA Library Prep Set, which involved end repair, A-tailing, adapter ligation, size selection, USER enzyme digestion (New England Biolabs), amplification and purification. Libraries were sequenced using an Illumina Novaseq X Plus-PE150 platform.

## Quantification of *S*. Tm in treatment-mimicking communities
To validate our screen, we selected four conditions: treatment with erythromycin, floxuridine, sertindole or zafirlukast. On the basis of the composition of Com20 after treatment with these drugs, we assembled treatment-mimicking communities that contained only the members with a mean relative abundance of ≥3% after 24 h of drug exposure. We incubated Com20 and treatment-mimicking communities in deep-well plates at 37 °C anaerobically. After 24 h, we performed a dilution series of these communities in deep-well plates in a total volume of 400 μl. We transferred 100 μl of each dilution and Com20 to flat-bottom plates and measured the OD$_{578}$. Fifty microlitres was transferred to new deep-well plates containing 250 μl mGAM per well. In addition, 200 μl *S*. Tm pilux (OD$_{578}$ 0.0025) was added to each well and the plate was incubated for 4.5 h at 37 °C. We retained the remaining volume of the dilution series for DNA isolation and 16S rRNA gene sequencing. After 4.5 h, we measured *S*. Tm luminescence as described above. We performed the experiment in triplicate, and the luminescence measurements in treatment-mimicking communities were normalized to the luminescence in Com20. We compared the log$_2$[fold change] of *S*. Tm luminescence and the OD$_{578}$ between treatment-mimicking communities and drug-treated communities to identify the dilution step that best matched the drug-treated community.

## S. Tm WT and S. Tm ΔfrdD::aphT competition assay

To test whether fumarate respiration has an essential role in outcompeting a close niche competitor, we performed a competition experiment between *E. coli* ED1α and *Salmonella*. *E. coli* ED1α, *S.* Tm SB300 WT, *S.* Tm SB300 Δ*frdD::aphT* and *S.* Tm SB300 WITS-tag (kanamycin resistant[55]), which were inoculated anaerobically for 2 nights at 37 °C. Next, we measured the $OD_{578}$ of all bacteria and mixed them in a total volume of 5 ml. *E. coli* ED1α was added to every condition with an initial $OD_{578}$ of 0.5. In condition one, we tested the growth of *Salmonella* by adding *S.* Tm WT and *S.* Tm WITS-tag with an initial $OD_{578}$ of 0.0005. In condition two, we tested the growth of *S.* Tm lacking the subunit D of fumarate reductase by adding *S.* Tm WT and *S.* Tm Δ*frdD::aphT* with an initial $OD_{578}$ of 0.0005. We incubated tubes at 37 °C anaerobically for 24 h before plating out on selective agar. Condition one was plated out on LB agar with streptomycin (50 µg ml⁻¹; from herein on LBStrep) to obtain all *Salmonella* counts and on LB agar, streptomycin (50 µg ml⁻¹) and kanamycin (30 µg ml⁻¹, henceforth LBStrepKan) to obtain counts from *S.* Tm WITS-tag. Condition two was plated out on the same selective agar plates; although LBStrep was used to obtain all *Salmonella* counts, on LBStrepKan, only *S.* Tm Δ*frdD::aphT* was able to grow. We subtracted LBStrepKan counts from LBStrep counts to obtain *S.* Tm WT and calculated their ratio. The experiment was performed in six replicates.

To verify our findings in a community context, we selected streptozotocin (40 µM), which targets *E. coli* ED1α, tiratricol (160 µM), which does not affect *E. coli* ED1α, and DMSO as a control solvent. We used Com20, which served as a simple synthetic community, Com21, because it mirrors Com20 but contains a niche competitor (*E. coli* ED1α) and the stool-derived community, because of its complexity. All three communities were inoculated from cryogenic stock in 5 ml mGAM anaerobically and incubated for one night at 37 °C. *S.* Tm WT and *S.* Tm Δ*frdD::aphT* were inoculated from plates in 5 ml mGAM anaerobically and incubated for one night at 37 °C. We added drugs and control solvent (DMSO) to a total amount of 4.5 ml mGAM in glass tubes. The next day, the $OD_{578}$ of all communities was determined, and communities were diluted to an initial OD of 0.005 and added to tubes containing mGAM and drugs or solvent in a total volume of 5 ml, which resulted in a 1× drug concentration. Drug-treated communities were incubated at 37 °C for 24 h anaerobically and *Salmonella* was subcultured. After 24 h, drug-treated communities were 10× diluted into fresh mGAM and the $OD_{578}$ of *S.* Tm was determined. We added *S.* Tm WT and *S.* Tm Δ*frdD::aphT* to each drug-treated community with an initial OD of 0.0005 and incubated these tubes for 24 h at 37 °C anaerobically. The following day, we plated each condition on either LBStrep to obtain all *S.* Tm counts or on LBStrepKan to count only *S.* Tm Δ*frdD::aphT*. We subtracted *S.* Tm Δ*frdD::aphT* counts from all *S.* Tm counts to obtain *S.* Tm WT counts and calculated the ratio of *S.* Tm WT/*S.* Tm Δ*frdD::aphT*. The experiment was repeated 5–6 times.

## Plasmid transformation of pathogenic Enterobacteriaceae

We incubated bacterial strains overnight in 6 ml LB medium at 27 °C (WA-314, YpsIII) or 37 °C (Kp MKP103, Ec CFT073). We centrifuged the overnight cultures for 5 min at 4,000*g*, washed the pellets twice with 5 ml of 300 mM sucrose solution, transferred 1 ml of 300 mM sucrose solution to an Eppendorf cap and centrifuged for 1 min at 10,000*g*. The supernatant was removed, the bacteria were resuspended in 100 µl of 300 mM sucrose solution and transferred to a Gene Pulser cuvette (0.2-cm electrode gap, Bio-Rad), and 100 ng of plasmid DNA (pEB1GM or pEB2GO, synthesized by GenScript) was added. Subsequently, we carried out electroporation using a Gene Pulser (Bio-Rad) and immediately added 1 ml LB. The bacterial suspension was then shaken at the corresponding temperature for 1 h and plated on LB gentamicin plates (15 µg ml⁻¹ for WA-314, YpsIII and Kp Ec CFT073; 75 µg ml⁻¹ for Kp MKP103) overnight. We verified the success of the electroporation by measuring chemiluminescence of the lux reporter.

## Adaptation of the S. Tm challenge assay to other pathogens

We screened other Gammaproteobacteria species in a similar manner to *S.* Tm in the challenge assay described above. We prepared drug master plates in the same way, except that we tested 10 drugs and the master plate concentration ranged from 10 mM to 1 mM. Moreover, only the outer rows were left empty to serve as medium controls. We tested post-treatment expansion of Gammaproteobacteria species in Com20, which we assembled as described above. For the luminescence-based assay, we used the human gut pathogens *E. coli* CFT073, *K. pneumoniae* MKP103, *S. flexneri* 24570, *Y. enterocolitica* WA-314, *Y. pseudotuberculosis* YPIII and *V. cholerae* A1552. With the exception of *V. cholerae*, all pathogens contained a variant of the pilux plasmid that enabled constitutive expression of the lux reporter. We incubated all pathogens anaerobically overnight at 37 °C in mGAM supplemented with 100 µg ml⁻¹ ampicillin (*S. flexneri*), 15 µg ml⁻¹ gentamicin (*E. coli*, *Y. enterocolitica* and *Y. pseudotuberculosis*) or 75 µg ml⁻¹ gentamicin (*K. pneumoniae*) and then subcultured by diluting 1:100 in the same medium. We proceeded as for the in vitro *S.* Tm challenge assay but incubated the plates at 37 °C for a species-specific amount of time (4.5 h for *E. coli*, 5 h for *S. flexneri* and *K. pneumoniae*, 5.5 h for *V. cholerae* and 7 h for *Y. enterocolitica* and *Y. pseudotuberculosis*).

To measure the growth of *V. cholerae*, we serially diluted the plates ($10^1$–$10^8$-fold) in PBS and selectively plated aerobically on LB agar with 100 µg ml⁻¹ ampicillin for pathogen enumeration. For the other pathogens, we measured their growth as described for *S.* Tm pilux with a Tecan Infinite 200 PRO microplate reader. For each treatment, we obtained two measurements: the $OD_{578}$ of communities after overnight incubation with the drugs and the luminescence emitted by the pathogens (CFU in the case of *V. cholerae*) as a proxy for pathogen growth in the drug-perturbed communities. For data analyses, both the luminescence (CFU for *V. cholerae*) and $OD_{578}$ values were normalized to the median of the controls in row E, which contained the unperturbed community (solvent controls).

We did not evaluate the effect of washing the community after drug treatment but before pathogen introduction given the results we obtained on a similar experiment using *S.* Tm (see the section 'Post-wash *S.* Tm challenge assay'). Moreover, even when the $IC_{25}$ of the pathogens was low, such as in the case of floxuridine (Supplementary Table 2), community treatment at the highest concentrations of the compound led to an increased pathogen growth (Supplementary Table 5), despite the potentially disrupting effect of the residual drug.

## General statistical analyses

We used R (v.4.2.0) for data processing and formatting. The package ggplot2 (v.3.5.1) was used for visualization. For hypothesis testing, *t*-tests, Kruskal–Wallis tests and Wilcoxon tests were performed as implemented in the package Rstatix (v.0.7.2).

## Analysis of community composition using 16S rRNA gene amplicon sequencing

DNA was extracted from pellets of 300 µl culture using a DNeasy UltraClean 96 Microbial kit (Qiagen, 10196-4) or from whole faecal pellets using a DNeasy PowerSoil HTP 96 kit (Qiagen, 12955–4). Library preparation and sequencing was performed at the NGS Competence Center NCCT. Genomic DNA was quantified with a Qubit dsDNA BR/HS Assay kit (Thermo Fisher) and adjusted to 100 ng input for library preparation. The first step PCR was performed in 25 µl reactions that included KAPA HiFi HotStart ReadyMix (Roche), 515F[56] and 806R[57] primers (covering about 350-bp fragment of the 16S V4 region) and template DNA (PCR program: 95 °C for 3 min, 28× (98 °C for 20 s, 55 °C for 15 s, 72 °C for 15 s), 72 °C for 5 min). Initial PCR products were

purified using 28 µl AMPure XP beads and eluted in 50 µl of 10 mM Tris-HCl. Indexing was performed in a second step PCR that included KAPA HiFi HotStart ReadyMix (Roche), index primer mix (IDT for Illumina DNA/RNA UD Indexes, Tagmentation) and purified initial PCR product as template (PCR program: 95 °C for 3 min, 8× (95 °C for 30 s, 55 °C for 30 s, 72 °C for 30 s), 72 °C for 5 min). After another round of bead purification (20 µl AMPure XP beads, eluted in 30 µl of 10 mM Tris-HCl), the libraries were checked for correct fragment length on an E-Base device using E-Gel 96 Gels with 2% mSYBR Safe DNA gel stain (Fisher Scientific), quantified with a QuantiFluor dsDNA system (Promega) and pooled equimolarly. The final pool was set to 4 nM (Illumina standard value) before being brought to a loading concentration of 8 pM. The pool was sequenced on an Illumina MiSeq device with a v.2 sequencing kit (input molarity 10 pM, 20% PhiX spike-in, 2 × 250 bp read lengths).

### Computational processing of 16S rRNA amplicon sequences

We used the R package DADA2 (v.1.21.0)[58] following its standard operating procedure available from GitHub (https://benjjneb.github.io/dada2/bigdata.html). In brief, after inspecting the quality profiles of the raw sequences, we trimmed and filtered the paired-end reads using the following parameters: trimLeft: 23, 24; truncLen: 240, 200; maxEE: 2, 2; truncQ: 11. The filtered forward and reverse reads were de-replicated separately and used for inference of amplicon sequence variants (ASVs) using default parameters, after which the reads were merged on a per-sample basis. Next, we filtered the merged reads to retain only those with a length between 250 and 256 bp and carried out chimera removal.

We performed the taxonomic assignment in two steps. First, the final set of ASVs was classified up to the genus level using a curated DADA2-formatted database based on the genome taxonomy database (GTDB)[59] (release R06-RS202; available at https://scilifelab.figshare.com/articles/dataset/SBDI_Sativa_curated_16S_GTDB_database/14869077). Next, ASVs belonging to genera expected to be in Com20 were further classified at the species level using a modified version of the aforementioned database that contained only full-length 16S rRNA sequences of the 20 members of the synthetic community. The sequence of each ASV was aligned against this database using the R package DECIPHER (v.2.24.0)[60]; we classified an ASV as a given species if it had sequence similarity of >98% to the closest member in the database. The abundance of each taxon of Com20 was obtained by aggregating reads at the species level. ASVs from in vitro communities and gnotobiotic mice were classified using the two-step processes; ASVs from SPF mice were classified using only the first step. We removed potential contaminant sequences from SPF mouse samples using the permutation filtering method implemented in the R package PERFect (v.1.14.0)[61].

### Overlap of pathways encoded by Com20 and Com21 and human gut metagenomes

We used the 16S rRNA gene abundance from control Com20 and Com21 in vitro communities and untreated gnotobiotic mice samples to predict the metabolic potential of the microbial communities using PICRUSt2 (v.2.4.1)[62]. As the composition of the synthetic communities is known, we retrieved the full-length sequences of the 16S rRNA gene for each of the member species and used them together with the species abundance data to predict metagenome functions. For our analyses, we used MetaCyc pathway abundances. We compared the number of metabolic pathways detected in the untreated in vitro and in vivo synthetic communities to actual human gut metagenomes. For this, we retrieved publicly available tables of MetaCyc pathway abundances processed using HUMAnN2 from GitHub (https://github.com/gavinmdouglas/picrust2_manuscript/tree/master/data/mgs_validation). These tables comprised 156 samples from the Human Microbiome Project[63] and 57 samples from Cameroon[64]. For each set

of samples, we considered a pathway as present if it was detected in ≥20% of samples.

### Classification of drug treatments according to *S*. Tm growth

We grouped treatments according to their effect on Com20 in vitro. To do so, we calculated the mean normalized luminescence and the 95% confidence interval (CI) of each drug–concentration combination. A treatment was classified as *S*. Tm favouring if its mean normalized luminescence was >2 and the 95% CI did not span 2, whereas the treatment was classified as *S*. Tm restricting if the mean normalized luminescence was <0.5 and the 95% CI did not span 0.5. The treatment was classified as *S*. Tm neutral if the mean normalized luminescence was between 0.5 and 2. Communities with a normalized $OD_{578}$ < 0.2 were also classified but marked for removal in downstream analyses to minimize the bias introduced by low-biomass samples.

### Assessment of drug treatment on the composition of synthetic communities in vitro

To assess changes in microbial diversity after drug treatment, we calculated the species richness and Shannon's index using the R package vegan (v.2.6-8)[65]. Tables of taxa abundances were rarefied to an even sequencing depth. We evaluated the association between *S*. Tm luminescence, $OD_{578}$ of Com20 and these diversity measures by fitting linear models, with luminescence as the response variable. Our linear models evaluated each response variable separately or included $OD_{578}$ and one diversity measure as the main effects. We estimated the proportion of variance explained by the models using the adjusted coefficient of determination ($R^2$) and assessed the goodness-of-fit of each model with the Akaike information criterion.

We assessed differences in multivariate homogeneity of group dispersions between colonization groups (that is, *S*. Tm favouring, *S*. Tm restricting and *S*. Tm neutral) and untreated controls, and calculated differences in beta diversity with a PERMANOVA test on Bray–Curtis distance matrices using vegan (v.2.6-8). We performed pairwise PERMANOVA tests contrasting each treatment group to untreated controls accounting for normalized $OD_{578}$ in the models; the $OD_{578}$ value of control samples was set to 1. *P* values were adjusted using the Benjamini–Hochberg method and a significance threshold of 0.1 was used.

We next assessed differences in the abundance of individual microbial species between colonization groups (that is, *S*. Tm favouring, *S*. Tm restricting and *S*. Tm neutral) and untreated controls. We transformed the unrarefied ASV abundances using the centred log-ratio to account for the compositional nature of the sequencing data. Positive centred log-ratio values imply that an ASV is more abundant than average; conversely, negative values imply that the ASV is less abundant than average. Low-biomass samples were removed. Next, we fitted linear regression models to determine which species were differentially abundant between each colonization group and controls using the R package MaAsLin2 (v.1.13.0)[66]. We included normalized $OD_{578}$ as a covariate to account for the biomass of the community, with the $OD_{578}$ values of control samples set to 1. *P* values were adjusted using the Benjamini–Hochberg method and a significance threshold of 0.1 was used.

### Evaluation of metabolic overlap between *S*. Tm and members of the synthetic community

We estimated potential niche overlap between *S*. Tm and each member of Com21 by calculating the competition and complementarity indices using PhyloMint (v.0.1.0)[67]. The metabolic competition index is a proxy of the metabolic overlap of two species; this index is a nonsymmetric measure and it is calculated on the basis of the number of compounds required but not synthesized by both species[68]. Conversely, the metabolic complementarity index is a proxy for potential syntrophy between species; this index is a nonsymmetric measure calculated

based on the number of compounds that one species produces that the second species requires but cannot synthesize[68]. In brief, PhyloMint takes as input the whole genome sequence of each strain, which it uses to obtain a genome-scale metabolic model with CarveMe (v.1.5.1)[69], extracts the metabolite seed sets and calculates the competition and complementarity indices.

### Computational processing and analysis of *S*. Tm and Com20 transcriptomes

We used the nf-core taxprofiler pipeline (v.1.2)[70] to pre-process and taxonomically classify the raw reads. In brief, the pipeline used fastp (v.0.23.4)[71] for adapter trimming and complexity filtering and Bowtie (v.2.5.2)[72] for the removal of eukaryotic contaminant reads, including the human genome (references retrieved from Zenodo: https://doi.org/10.5281/zenodo.4629921). Then, clean reads were taxonomically classified using Kraken2 (v.2.1.3)[73] and Bracken (v.2.9)[74] against a GTDB-formatted database based on the Unified Human Gut Genome catalogue[75] (available at http://ftp.ebi.ac.uk/pub/databases/metagenomics/mgnify_genomes/human-gut/v2.0.2/). We retained the clean reads, which we then used as input for the nf-core metadenovo pipeline (v.1.0.1; available at https://nf-co.re/metatdenovo/), using as the mapping reference the whole genome sequences of all Com20 members or *S*. Tm (Supplementary Table 14). In brief, metadenovo performed gene calling on the whole genomes used as the reference using prodigal, mapped the reads against the predicted genes using BBmap (v.39.01) from the BBTools suite (available at https://sourceforge.net/projects/bbmap/), obtained counts per each gene present in the assembly using the package Subread (v.2.0.1; https://subread.sourceforge.net/) and annotated the genes using EggNOG mapper (v.2.1.9)[76] and KOfamscan (v.1.3.0)[77].

For *S*. Tm, we assessed the distribution of transcript abundances across treatments using the TPM values. We then used DESeq2 (v.1.44.0)[78] to identify genes differentially expressed between each treatment and the unperturbed community (DMSO controls). We performed log[fold change] shrinkage to account for genes with low expression and high variability. We considered a gene to be differentially expressed if the absolute value of $\log_2$[fold change] was >0.585 and the *s* value was <0.01. We then used the R package clusterProfiler (v.4.12.6)[79] to carry out an overrepresentation analysis of the differentially expressed genes based on KEGG annotations. We used variance-stabilized transformed counts of the 500 most abundant genes to perform a principal component analysis.

For Com20, we assessed the distribution of transcript abundances of each species across treatments using TPM values. Given the large variation in expression levels between species across treatments, we were unable to perform a differential abundance analysis. Therefore, for each bacterium on each treatment, we performed an overrepresentation analysis with clusterProfiler and we examined the pathways represented by the 20% most expressed genes on each species based on KEGG annotations. We also determined the fraction of the top 20% highest expressed genes that were previously identified as stress-related markers in bacteria[22]. In brief, a previous study[22] assessed the gene expression profiles of 32 bacterial pathogens across 11 stress conditions; we retrieved the 'probability to be differentially expressed (PTDEX)' score from this publication and used it to classify a gene as a marker of stress if its PTDEX ≥ 0.25 in at least 6 stress conditions.

### In vivo colonization assays for *S*. Tm

Animal experiments were approved by the local authorities in Tübingen (Regierungspräsidium Tübingen, H02/20G and H02/21G). Animals were housed under a 12–12-h light–dark cycle at a temperature of $22 \pm 2\,°C$ and a relative humidity of 50–56%. Group sizes were determined using power analysis with G*Power. We used 5–6-week-old mice. Male and female mice were housed in separate cages. These cages were then randomly assigned to the experimental groups, ensuring that each group included mice of both sexes. In the in vivo experiments, blinding was not performed because drug-specific side effects needed to be evaluated. However, pathoscoring was conducted in a blinded manner by two independent assessors.

### Defined colonized and humanized mice

Germ-free C57BL/6J mice were bred in-house (Gnotobiotic Mouse Facility, Tübingen) under germ-free conditions in flexible film isolators (Zoonlab) and transferred to the Isocage P system (Tecniplast) to perform experiments. We fed the mice with autoclaved drinking water and γ-irradiated maintenance chow (Altromin) ad libitum. We kept mice in groups of 3–4 animals and tracked their health status every day.

For the Com20 model, we used both female ($n = 25$) and male ($n = 14$) mice. For the humanized model, we used three female mice and seven male mice to test the effect of the drug on the host before infection, and ten female mice and five male mice for the *S*. Tm infection. The gut microbiome of these humanized mice contained a mean of 58 ASVs before treatment.

### SPF mice

Male ($n = 54$) SPF C57BL/6J mice (632C57BL/6J) were purchased from Charles River Laboratories at the age of 35–41 days. After delivery, we kept mice in groups of 3 in individually ventilated cages for a 2-week acclimatization period. Mice were fed with autoclaved drinking water and a maintenance diet for mice (Sniff) ad libitum. We performed the experiments in a laminar flow system (Tecniplast BS60) and scored animals every day. The gut microbiome of these mice contained a mean of 199 ASVs before treatment.

### Preparation of bacterial communities and colonization of germ-free mice

We prepared Com20 under anaerobic conditions (2% $H_2$, 12% $CO_2$, the rest $N_2$) in a chamber (Coy Laboratory Products). Consumables, glassware and media were pre-reduced at least 2 days before inoculation of bacteria. We grew each strain as a monoculture overnight at 37 °C in 5 ml of their respective growth medium. The next day, we subcultured bacteria 1:100 in 5 ml fresh medium and incubated them for 16 h at 37 °C, except *Eggerthella lenta*, which was grown for 2 days. We measured the $OD_{578}$ and mixed bacteria in equal ratios to a total $OD_{578}$ of 0.5 ($OD_{578}$ of 0.025 for each of the 20 strains) in a final volume of 10 ml. After adding 2.5 ml of 50% glycerol (with a few crystals of palladium black (Sigma-Aldrich)), 200 µl aliquots were prepared in 2 ml glass vials (Supelco, Ref. 29056-U) and frozen at −80 °C. We used frozen vials within 3 months.

For the human donor-derived community, we inoculated 500 µl of the stool mixture into 100 ml mGAM and incubated it overnight at 37 °C under anaerobic conditions. The following day, we prepared 200 µl aliquots following the protocol used for Com20 and stored the vials at −80 °C. We used these aliquots within 3 months.

To colonize germ-free mice, cages were transferred to an ISOcage Biosafety Station (IBS; Tecniplast) through a 2% Virkon S disinfectant solution (Lanxess) dipping bath. We kept glycerol stocks of the frozen Com20 community or the complex human microbiome (one per mouse) on dry ice before thawing them during transfer into the IBS. We used the mixtures directly after thawing with a maximal time of exposure to oxygen of 3 min. We colonized mice by oral gavage volume of 50 µl and gavaged them again after 48 h using the same protocol. The IBS was sterilized with 3% perchloracetic acid (Wofasteril, Kesla Hygiene).

To monitor the in vivo stability of Com20 in gnotobiotic mice, we collected fresh faecal samples from every defined colonized mouse after 2, 6, 28 and 57 days after the second colonization. DNA was extracted using a DNeasy PowerSoil HTP 96 kit, and community composition was analysed by 16S rRNA amplicon sequencing, as described above.

## In vivo *S.* Tm challenge

The day before infection, we inoculated a *S.* Tm culture in LB broth supplemented with 0.3 M NaCl using colonies from a plate and grew the culture for 12 h on a rotator (Stuart, SB3, speed 9) at 37 °C. Fifty microlitres of *S.* Tm was subcultured in 5 ml LB broth supplemented with 0.3 M NaCl and incubated for 3 h in the same conditions. We washed 1 ml of the subculture twice with 1 ml of ice-cold PBS in a 2 ml Eppendorf tube by centrifugation at 4 °C and 14,000$g$ for 2 min. The pellet was resuspended in 1 ml of ice-cold PBS and kept on ice until oral administration. Mice were infected with a *S.* Tm load of $5 \times 10^6$ CFU in 50 µl PBS.

## *S.* Tm growth inhibition in defined colonized mice

To determine whether Com20 confers colonization resistance in mice, we colonized germ-free mice with Com20 for 28 days. We then treated them with 50 µl of 25% DMSO (solvent control) to test for colonization resistance levels in this mouse model. For comparison, we treated conventional SPF mice with a complex microbiome in the same manner as the mice colonized with Com20. The next day we infected all groups with 50 µl of $5 \times 10^6$ CFU of *S.* Tm. After 16–20 h, mice were euthanized by $CO_2$ and cervical dislocation, dissected and their intestinal contents were collected from the colon. We weighed the faecal samples, diluted them in buffer (2.5 g BSA, 2.5 ml Tergitol and 497.5 ml PBS) and plated the samples on MacConkey agar containing 50 µg ml$^{-1}$ streptomycin. After incubation overnight at 37 °C, we counted colonies of *S.* Tm.

## Treatment with non-antibiotic drugs and infection with *S.* Tm

Five non-antibiotic drugs were chosen on the basis of the *S.* Tm challenge assay: clotrimazole (38 mg kg$^{-1}$), zafirlukast (20 mg kg$^{-1}$), chlorpromazine (3 mg kg$^{-1}$), terfenadine (25 mg kg$^{-1}$) and clomiphene (60 mg kg$^{-1}$). We dissolved the drugs in 25% DMSO (DMSO and autoclaved drinking water), aliquoted them and stored them in 2 ml glass vials (Supelco, 29056-U) at −80 °C. For every experiment, we prepared fresh drug solutions. Defined colonized (28 days after colonization), humanized (10 days of colonization) and SPF mice were orally gavaged daily for 6 days with 50 µl non-antibiotic drug or 25% DMSO. We collected fresh faecal samples immediately before the first treatment (day 0) and after 6 days of treatment (day 6), immediately before the infection with *S.* Tm. Humanized mice were infected for 4 days, during which faecal samples were collected daily after infection.

Fifteen to twenty hours afterwards for defined and SPF mice and 4 days afterwards for humanized mice, we euthanized mice by $CO_2$ and cervical dislocation, dissected them and collected intestinal contents from colon and caecum in pre-weighed 2 ml Eppendorf tubes. After weighing the samples, we added 500 µl buffer (2.5 g BSA, 2.5 ml Tergitol and 497.5 ml PBS) and one sterile steel ball (Agrolager, RB-5/G20W) per tube. We collected half a spleen, mesenteric lymph nodes and half a liver lobe in 2-ml Eppendorf tubes containing 500 µl buffer and one steel ball. All samples were lysed with a TissueLyser II (Qiagen) for 1 min at 25 Hz. We plated intestinal contents and organs on MacConkey plates supplemented with 50 µg ml$^{-1}$ streptomycin, and in case of humanized mice, remaining intestinal content was kept at −20 °C to measure mouse lipocalin-2. We incubated the plates at 37 °C aerobically overnight and counted colonies the next day to determine the CFU per organ.

## ELISA of mouse lipocalin-2

We measured lipocalin-2 in humanized mice every day after infection (days 1–4) as a proxy for intestinal inflammation. We collected faecal samples each day after infection, which we centrifuged at 16,900$g$ for 10 min at 4 °C; we transferred the supernatant to new 1.5 ml Eppendorf tubes (see the section 'Treatment with non-antibiotic drugs and infection with *S.* Tm'). We stored the supernatant at −20 °C and thawed it no more than three times before use. For the ELISA (Mouse Lipocalin-2/NGAL, R&D Systems, DY1857-05) we used 96-well Maxisorp NUNC plates (Thermo Scientific, 439454) and performed the assay and the analysis according to the manufacturer's instructions.

## Pathoscoring

Histopathological analysis of caecal tissue was carried out as previously described[12]. The following features were scored: submucosal oedema (0–3), infiltration of polymorphonuclear neutrophils (0–4), loss of goblet cells (0–3) and epithelial damage (0–3). The individual scores were then summed to provide a final pathology score, categorized as follows: 0–3 (no inflammation), 4–8 (mild inflammation) and 9–13 (severe inflammation). Scoring was conducted by two independent, blinded examiners. Statistical analysis was performed using generalized linear mixed models (lme4 v.1.1-35.5 package), with the animal identifier included as a random effect to account for non-independence owing to score replicates. Maximum likelihood estimation was carried out using the Laplace approximation. A generalized linear model with an appropriate link function was used to accommodate log-normally distributed errors, as determined from model residuals. Model convergence was ensured using iterative derivative-free bound optimization through quadratic approximation (bobyqa). The package emmeans (v.1.10.6) was used to extract estimates of marginal means and contrasts from the model. Fixed-effects coefficients and post hoc contrasts were evaluated using two-sided Wald $z$-tests.

## Immunohistochemistry

We used 25 ml 4% paraformaldehyde solution (ROTI Histofix, Carl Roth, P087.2) to fix the caecal tissue of mice for 24 h at room temperature. The next day, tissue was transferred to a new 50 ml Falcon tube containing 25 ml fresh 4% paraformaldehyde solution and was fixed for another 24 h at room temperature. We placed the tissue in embedding cassettes before it was dehydrated, embedded in paraffin and cut to 2 µm tissue slices. Slices were deparaffinized and rehydrated, antigens were retrieved with Bond citrate solution (AR9961, Leica) and Bond EDTA solution (AR9640, Leica). We incubated the slides with primary antibodies in Bond primary antibody diluent (AR9352, Leica) followed by secondary antibodies (rabbit anti-rat IgG H&L preadsorbed; 1:1,000 (Abcam) or polymer anti-rabbit poly-HRP-IgG (Leica)) and stained using a Bond Polymer Refine Detection kit (DS9800, Leica) (Supplementary Table 13). The slides were scanned using a whole-slide scanner (Aperio AT2, Leica) at ×20 magnification. The following antibodies were used: anti-CD31 (rat, 1:40; Dako, M0823); anti-RelA (rabbit, 1:400; Novus Biological, NB100-2176); HIF1α (rabbit, 1:500; Novus Biological, NB100479); CD4 (rat, 1:1,000; Thermo Fisher, 14-9766-82); CD8 (rabbit, 1:400; Cell Signaling, 98941S); CD11b (rabbit, 1:10,000; Abcam, ab133357); CD11c (rabbit, 1:300; Cell Signaling, 97585); F4/80 (rabbit, 1:400, Cell Signaling, 70076); cleaved caspase-3 (rabbit, 1:300; Cell Signaling, 9661); Ki-67 (rabbit, 1:100; Thermo Fisher, RM-9106-S1); and B220 (rat, 1:3,000; BD, 553084). Slide scans were imported into QuPath (v.0.5.1)[80] for quality control and stain vector normalization. The caecal mucosal area (comprising epithelial cells and the lamina propria, delineated by the muscularis mucosae) was segmented to obtain approximately six representative fields of view per animal per staining. This resulted in 664 unique fields of view for the experiment without *S.* Tm infection and 898 for the infection experiment. On average, we analysed 30,610 cells per field (95% CI: 29,007–32,213) in the infection experiment and 15,759 cells per field (95% CI: 14,821–16,697) in the non-infection experiment, which totalled 22,621,076 analysed cells for the infection experiment and 8,573,164 for the non-infection experiment. For markers in which supracellular signals were of interest, Fiji (v.1.10.6) was used to generate segmentation masks, which were then used to extract morphometric data using the BioVoxxel Toolbox (v.2.6.0) extended particle analyzer[81]. The primary readout was defined as

either the number of positive cells per mm$^2$, determined through histogram-informed thresholding with additional correction for residual staining to minimize bias from sample preparation, or, for markers without a clear biological dichotomy, the quantitative characterization of pixel intensities per cell. These intensities were hierarchically summarized to derive the average cell staining intensity per field of view. To account for potential biases due to variations in tissue cellularity (for example, inflammatory cell infiltration), both readouts (positive cells per mm$^2$ and average cell staining intensity) were also analysed using the number of nuclei instead of surface area as a covariate. Generalized linear mixed models were fitted using lme4 (v.1.1-35.5) to reflect non-independence of observations due to the hierarchical structure of the data[82]. Maximum likelihood estimation was performed using the Laplace approximation and linear regression was generalized using appropriate link functions for normally or log-normally distributed errors, as determined from model residuals. Model convergence was ensured through iterative derivative-free bound optimization by quadratic approximation (bobyqa) for maximum likelihood estimation. The package emmeans (v.1.10.6) was used to extract estimates of marginal means and contrasts from the model. Fixed-effects coefficients and post hoc contrasts were evaluated using two-sided Wald $z$-tests.

## Assessment of drug treatment on the microbiome composition of gnotobiotic and SPF mice

We assessed the effect of drugs on the composition of the gut microbiome of gnotobiotic, humanized and SPF mice after 6 days of treatment and compared with untreated controls. For this comparison, we carried out analysis of covariance incorporating the abundance of each ASV at day 0 (pretreatment), thus estimating the baseline adjusted difference between groups at day 6. Analyses of covariance models were fitted using a multiple linear regression; robust standard errors were calculated using the R package sandwich (v.3.0-2)[83], which were then evaluated by a coefficient test as implemented in the R package lmtest (v.0.9-40)[84]. To account for the compositional nature of the sequencing data, we transformed ASV abundances with the centred log-ratio. $P$ values were adjusted using the Benjamini–Hochberg method and a significance threshold of 0.1 was used.

### Reporting summary

Further information on research design is available in the Nature Portfolio Reporting Summary linked to this article.

## Data availability

The 16S rRNA gene amplicon sequencing and transcriptomic data generated during this study have been deposited in the European Nucleotide Archive with accession PRJEB65315. The following databases were used in this study: Genome Taxonomy Database (GTDB) release 202 (https://data.gtdb.ecogenomic.org/releases/release202/); AMRFinderPlus database v.2023-09-26.1; DADA2-formatted ASV database based on GTDB release R06-RS20259 (https://scilifelab.figshare.com/articles/dataset/SBDI_Sativa_curated_16S_GTDB_database/14869077); Kraken2 and Bracken GTDB-formatted database based on the Unified Human Gut Genome catalogue (http://ftp.ebi.ac.uk/pub/databases/metagenomics/mgnify_genomes/human-gut/v2.0.2/). Source data are provided with this paper.

## Code availability

The R notebooks with the code used for data analyses are available from GitHub (https://github.com/Lisa-Maier-Lab/HTD_CR).

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

**Acknowledgements** The authors thank all members of the Maier Laboratory and A. Typas for discussions and comments on the manuscript; members of the Foster Laboratory for plasmid pIJ11282 *ilux*; L. Lo Presti for proofreading; R. Unni for comments on the figures and legends; C. Voogdt for support with *P. vulgatus* genetics; H. Reisinger and M. Montag for technical support; L. Michaelis, P. Ruh and J. Zent for support with the animal work; B. Stecher for help with pathoscoring; and staff at the Gnotobiotic Research Center Tübingen (GRCT) and the NGS Competence Center Tübingen (NCCT). L.M. acknowledges funding from the DFG (Cluster of Excellence CMFI EXC 2124, Emmy Noether Programme MA 8164/1-1), the ERC (gutMAP 101076967) and DZIF (CEGIMIR, TTU06.709). J.d.l.C.-Z. and L.M. received support from the de.NBI Cloud within the German Network for Bioinformatics Infrastructure (de.NBI) and ELIXIR-DE (Forschungszentrum Jülich and W-de.NBI-001, W-de.NBI-004, W-de.NBI-008, W-de.NBI-010, W-de.NBI-013, W-de.NBI-014, W-de.NBI-016, W-de.NBI-022). K.C.H. acknowledges funding from NIH RM1 GM135102 and R01 AI147023, and NSF grants EF-2125383 and IOS-2032985. K.C.H. is a Chan Zuckerberg Biohub Investigator. T.H.N. is supported by an NSF Graduate Research Fellowship. J. Homolak is supported by the Croatian Science Foundation Fellowship (MOBODL-2023-12-2591).

**Author contributions** Conceptualization: T.Z. and L.M. Investigation and formal analyses: A.G. ($IC_{25}$ determination, all variations of the in vitro *S.* Tm challenge assay, in vitro assays for validation, mimicking communities, pairwise co-culture and single-species dropout, *S.* Tm Δ*frdD*::*aphT* competition assay, all animal experiments and their corresponding streamlined analyses), J.d.l.C.-Z. (processing of amplicon and metatranscriptome data, metabolic overlap analyses, microbial genome annotation and processing, microbial ecology analyses, phylogenetic signal analyses, analysis of in vivo and in vitro invasion assays and data visualization), T.Z (in vitro assays for validation, establishment and performance of in vitro *S.* Tm assay in Com20 and stool-derived communities), P.M. (in vitro assays for validation, analysis for other pathogens and mimicking communities, *P. vulgatus* Prestwick screens and analysis of molecular properties), C.G. (assisted in all experiments and performed ELISA), H.C. (Prestwick screen for pathogens), K.S., C.P. (assistance with Com20 and Com21 in vitro *S.* Tm assays), V.S. (construction of the *P. vulgatus* knockout strain), S.G., J. Hetzer (immunohistochemistry), J. Homolak (analysis of immunohistochemistry), E.B. (generation of luminescent Gammaproteobacteria other than *S.* Tm) and T.H.N. (preparation and advice on stool-derived communities). Writing original draft: J.d.l.C.-Z., A.G., K.C.H. and L.M. Writing, review and editing: all authors. Supervision: M.H., K.C.H. and L.M. Funding acquisition: J. Homolak, K.C.H. and L.M.

**Funding** Open access funding provided by Eberhard Karls Universität Tübingen.

**Competing interests** The authors declare no competing interests.

**Additional information**
**Correspondence and requests for materials** should be addressed to Lisa Maier.

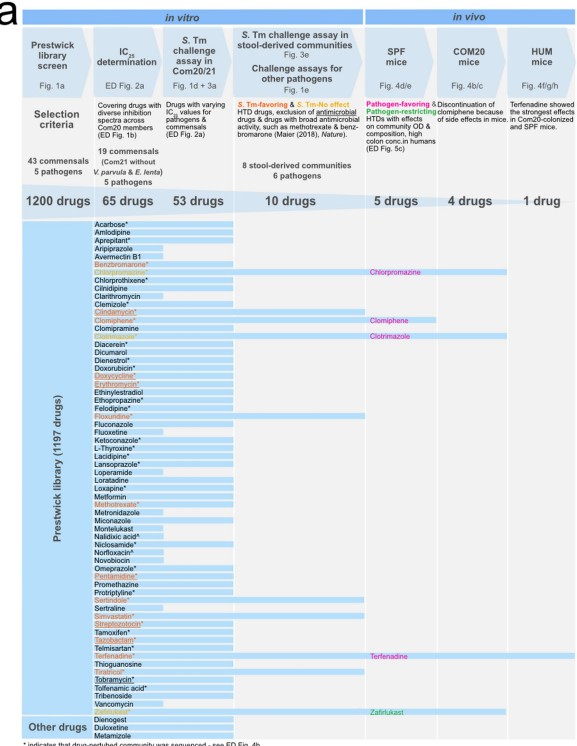

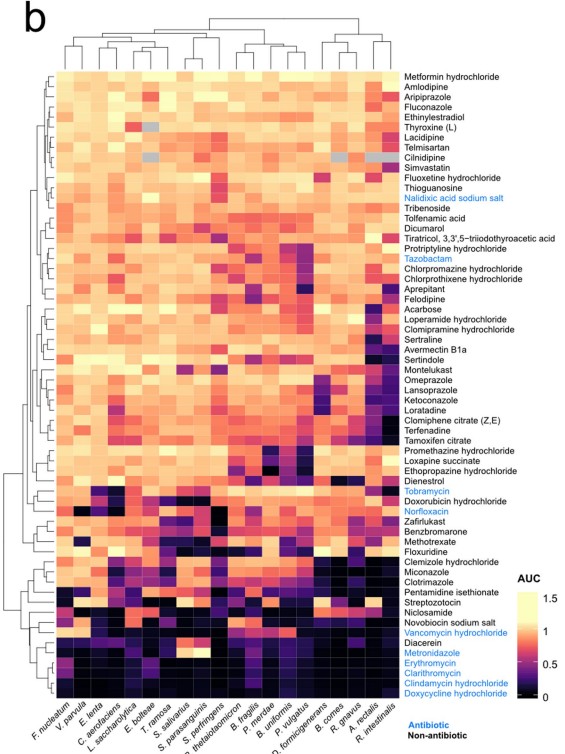

**Extended Data Fig. 1 | Drugs selected for the *S. Tm* invasion assay have a wide range of effects on individual members of Com20. a**) Schematic view of the selection of drugs included in the different in vitro and in vivo models. Each column corresponds to a given assay, from the Prestwick library assay to testing in humanized mice. The model, the bacteria tested, the total number of drugs and the inclusion or exclusion criteria of each assay are described on the corresponding column. Drugs included in each step are indicated by blue lines.

Other colors are explained in the figure. **b**) Heatmap of the growth of each member of Com20 after treatment with 62 drugs. Area under the curve (AUC) values normalized to untreated controls are shown. Columns and rows were clustered hierarchically with complete linkage. The plot includes the values of 62 drugs from the Prestwick library tested at 20 μM, replotted from[1]; the 3 "other drugs" in a) are not included.

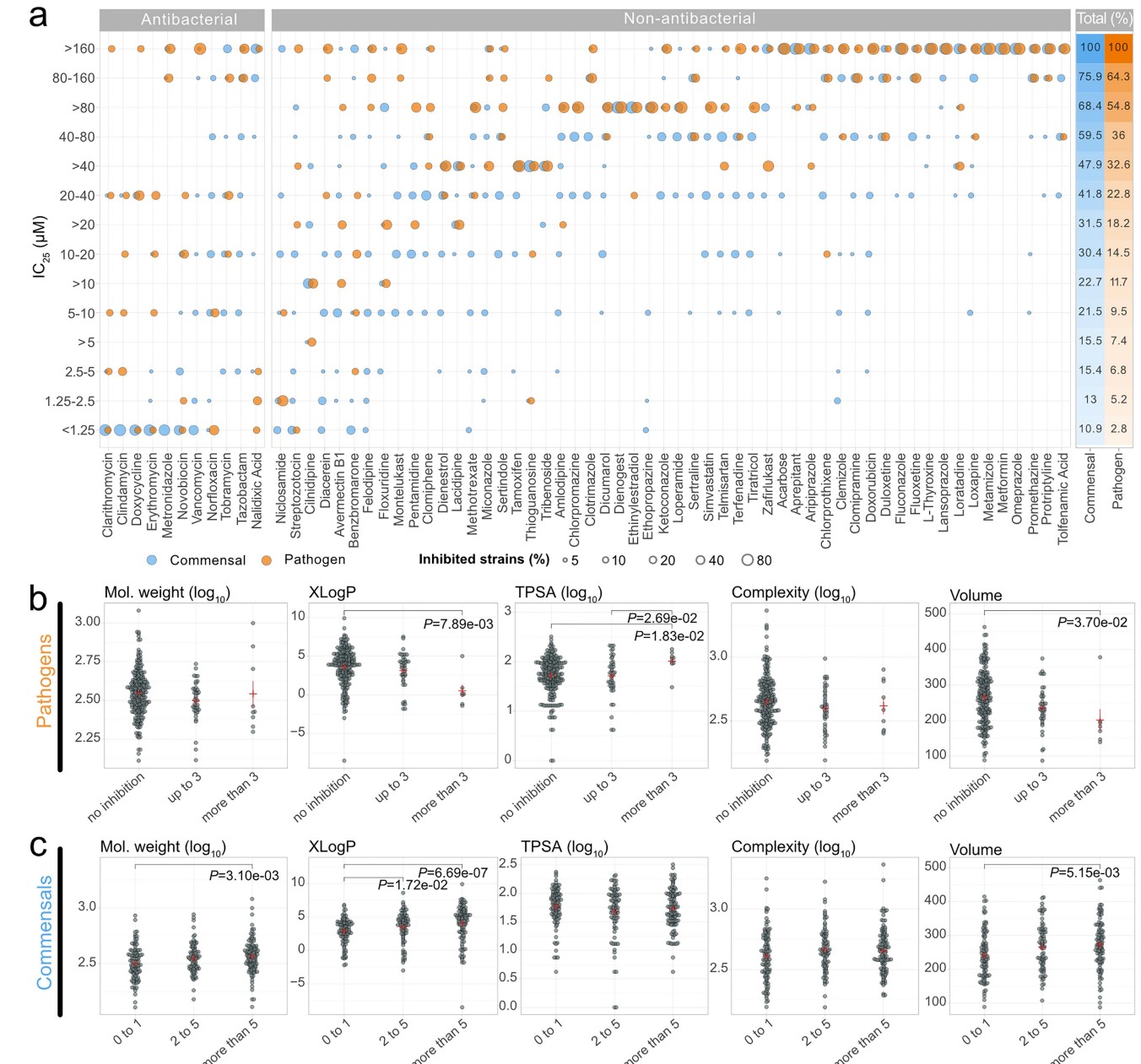

● Commensal  ● Pathogen     Inhibited strains (%)  ○5 ○10 ○20 ○40 ○80

**Extended Data Fig. 2 | Commensal gut bacteria are less resistant to non-antibiotic drugs than pathogenic Gammaproteobacteria species. a)** Inhibitory concentration of 25% growth inhibition (IC$_{25}$) values for a panel of 19 gut commensals and 5 pathogenic taxa (Supplementary Table 2). The size of the circles indicate the percentage of species inhibited at a given concentration range. For example, clarithromycin inhibits 20% of pathogens at concentrations between 2.5 and 5 µM. For concentrations labeled as greater than (e.g. >20 µM), the inhibition threshold was not reached at the indicated value, and values greater than this (e.g. 40 µM) were not evaluated. Side heatmap shows the cumulative proportion of gut commensals and pathogens inhibited at a given

concentration. **b)** and **c)** Chemical properties of compounds from the Prestwick library tested on pathogens (b) and commensals (c). Drugs were separated into 3 groups based on the number of bacteria they inhibited (Number of drugs in (b): No inhibition: 253, up to 3: 42, more than 3: 9. Number of drugs in (c): 0 to 1: 104, 2 to 5: 84, more than 5: 116). The properties assessed include molecular weight, hydrophobicity (XLogP), polar surface area (TPSA in Å$^2$), complexity and 3D volume (in Å$^3$). All properties were obtained from PubChem. Red horizontal and vertical lines represent mean ±1 s.e.m. Adjusted P values < 0.05 from Kruskal-Wallis test followed by a Dunn's test with Bonferroni correction are shown.

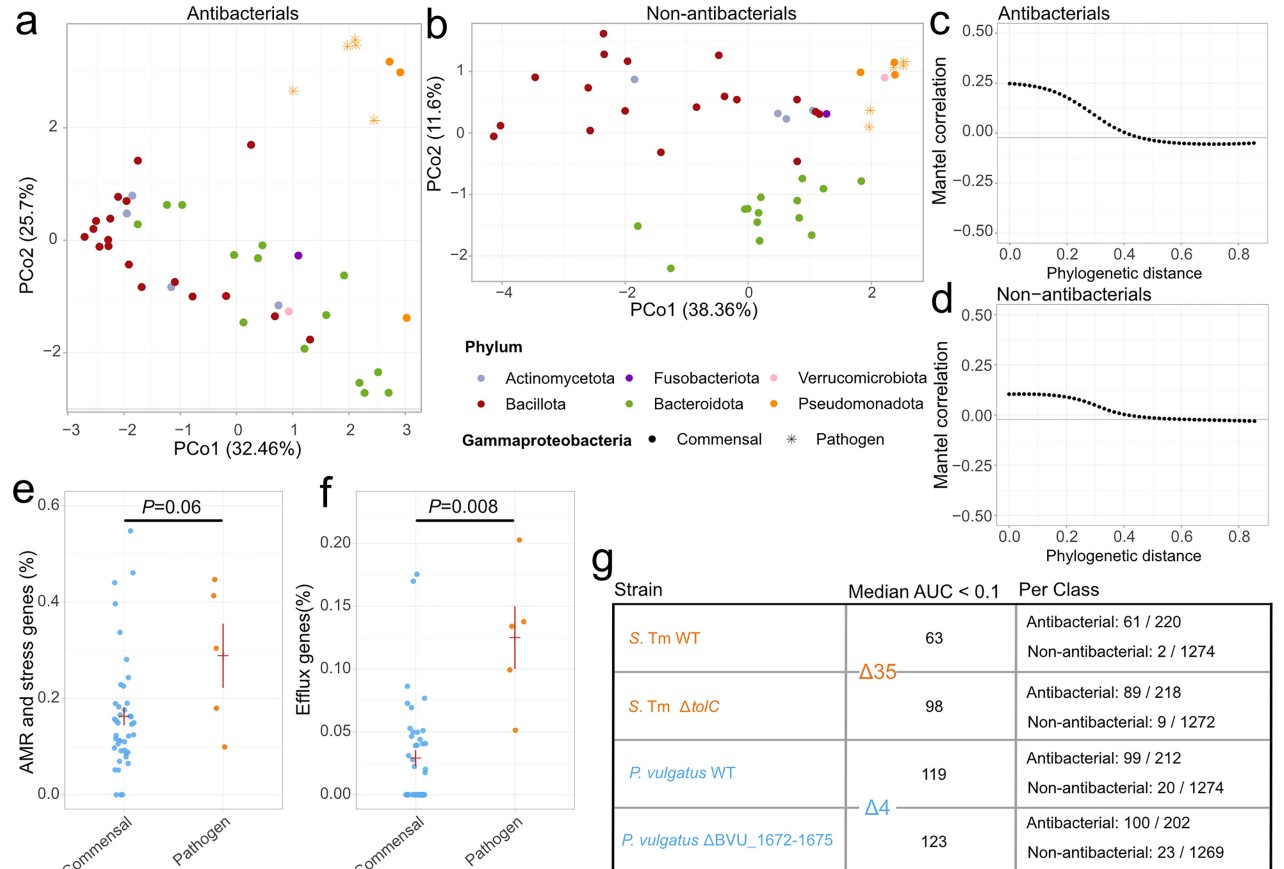

**Extended Data Fig. 3 | Differences in the repertoire of stress-related genes might help explain the response of commensals and Gammaproteobacteria pathogens to drugs. a**) and **b**) Principal coordinate analyses of Euclidean distances calculated using bacterial growth (AUC) after treatment with 102 antibiotics (a) and 117 non-antibiotics (b), colored by phylum (drug selection criteria: see Methods). **c**) and **d**) Change in the strength of the association of the bacterial response to 102 antibiotics (c) and 117 non-antibiotics (d) across evolutionary distances. The phylocorrelograms show the decrease in correlation of the response to the drugs between species with increased phylogenetic distance. Black horizontal line shows the expected correlation under the null hypothesis of no phylogenetic autocorrelation. **e**) and **f**) Percentage of genes encoded in the genome of 42 commensal bacteria and 5 Gammaproteobacteria pathogens involved in antimicrobial resistance and stress response (e) and efflux processes (f). Red horizontal and vertical lines represent mean ±1 s.e.m. One-tailed t-test. **g**) Number of drugs that inhibited growth (defined as area under the growth curve <0.1, normalized to DMSO controls) of *S*. Tm WT and *P. vulgatus* WT compared to their respective mutants lacking major RND-type efflux pumps (*S*. Tm Δ*tolC*, *P. vulgatus* ΔBVU_1672-1675). Colored numbers indicate the difference in active drugs between WT and mutant strains. The column on the right breaks down the total number of active drugs into antibiotic and non-antibiotic categories, and shows the number of drugs analyzed per category after a quality check.

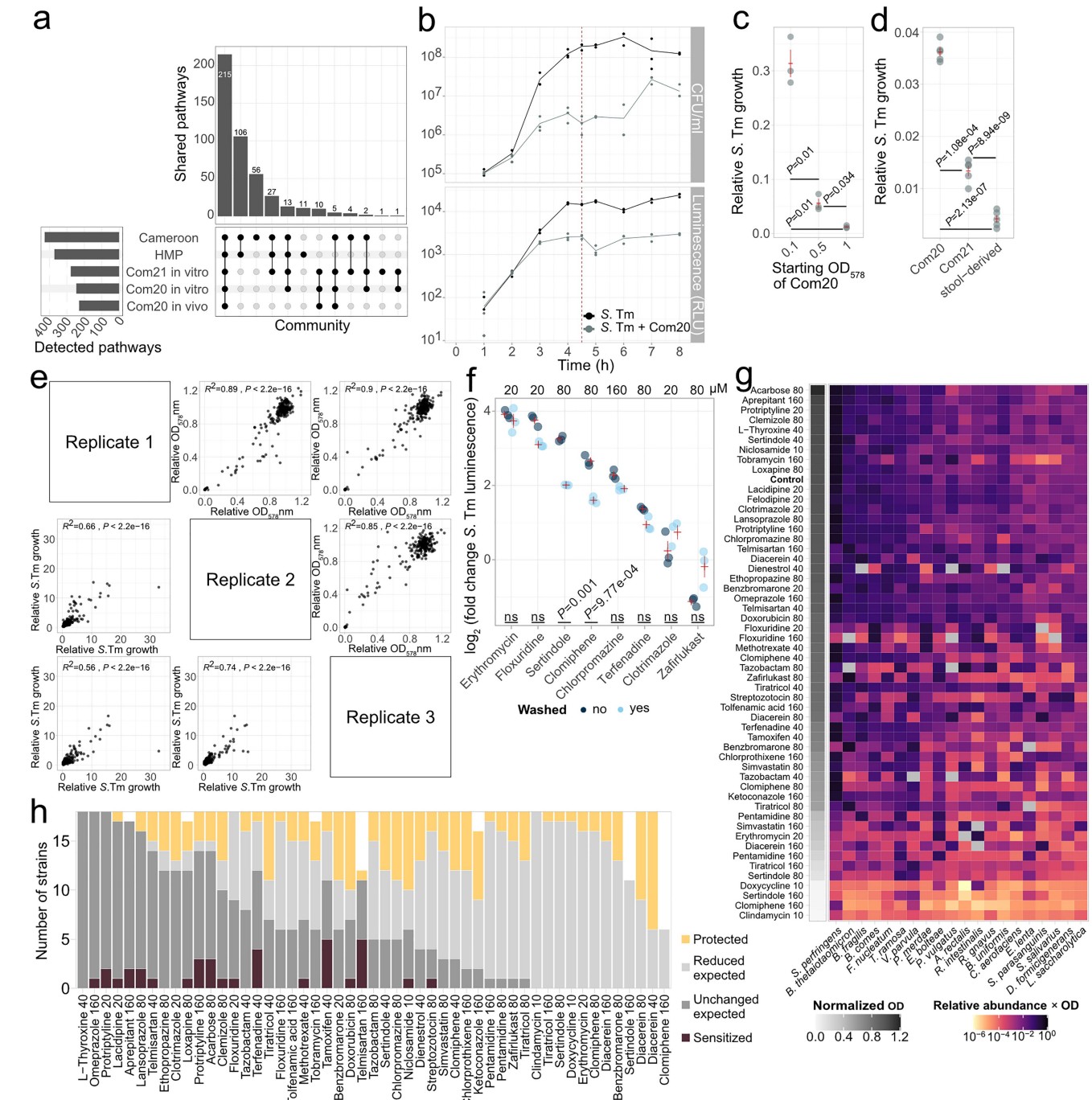

**Extended Data Fig. 4** | See next page for caption.

**Extended Data Fig. 4 | Com20 broadly recapitulates the functional potential of the human gut microbiome, reduces the growth of *S*. Tm and allows the emergence of complex ecological dynamics in vitro. a**) Upset plot showing the overlap in the number of MetaCyc metabolic pathways predicted to be present in Com20 (in vivo and in vitro), Com21 (in vitro), and human gut metagenomes from the Human Microbiome Project (HMP) and a Cameroonian cohort. **b**) Growth curves of *S*. Tm based on CFU counts on selective plating (top) or *S*. Tm-specific luminescence (bottom). Lines represent the mean of three biological replicates, the color indicates whether the pathogen was cultured alone or within Com20. At 4.5 h (red vertical line), *S*. Tm growth curves transitioned to stationary phase and luminescence could be used as a proxy of *S*. Tm levels. **c**) Relative growth of *S*. Tm in untreated Com20 compared to pure culture at different starting $OD_{578}$ of the community. *S*. Tm was quantified by luminescence. Red horizontal and vertical lines represent the mean of three biological replicates ± 1 s.e.m. *P* values with Benjamini-Hochberg correction from two-sided t-test are shown. **d**) Relative growth of *S*. Tm in untreated Com20, Com21 or a human stool-derived community at a starting $OD_{578}$ of 0.5, compared to pure culture. *S*. Tm was quantified by luminescence. Red horizontal and vertical lines represent the mean of three biological replicates ± 1 s.e.m. *P* values with Benjamini-Hochberg correction from two-sided t-test are shown. **e**) Association of community $OD_{578}$ and *S*. Tm luminescence relative to untreated Com20 across 53 treatments (253 drug-concentration pairings) in each of three biological replicates. $R^2$ and *P* values from linear regression models are shown. **f**) Growth of *S*. Tm in drug-treated Com20 with and without an intermediate washing step after treatment, but before inoculation of the pathogen. Each point represents a biological replicate. Drug concentrations indicated at the top. Red horizontal and vertical lines represent mean ± 1 s.e.m. of three biological replicates per treatment. *P* values from two-sided t-test are shown. ns: not significant; *P* value > 0.05. **g**) Biomass-scaled relative abundance of each member of drug-treated Com20 across 53 drug treatments plus a control, calculated by multiplying the normalized $OD_{578}$ of the community by the relative abundance of each taxon from 16S rRNA gene sequencing. The gray-scale column on the left shows the mean $OD_{578}$ of three biological replicates of the community normalized to an untreated control. **h**) Distribution of bacterial species displaying emergent (protection or sensitization in community) or expected responses to 51 drug treatments. An expected response in a community (gray) refers to a similar growth pattern in monocultures compared to the OD-scaled relative abundance (relative abundance × OD). Measures are normalized to the value of untreated Com20. Community protection (yellow) means that the species is affected by drug treatment in monoculture but remains unaltered in the treated Com20; conversely, community sensitization (burgundy) means that species growth is not affected in monoculture but its abundance decreases in the treated Com20. Missing values were due to quality control issues on bacterial growth.

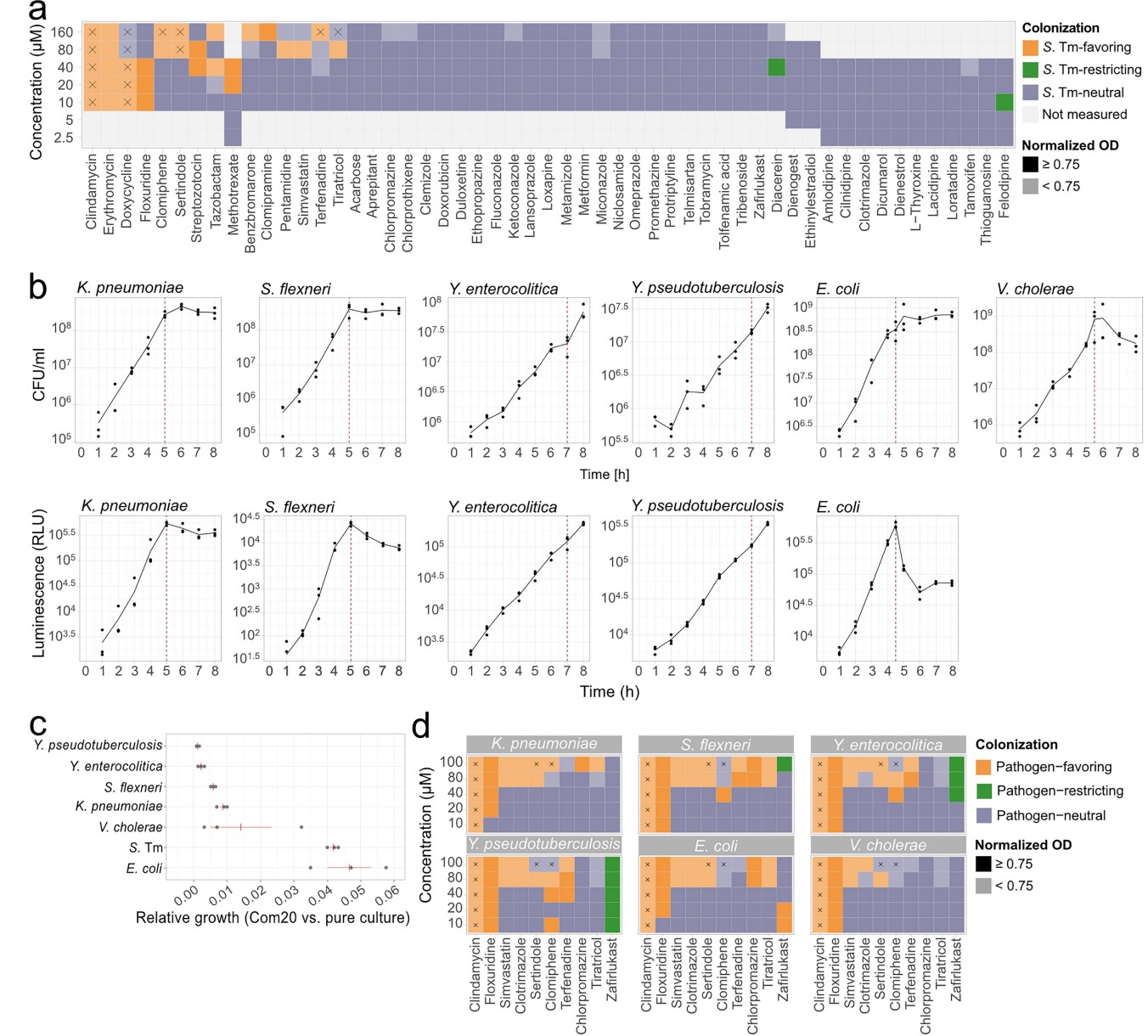

**Extended Data Fig. 5 | Drug treatment can hamper the ability of Com20 to resist the growth of *S*. Tm and other Gammaproteobacteria pathogens. a**) Classification of drug (columns) and concentration (rows) pairings according to the growth of *S*. Tm in drug-treated Com20. Samples marked with an X resulted in a community biomass <0.2 relative to an untreated community and were excluded from downstream analyses. Note that zafirlukast tends to limit *S*. Tm growth, but due to high variance, it is not classified as *S*. Tm-restricting. **b**) Growth curves for pathogenic members of Gammaproteobacteria based on plating (top) or pathogen-specific luminescence (bottom). The curve indicates the mean of three biological replicates. The time point for luminescence

measurements in the challenge assays was selected based on the plating results, indicated by vertical red lines. The growth of *V. cholerae* was not quantified by luminescence. **c**) Growth of Gammaproteobacteria species in untreated Com20 relative to pure culture. Pathogen loads from luminescence, except for *V. cholerae*, which was quantified by selective plating. Each point corresponds to a biological replicate. Red horizontal and vertical lines represent mean ±1 s.e.m. of three biological replicates per condition **d**) Classification of drug-concentration pairings according to the growth of Gammaproteobacteria pathogens in Com20, similar to panel (a). Samples marked with an X resulted in a community biomass <0.2 relative to untreated Com20.

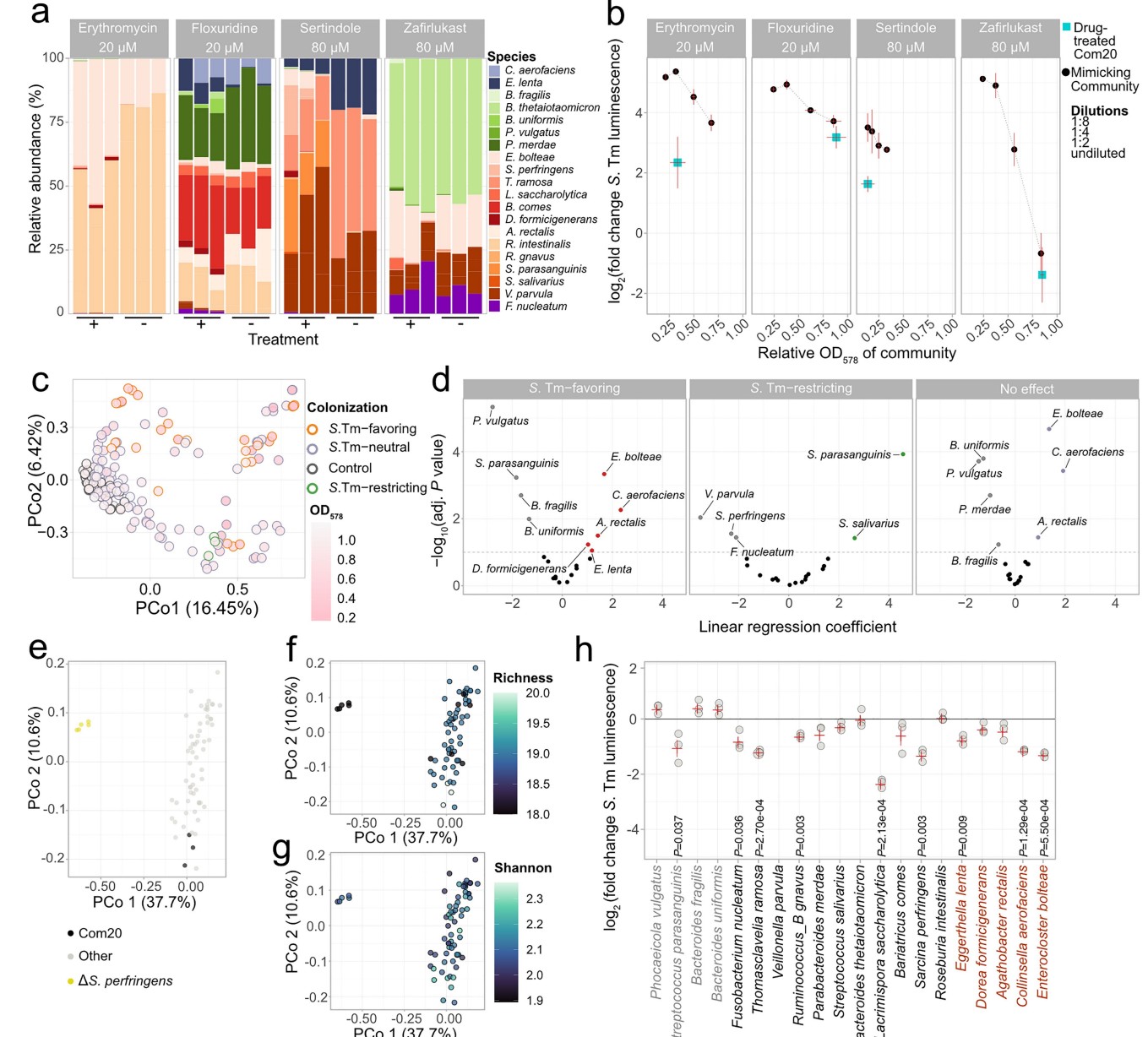

**Extended Data Fig. 6 | Community composition, biomass and the presence of competitors alter the growth of *S.* Tm after drug treatment. a**) Composition of Com20 after drug treatment with erythromycin, floxuridine, sertindole or zafirlukast (+) or after manually mixing members at the corresponding proportions to mimic the treatments (-). Abundances measured by 16S rRNA gene sequencing. Only species with a mean relative abundance ≥3% after 24 h of drug exposure were included in the mimic communities. Each bar represents one biological replicate. **b**) Growth of *S.* Tm in the treatment-mimicking communities from (a). The communities were diluted to emulate alterations in both the composition and the biomass of the communities after treatment. Black points represent the mean of three biological replicates, and red lines represent ±1 s.e.m. Blue points indicate the growth of *S.* Tm in drug-treated Com20. **c**) Principal coordinate analysis of drug-treated Com20 based on Bray-Curtis distances. Each point represents a drug-treated community. Circle border indicates colonization group and fill indicates the OD$_{578}$ of the community. **d**) Volcano plot showing effect size and adjusted *P* values of linear regression models of the abundance of members of Com20 in each treatment group compared to untreated controls. Dashed lines indicate the adjusted *P* value significance threshold of 0.1. **e-g**) Principal coordinate analysis of 19-member communities, colored by the presence of *S. perfringens* e), species richness f), and Shannon index g). **h**) *S.* Tm luminescence in pairwise co-cultures with each member of Com20 (pathogen to commensal ratio =1:500), compared to *S.* Tm in pure culture. Color of species names and order of the taxa in the y-axis is based on the differential abundance analysis between *S.* Tm-favoring conditions and controls shown in (d). Red horizontal and vertical lines represent the mean ±1 s.e.m. of three biological replicates. *P* values ≤ 0.5 are shown from two-sided t-tests.

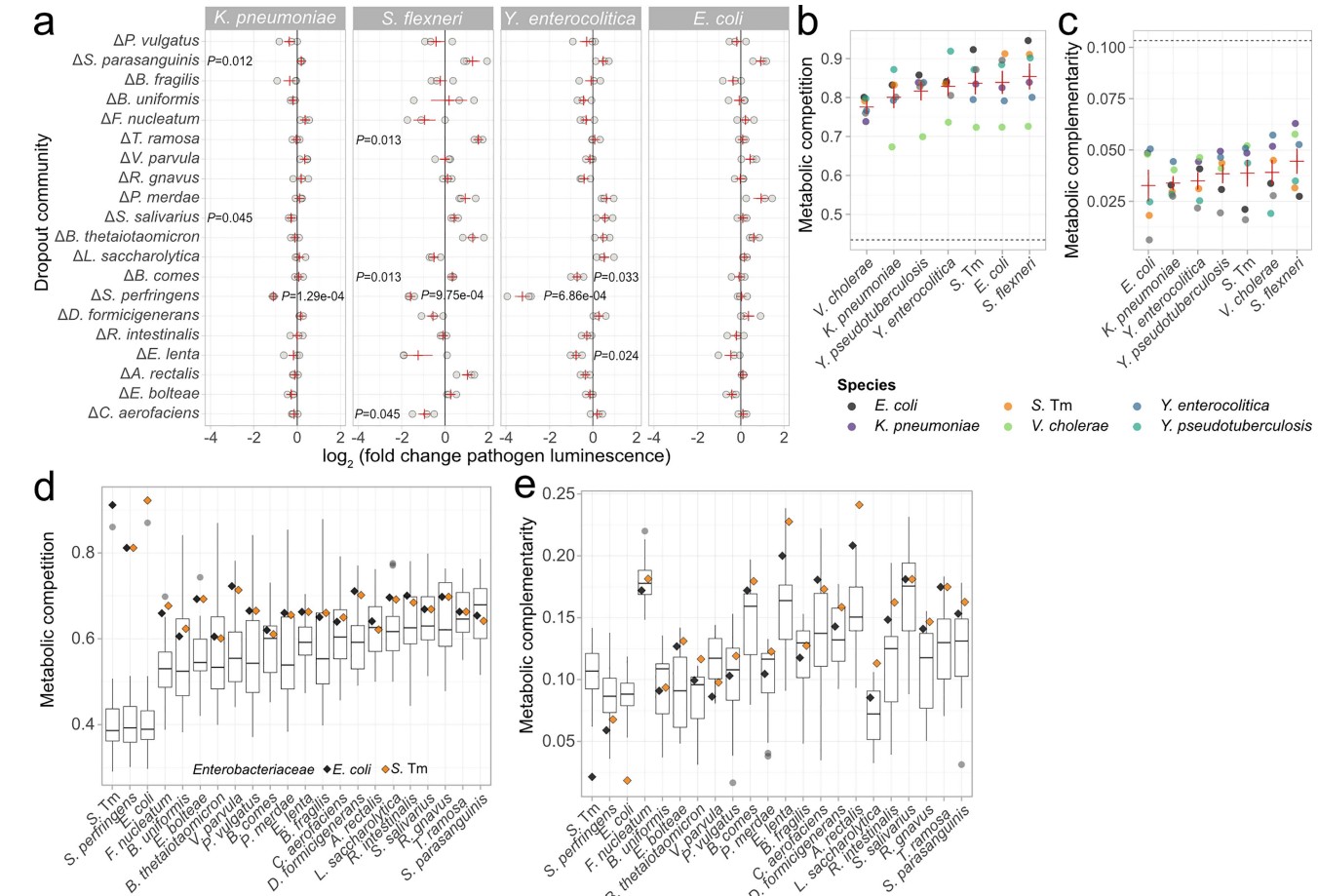

**Extended Data Fig. 7 | Community composition influences the growth of various Gammaproteobacteria pathogens. a**) Growth of *K. pneumoniae, S. flexneri, Y. enterocolitica* and *E. coli* in 19-member communities (each missing the species indicated in the y-axis) compared to untreated Com20. Red vertical and horizontal lines represent the mean ±1 s.e.m. across three biological replicates. *P* values from two-tailed t-tests; only values ≤ 0.05 are shown. **b,c**) Metabolic competition b) and complementarity c) indices of pathogenic Gammaproteobacteria species, calculated from genome-scale metabolic models (see Methods). Dashed horizontal lines indicate the mean levels of

*S*. Tm with members of Com21. Red points and bars represent mean ±1 s.e.m. **d,e**) Distribution of competition d) and complementarity c) index values of each member of Com21 against all other members of the community and against *S*. Tm are shown by the boxplots (n = 21 other species per boxplot). The exact values of the indices against *E. coli* and *S*. Tm are indicated by black and orange diamonds, respectively. Note that the indices are not symmetric. In b-e) boxplots show the median, IQR, whiskers to the min/max within 1.5 IQR, and outliers as individual points.

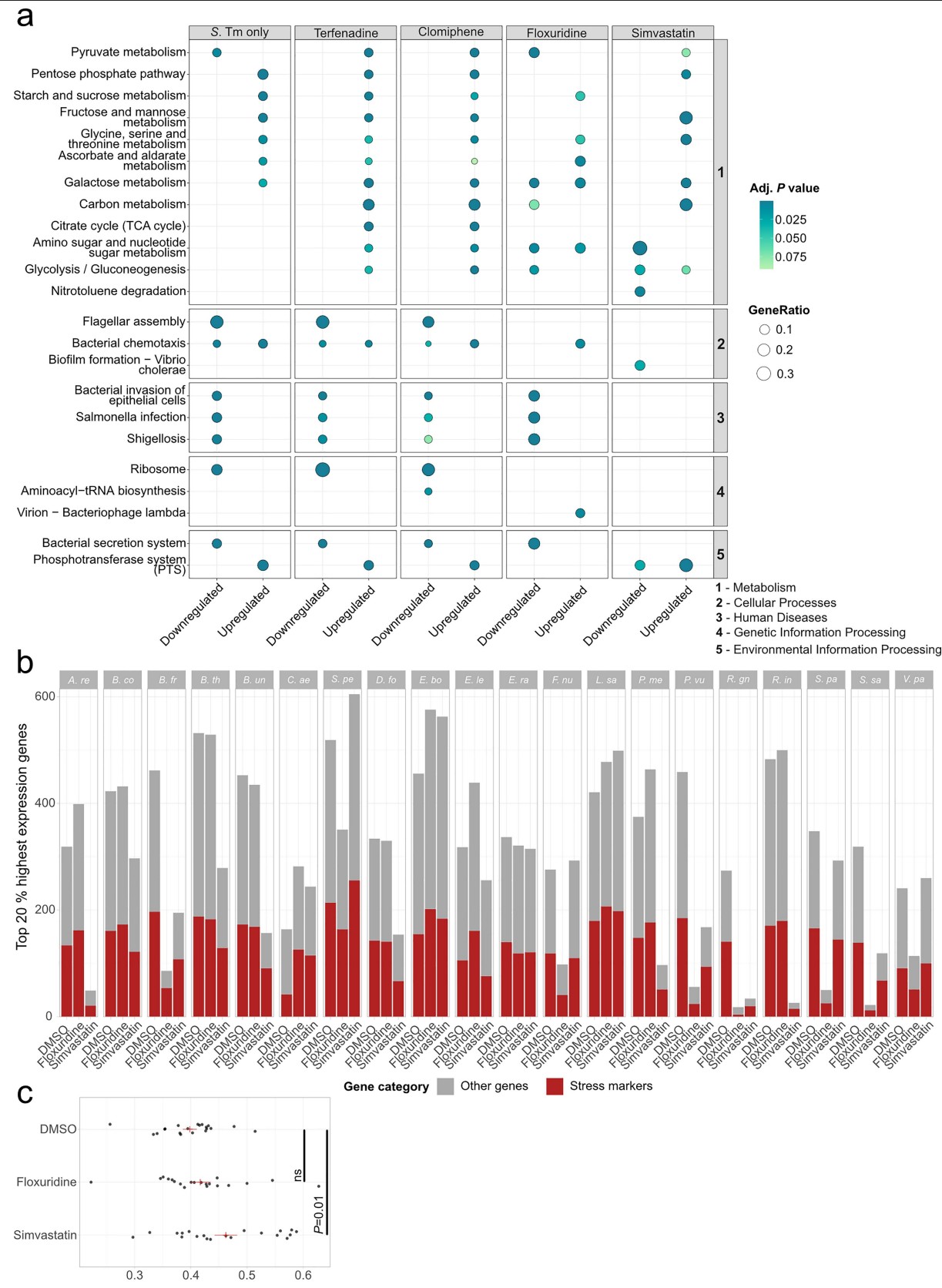

**Extended Data Fig. 8** | See next page for caption.

**Extended Data Fig. 8 | Alterations in the transcriptomic profiles of *S.* Tm and commensals vary between drug treatments. a**) KEGG pathways more represented than expected among genes that are up- and -downregulated in *S.* Tm after growth in drug-treated Com20 or pure culture compared to untreated Com20. Circle size represents the fraction of differentially expressed genes from DESeq2 analysis that map to a pathway, relative to the total number of differentially expressed genes. Values were calculated separately for upregulated and downregulated genes. Circle colors represent the adjusted *P* values from overrepresentation analysis. Pathways grouped by their KEGG category. **b**) Number of genes in the 20 % of highest expression on each member of Com20 upon treatment or in untreated Com20; red bar indicates the number of genes in the set of stress-related markers identified by Avican et al.[22]. **c**) Fraction of the top 20 % most expressed genes that are stress-related marker genes[22] in each of the 20 species of Com20 across controls and treatments; in other words, from b): red bar/(red bar + gray bar). Red horizontal and vertical lines represent mean ± 1 s.e.m. *P* values with Benjamini-Hochberg correction from one-tailed t-tests.

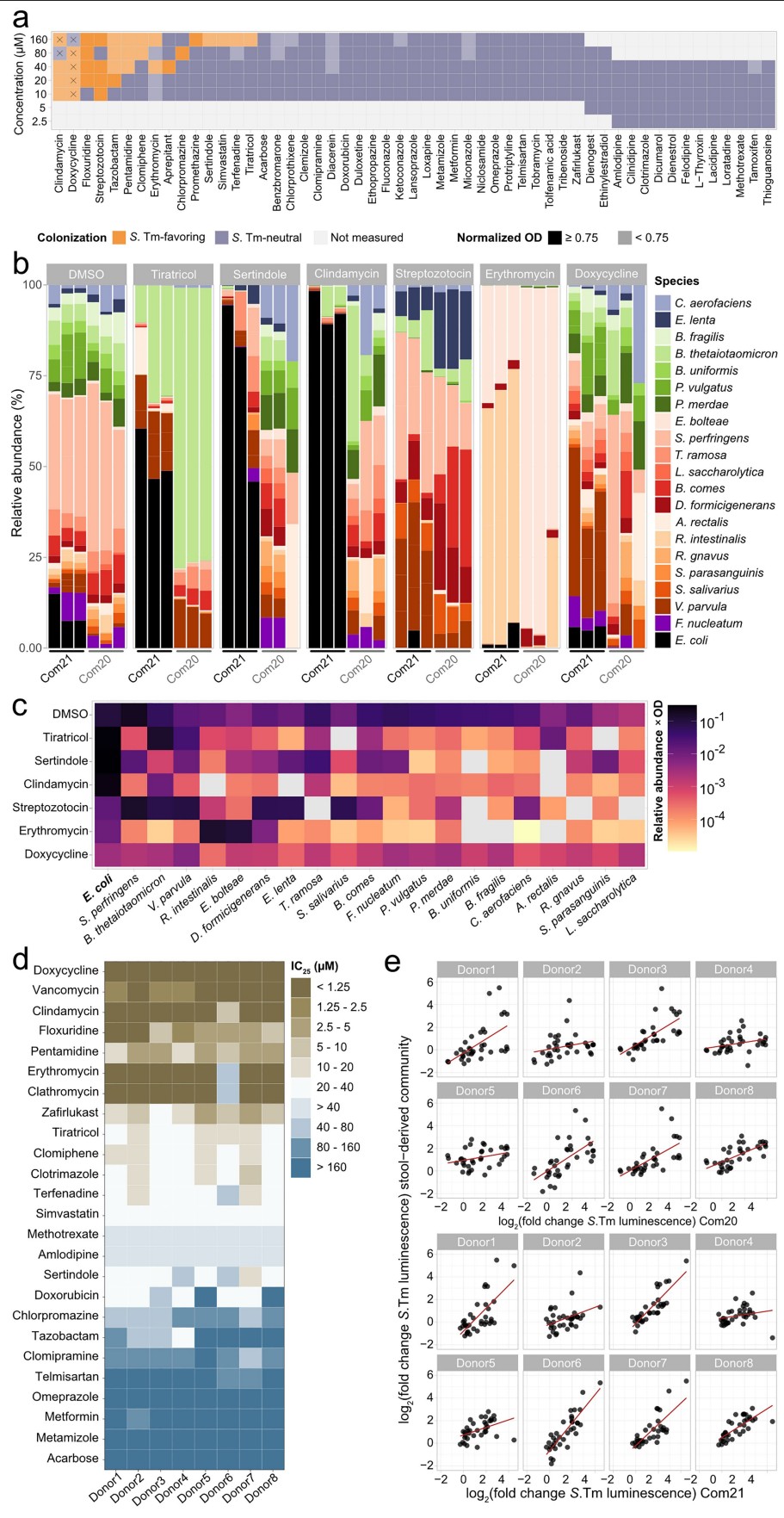

**Extended Data Fig. 9 | Pathogen growth in drug-altered Com20 is replicated in synthetic and stool-derived communities containing *E. coli*.**
**a**) Classification of drug (columns) and concentration (rows) pairings according to the growth of *S*. Tm in drug-treated Com21. Samples marked with an X resulted in a community biomass <0.2 relative to an untreated community. **b**) Relative abundance of members of Com20 and Com21 in untreated controls (DMSO) and after treatment with diverse drugs. Abundances determined via 16S rRNA gene sequencing. Each bar represents one biological replicate. **c**) Biomass-scaled relative abundance of each member of drug-treated Com21, calculated by multiplying the normalized $OD_{578}$ of the community by the relative abundance of each taxon obtained by 16S rRNA gene sequencing. Values indicate the mean of three biological replicates. **d**) $IC_{25}$ values of 25 drugs in stool-derived communities of 8 healthy human donors. Tiles show the mean of three biological replicates. **e**) Association between the growth of *S*. Tm in Com20 (top) and Com21 (bottom), and the growth of the pathogen in stool-derived microbial communities across 10 drugs at various concentrations. Linear regression lines shown in red.

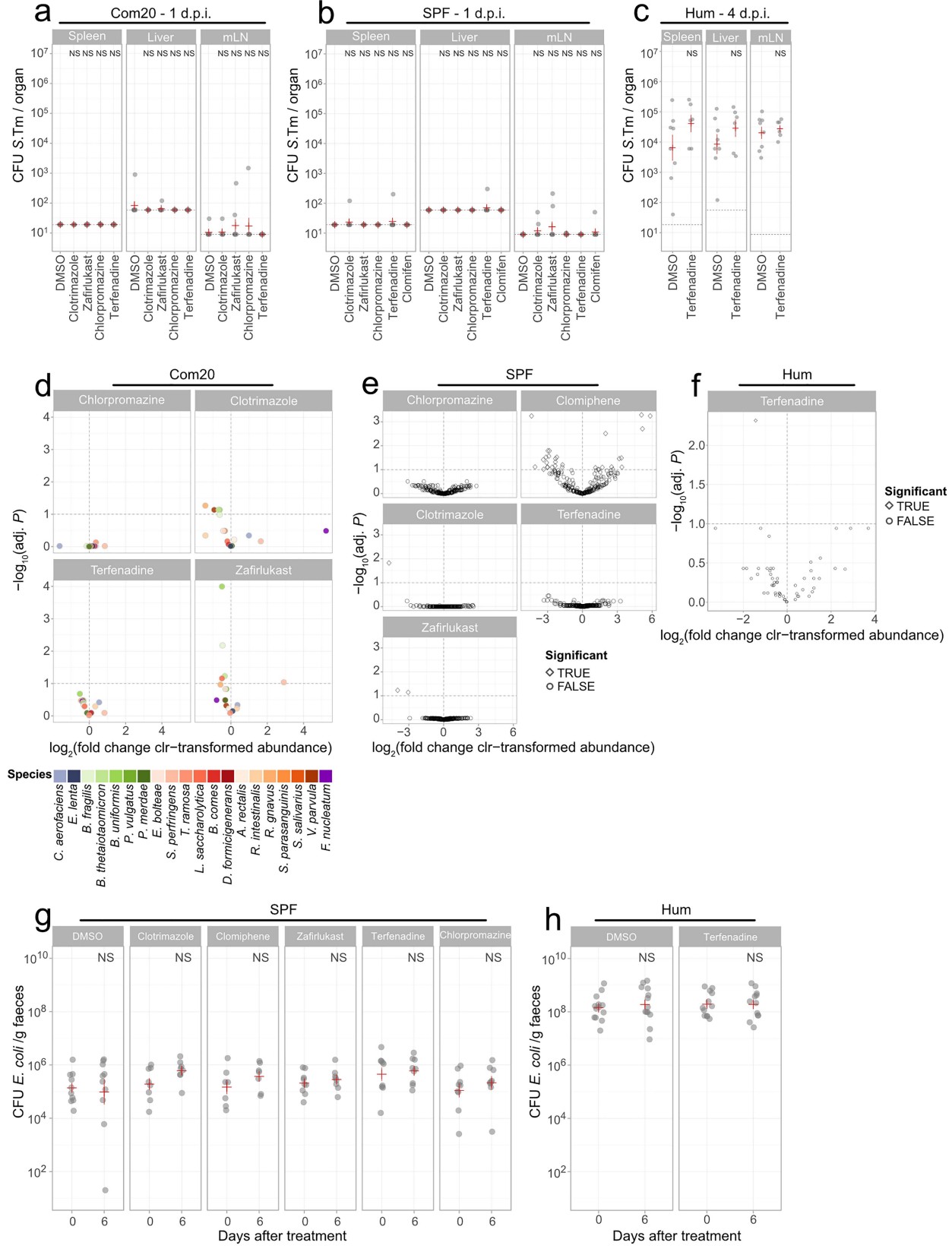

**Extended Data Fig. 10** | See next page for caption.

**Extended Data Fig. 10 | Six-day treatment of mice with the non-antibiotics chlorpromazine, clomiphene, clotrimazole, terfenadine, and zafirlukast results in mild effects on the microbiome and no change on *E. coli* levels. a-c**) *S.* Tm load in the spleen, liver, and mesenteric lymph nodes (mLN) of mice colonized with **a**) Com20, number of mice per treatment: DMSO: 8, clotrimazole: 7, zafirlukast: 8, chlorpromazine: 8, terfenadine: 7; **b**) SPF mice, DMSO: 9, clotrimazole: 9, zafirlukast: 9, chlorpromazine: 9, terfenadine: 9, clomiphene: 9, and **c**) humanized (Hum) mice, DMSO: 8, terfenadine: 6. Days post infection (d.p.i.): 1 day a,b) or 4 days c). Black horizontal lines indicate detection limits. Red horizontal lines represent the mean the above stated biological replicates, with vertical bars showing ± 1 s.e.m. *P* values from one-tailed Wilcoxon test with Benjamini-Hochberg correction. **d-f**) Effect size and adjusted *P* values from linear models of the abundance of Com20 members d),

native microbiota ASVs e), and human-derived-microbiota ASVs f) in mice 6 days after drug treatment compared to untreated controls. Models were adjusted for species abundance on day 0. The dashed horizontal line marks the significance threshold of 0.1. Point shape indicates whether an ASV was significantly different (TRUE) from controls. **g-h**) *E. coli* levels in the feces of g) SPF mice, number of mice per treatment: DMSO: 10, clotrimazole: 8, zafirlukast: 9, chlorpromazine: 9, terfenadine: 9, clomiphene: 7; or h) humanized mice, DMSO: 13, terfenadine: 12, measured by selective plating on MacConkey agar, before and 6 days after drug treatment. Red horizontal bars show the mean of the above number of biological replicates, with vertical bars representing ± 1 s.e.m. Adjusted *P* values from a two-tailed Wilcoxon test > 0.1 in all cases. NS: not significant; *P* > 0.05.

# Reporting Summary

## Statistics

For all statistical analyses, confirm that the following items are present in the figure legend, table legend, main text, or Methods section.

| n/a | Confirmed | |
|---|---|---|
| ☐ | ☒ | The exact sample size (*n*) for each experimental group/condition, given as a discrete number and unit of measurement |
| ☐ | ☒ | A statement on whether measurements were taken from distinct samples or whether the same sample was measured repeatedly |
| ☐ | ☒ | The statistical test(s) used AND whether they are one- or two-sided<br>*Only common tests should be described solely by name; describe more complex techniques in the Methods section.* |
| ☐ | ☒ | A description of all covariates tested |
| ☐ | ☒ | A description of any assumptions or corrections, such as tests of normality and adjustment for multiple comparisons |
| ☐ | ☒ | A full description of the statistical parameters including central tendency (e.g. means) or other basic estimates (e.g. regression coefficient) AND variation (e.g. standard deviation) or associated estimates of uncertainty (e.g. confidence intervals) |
| ☐ | ☒ | For null hypothesis testing, the test statistic (e.g. *F*, *t*, *r*) with confidence intervals, effect sizes, degrees of freedom and *P* value noted<br>*Give P values as exact values whenever suitable.* |
| ☒ | ☐ | For Bayesian analysis, information on the choice of priors and Markov chain Monte Carlo settings |
| ☒ | ☐ | For hierarchical and complex designs, identification of the appropriate level for tests and full reporting of outcomes |
| ☐ | ☒ | Estimates of effect sizes (e.g. Cohen's *d*, Pearson's *r*), indicating how they were calculated |

*Our web collection on statistics for biologists contains articles on many of the points above.*

## Software and code

Policy information about availability of computer code

| | |
|---|---|
| Data collection | Gen5 (v. 3.05 or higher, Agilent), Infinite F200 PRO i-control software (Tecan) |
| Data analysis | No custom algorithms used for data analysis; the code used is available at https://github.com/Lisa-Maier-Lab/HTD_CR.<br>R packages (R v. 4.2.0, ape v. 5.8, Ade4 v. 1.7-22, phylosignal v. 1.3.1, DECIPHER v. 2.24.0, PERFect v. 1.14.0, vegan v. 2.6-8, MaAsLin2 v. 1.13.0, DESeq2 v. 1.44.0, clusterProfiler v. 4.12.6, sandwich v. 3.0-2, lmtest v. 0.9-40, lme4 v. 1.1-35.5, emmeans v. 1.10.6, ggplot2 v. 3.5.1,Rstatix v. 0.7.2, BioVoxxel Toolbox v. 2.6.0)<br>Bioinformatics software (PICRUSt2 v. 2.4.1., PhyloMint v. 0.1.0, Bowtie v. 2.5.2, Kraken2 v. 2.1.3, Bracken v. 2.9, CarveMe v. 1.5.1, nf-core metadenovo v. 1.0.1, nf-core taxprofiler v. 1.2, phylophlan v. 3.0, FragGeneScan v. 1.31, AMRFinderPlus v. 3.11, argNorm v. 0.2, DADA2 v. 1.21.0, BBmap v. 39.01, Subread v. 2.0.1, EggNOG mapper v. 2.1.9, KOfamscan v. 1.3.0, QuPath v. 0.5.1, fastp v. 0.23.4, Fiji v. 1.10.6)<br>R packages (R v. 4.2.0, ape v. 5.8, Ade4 v. 1.7-22, phylosignal v. 1.3.1, DECIPHER v. 2.24.0, PERFect v. 1.14.0, vegan v. 2.6-8, MaAsLin2 v. 1.13.0, DESeq2 v. 1.44.0, clusterProfiler v. 4.12.6, sandwich v. 3.0-2, lmtest v. 0.9-40, lme4 v. 1.1-35.5, emmeans v. 1.10.6, ggplot2 v. 3.5.1,Rstatix v. 0.7.2, BioVoxxel Toolbox v. 2.6.0)<br>Bioinformatics software (PICRUSt2 v. 2.4.1., PhyloMint v. 0.1.0, Bowtie v. 2.5.2, Kraken2 v. 2.1.3, Bracken v. 2.9, CarveMe v. 1.5.1, nf-core metadenovo v. 1.0.1, nf-core taxprofiler v. 1.2, phylophlan v. 3.0, FragGeneScan v. 1.31, AMRFinderPlus v. 3.11, argNorm v. 0.2, DADA2 v. 1.21.0, BBmap v. 39.01, Subread v. 2.0.1, EggNOG mapper v. 2.1.9, KOfamscan v. 1.3.0, QuPath v. 0.5.1, fastp v. 0.23.4, Fiji v. 1.10.6) |

For manuscripts utilizing custom algorithms or software that are central to the research but not yet described in published literature, software must be made available to editors and reviewers. We strongly encourage code deposition in a community repository (e.g. GitHub). See the Nature Portfolio guidelines for submitting code & software for further information.

# Data

Policy information about availability of data

 All manuscripts must include a data availability statement. This statement should provide the following information, where applicable:

- Accession codes, unique identifiers, or web links for publicly available datasets
- A description of any restrictions on data availability
- For clinical datasets or third party data, please ensure that the statement adheres to our policy

Raw sequencing reads from the 16S rRNA and transcriptome analyses have been deposited in the European Nucleotide Archive (accession ID: ID PRJEB65315). All other data are provided in the Supplementary Information or the Source Data files.
Databases used in this study:
Genome Taxonomy Database (GTDB) release 202 (https://data.gtdb.ecogenomic.org/releases/release202/)
AMRFinderPlus database v. 2023-09-26.1
DADA2-formatted ASV database based on GTDB release R06-RS20259 ( https://scilifelab.figshare.com/articles/dataset/
SBDI_Sativa_curated_16S_GTDB_database/14869077).
Kraken2 and Bracken GTDB-formated database based on the Unified Human Gut Genome catalog (http://ftp.ebi.ac.uk/pub/databases/metagenomics/
mgnify_genomes/human-gut/v2.0.2/).

# Research involving human participants, their data, or biological material

Policy information about studies with human participants or human data. See also policy information about sex, gender (identity/presentation), and sexual orientation and race, ethnicity and racism.

| | |
|---|---|
| Reporting on sex and gender | The study was open to all genders and age groups. |
| Reporting on race, ethnicity, or other socially relevant groupings | The study was open to all these groups. |
| Population characteristics | The study was only open to participants that have, to the best of their knowledge, a healthy gut microbiota. The participants should not have taken any antibiotics for 12 months prior to sample donation, and should not have been diagnosed with any disease related to the intestinal microbiota. These include inflammatory bowel disease, colon cancer, lactose intolerance, diabetes as well as recently experienced diarrhoea. The sample processing procedure required participants to provide their fecal samples within minutes of collection; therefore, all samples were collected on-site in our laboratories. All participants were international lab members aged between 20 and 40 years. |
| Recruitment | Lab members of our laboratories |
| Ethics oversight | Ethics Committee of the University Hospital Tübingen, project ID 314/2022B02 |

Note that full information on the approval of the study protocol must also be provided in the manuscript.

# Field-specific reporting

Please select the one below that is the best fit for your research. If you are not sure, read the appropriate sections before making your selection.

☒ Life sciences  ☐ Behavioural & social sciences  ☐ Ecological, evolutionary & environmental sciences

For a reference copy of the document with all sections, see nature.com/documents/nr-reporting-summary-flat.pdf

# Life sciences study design

All studies must disclose on these points even when the disclosure is negative.

| | |
|---|---|
| Sample size | For animal experiments, sample sizes were determined using power analysis with G*Power. For all non-animal experiments, no formal sample size calculation was performed. We used three independent biological replicates per condition, which is standard practice in our microbiological assays. Based on our experience, this number provides sufficient statistical power to detect reproducible effects, supports basic statistical analyses, and allows for the identification of potential outliers, while maintaining a balance between resource use and feasibility. |
| Data exclusions | Replicates with inconsistent growth behavior were excluded from our analysis (for details please see "Methods") |
| Replication | We have three biological replicates for each strain in the Prestwick library screen and two to three biological replicates for the IC25 testing. For the in vitro invasion assays, we have 3 -5  biological replicates and show correlations between replicates in ED Fig. 4e . Transcriptomic analyses were performed in three biological replicates. All other experiments were performed in at least three independent replicates. For the data reported, all attempts for replication were successful. |

| Randomization | No randomization was applied in the microbiological experiments. For the animal experiments, male and female mice were housed in separate cages. These cages were then randomly assigned to the experimental groups, ensuring that each group included mice of both sexes. In experiments involving human stool-derived microbial communities, all donor samples were included in every experimental group. |
|---|---|
| Blinding | No blinding was applied in the in vitro experiments. In the in vivo experiments, blinding was not performed because drug-specific side effects needed to be evaluated. However, pathoscoring was conducted in a blinded manner by two independent assessors. |

# Behavioural & social sciences study design

All studies must disclose on these points even when the disclosure is negative.

| Study description | Briefly describe the study type including whether data are quantitative, qualitative, or mixed-methods (e.g. qualitative cross-sectional, quantitative experimental, mixed-methods case study). |
|---|---|
| Research sample | State the research sample (e.g. Harvard university undergraduates, villagers in rural India) and provide relevant demographic information (e.g. age, sex) and indicate whether the sample is representative. Provide a rationale for the study sample chosen. For studies involving existing datasets, please describe the dataset and source. |
| Sampling strategy | Describe the sampling procedure (e.g. random, snowball, stratified, convenience). Describe the statistical methods that were used to predetermine sample size OR if no sample-size calculation was performed, describe how sample sizes were chosen and provide a rationale for why these sample sizes are sufficient. For qualitative data, please indicate whether data saturation was considered, and what criteria were used to decide that no further sampling was needed. |
| Data collection | Provide details about the data collection procedure, including the instruments or devices used to record the data (e.g. pen and paper, computer, eye tracker, video or audio equipment) whether anyone was present besides the participant(s) and the researcher, and whether the researcher was blind to experimental condition and/or the study hypothesis during data collection. |
| Timing | Indicate the start and stop dates of data collection. If there is a gap between collection periods, state the dates for each sample cohort. |
| Data exclusions | If no data were excluded from the analyses, state so OR if data were excluded, provide the exact number of exclusions and the rationale behind them, indicating whether exclusion criteria were pre-established. |
| Non-participation | State how many participants dropped out/declined participation and the reason(s) given OR provide response rate OR state that no participants dropped out/declined participation. |
| Randomization | If participants were not allocated into experimental groups, state so OR describe how participants were allocated to groups, and if allocation was not random, describe how covariates were controlled. |

# Ecological, evolutionary & environmental sciences study design

All studies must disclose on these points even when the disclosure is negative.

| Study description | Briefly describe the study. For quantitative data include treatment factors and interactions, design structure (e.g. factorial, nested, hierarchical), nature and number of experimental units and replicates. |
|---|---|
| Research sample | Describe the research sample (e.g. a group of tagged Passer domesticus, all Stenocereus thurberi within Organ Pipe Cactus National Monument), and provide a rationale for the sample choice. When relevant, describe the organism taxa, source, sex, age range and any manipulations. State what population the sample is meant to represent when applicable. For studies involving existing datasets, describe the data and its source. |
| Sampling strategy | Note the sampling procedure. Describe the statistical methods that were used to predetermine sample size OR if no sample-size calculation was performed, describe how sample sizes were chosen and provide a rationale for why these sample sizes are sufficient. |
| Data collection | Describe the data collection procedure, including who recorded the data and how. |
| Timing and spatial scale | Indicate the start and stop dates of data collection, noting the frequency and periodicity of sampling and providing a rationale for these choices. If there is a gap between collection periods, state the dates for each sample cohort. Specify the spatial scale from which the data are taken |
| Data exclusions | If no data were excluded from the analyses, state so OR if data were excluded, describe the exclusions and the rationale behind them, indicating whether exclusion criteria were pre-established. |
| Reproducibility | Describe the measures taken to verify the reproducibility of experimental findings. For each experiment, note whether any attempts to repeat the experiment failed OR state that all attempts to repeat the experiment were successful. |
| Randomization | Describe how samples/organisms/participants were allocated into groups. If allocation was not random, describe how covariates were controlled. If this is not relevant to your study, explain why. |

| Blinding | *Describe the extent of blinding used during data acquisition and analysis. If blinding was not possible, describe why OR explain why blinding was not relevant to your study.* |
|---|---|

Did the study involve field work?  ☐ Yes  ☐ No

## Field work, collection and transport

| Field conditions | *Describe the study conditions for field work, providing relevant parameters (e.g. temperature, rainfall).* |
|---|---|
| Location | *State the location of the sampling or experiment, providing relevant parameters (e.g. latitude and longitude, elevation, water depth).* |
| Access & import/export | *Describe the efforts you have made to access habitats and to collect and import/export your samples in a responsible manner and in compliance with local, national and international laws, noting any permits that were obtained (give the name of the issuing authority, the date of issue, and any identifying information).* |
| Disturbance | *Describe any disturbance caused by the study and how it was minimized.* |

# Reporting for specific materials, systems and methods

We require information from authors about some types of materials, experimental systems and methods used in many studies. Here, indicate whether each material, system or method listed is relevant to your study. If you are not sure if a list item applies to your research, read the appropriate section before selecting a response.

### Materials & experimental systems

| n/a | Involved in the study |
|---|---|
| ☐ | ☒ Antibodies |
| ☒ | ☐ Eukaryotic cell lines |
| ☒ | ☐ Palaeontology and archaeology |
| ☐ | ☒ Animals and other organisms |
| ☒ | ☐ Clinical data |
| ☒ | ☐ Dual use research of concern |
| ☒ | ☐ Plants |

### Methods

| n/a | Involved in the study |
|---|---|
| ☒ | ☐ ChIP-seq |
| ☒ | ☐ Flow cytometry |
| ☒ | ☐ MRI-based neuroimaging |

## Antibodies

| Antibodies used | anti-CD31 (rat, 1:40; Dako Ref. M0823), anti-RelA (rabbit, 1:400; Novus Biological Ref. NB100-2176), Hif1alpha (rabbit, 1:500; Novus Biological Ref. NB100479), CD4 (rat, 1:1000; Thermo fisher Ref. 14-9766-82), CD8 (rabbit, 1:400; Cell Signaling Ref. 98941S), CD11b (rabbit, 1:10000; Abcam Ref. ab133357), CD11c (rabbit, 1:300; Cell Signaling 97585), F4/80 (rabbit, 1:400, Cell Signaling Ref. 70076), Cl. Casp. 3 (rabbit, 1:300; Cell Signaling Ref. 9661), KI67 (rabbit, 1:100; Thermo Fisher Ref. RM-9106-S1), B220 (rat, 1:3000; BD Ref. 553084), Rabbit Anti-Rat IgG H&L (preadsorbed, 1:1000, abcam, cat. No. ab102248); Polymer Anti-rabbit Poly-HRP-IgG (Leica, cat. No. DS9800). |
|---|---|
| Validation | All primary and secondary antibodies used in this study were commercially available and well-characterized. They were selected based on validated specificity for the target antigen and confirmed suitability for the species and application, as documented by the manufacturers. No custom or unvalidated antibodies were used. Full details of all primary and secondary antibodies, including links to the manufacturers' information, are provided in the Methods section and in Supplementary Table 13.<br>Primary antibodies:<br>CD31, rat, 1:40, Dako, cat. No. M0823; https://www.agilent.com/store/productDetail.jsp?catalogId=M082329-2<br>Rel A, rabbit, 1:400, Novus Biologicals, cat. No. NB100-2176; https://www.novusbio.com/products/rela-nfkb-p65-antibody_nb100-2176<br>Hif1 alpha, rabbit; 1:500, Novus Biologicals, cat. No. NB100479; https://www.novusbio.com/search?keywords=NB100479<br>CD4, rat, 1:1000, Thermo fisher, cat. No. 14-9766-82; https://www.thermofisher.com/antibody/product/CD4-Antibody-clone-4SM95-Monoclonal/14-9766-82<br>CD8, rabbit, 1:400, Cell Signaling, cat. No. 98941S; https://www.cellsignal.de/products/primary-antibodies/cd8a-d4w2z-xp-rabbit-mab-mouse-specific/98941<br>CD11b, rabbit, 1:10000, abcam, cat. No. ab133357; https://www.abcam.com/cd11b-antibody-epr1344-ab133357.html<br>CD11c, rabbit, 1:300, Cell Signaling, cat. No. 97585; https://www.cellsignal.com/products/primary-antibodies/cd11c-d1v9y-rabbit-mab/97585?_=1623060071874&Ntt=97585&tahead=true<br>F4/80, rabbit, 1:400, Cell Signaling, cat. No. 70076; https://www.cellsignal.com/products/primary-antibodies/f4-80-d2s9r-xp-rabbit-mab/70076<br>Cl. Casp.3, rabbit, 1:300, Cell Signaling, cat. No. 9661; https://www.cellsignal.com/products/primary-antibodies/cleaved-caspase-3-asp175-antibody/9661?_=1594014347096&Ntt=9661&tahead=true<br>KI67, rabbit, 1:100, Thermo Fisher, cat. No. RM-9106-S1; https://www.fishersci.de/shop/products/ki-67-rabbit-monoclonal-antibody/12603707 |

B220, rat; 1:3000, BD, cat. No. 553084; https://www.bdbiosciences.com/en-us/products/reagents/flow-cytometry-reagents/research-reagents/single-color-antibodies-ruo/purified-rat-anti-mouse-cd45r-b220.553084
Secondary antibodies:
Rabbit Anti-Rat IgG H&L preadsorbed, 1:1000, abcam cat. No. ab102248; https://www.abcam.com/en-us/products/secondary-antibodies/rabbit-rat-igg-h-l-preadsorbed-ab102248
Polymer Anti-rabbit Poly-HRP-IgG, Leica, cat. No. DS9800, https://shop.leicabiosystems.com/de/actions/ViewProductAttachment-OpenFile?
LocaleId=en_US&DirectoryPath=IFUs&FileName=ds9800.pdf&UnitName=LBS&srsltid=AfmBOopSn8V165zn5Eqkvt4WsmLm-KdlQpiq4SDhj3QGLRGb9ofvl6QH

# Eukaryotic cell lines

Policy information about cell lines and Sex and Gender in Research

| | |
|---|---|
| Cell line source(s) | *State the source of each cell line used and the sex of all primary cell lines and cells derived from human participants or vertebrate models.* |
| Authentication | *Describe the authentication procedures for each cell line used OR declare that none of the cell lines used were authenticated.* |
| Mycoplasma contamination | *Confirm that all cell lines tested negative for mycoplasma contamination OR describe the results of the testing for mycoplasma contamination OR declare that the cell lines were not tested for mycoplasma contamination.* |
| Commonly misidentified lines (See ICLAC register) | *Name any commonly misidentified cell lines used in the study and provide a rationale for their use.* |

# Palaeontology and Archaeology

| | |
|---|---|
| Specimen provenance | *Provide provenance information for specimens and describe permits that were obtained for the work (including the name of the issuing authority, the date of issue, and any identifying information). Permits should encompass collection and, where applicable, export.* |
| Specimen deposition | *Indicate where the specimens have been deposited to permit free access by other researchers.* |
| Dating methods | *If new dates are provided, describe how they were obtained (e.g. collection, storage, sample pretreatment and measurement), where they were obtained (i.e. lab name), the calibration program and the protocol for quality assurance OR state that no new dates are provided.* |

☐ Tick this box to confirm that the raw and calibrated dates are available in the paper or in Supplementary Information.

| | |
|---|---|
| Ethics oversight | *Identify the organization(s) that approved or provided guidance on the study protocol, OR state that no ethical approval or guidance was required and explain why not.* |

Note that full information on the approval of the study protocol must also be provided in the manuscript.

# Animals and other research organisms

Policy information about studies involving animals; ARRIVE guidelines recommended for reporting animal research, and Sex and Gender in Research

| | |
|---|---|
| Laboratory animals | Germ-free mice: Five to six week-old male and female C57BL/6J mice were bred in house (Gnotobiotic Mouse Facility, Tübingen). Specific pathogen free mice: male specific pathogen free C57BL/6J mice (cat. no. 632C57BL/6J) were purchased from Charles River Laboratories (Sulzfeld, Germany, Room A004) at the age of 35-41 days. Animals were housed under a 12:12-hour light-dark cycle at a temperature of 22 ± 2 °C and a relative humidity of 50–56%. |
| Wild animals | The study did not involve wild animals |
| Reporting on sex | Gnotobiotic animals: Female (n = 25) and male (n = 14) mice, SPF animals: only male mice |
| Field-collected samples | No field samples were collected in this study |
| Ethics oversight | Animal experiments were approved by the local authorities in Tübingen, Germany (Regierungspräsidium Tübingen, H02/20G and H02/21G). |

Note that full information on the approval of the study protocol must also be provided in the manuscript.

# Clinical data

Policy information about clinical studies

All manuscripts should comply with the ICMJE guidelines for publication of clinical research and a completed CONSORT checklist must be included with all submissions.

| | |
|---|---|
| Clinical trial registration | *Provide the trial registration number from ClinicalTrials.gov or an equivalent agency.* |
| Study protocol | *Note where the full trial protocol can be accessed OR if not available, explain why.* |
| Data collection | *Describe the settings and locales of data collection, noting the time periods of recruitment and data collection.* |
| Outcomes | *Describe how you pre-defined primary and secondary outcome measures and how you assessed these measures.* |

# Dual use research of concern

Policy information about dual use research of concern

## Hazards

Could the accidental, deliberate or reckless misuse of agents or technologies generated in the work, or the application of information presented in the manuscript, pose a threat to:

No | Yes
- ⊠ ☐ Public health
- ⊠ ☐ National security
- ⊠ ☐ Crops and/or livestock
- ⊠ ☐ Ecosystems
- ⊠ ☐ Any other significant area

## Experiments of concern

Does the work involve any of these experiments of concern:

No | Yes
- ⊠ ☐ Demonstrate how to render a vaccine ineffective
- ⊠ ☐ Confer resistance to therapeutically useful antibiotics or antiviral agents
- ⊠ ☐ Enhance the virulence of a pathogen or render a nonpathogen virulent
- ⊠ ☐ Increase transmissibility of a pathogen
- ⊠ ☐ Alter the host range of a pathogen
- ⊠ ☐ Enable evasion of diagnostic/detection modalities
- ⊠ ☐ Enable the weaponization of a biological agent or toxin
- ⊠ ☐ Any other potentially harmful combination of experiments and agents

# Plants

| | |
|---|---|
| Seed stocks | *Report on the source of all seed stocks or other plant material used. If applicable, state the seed stock centre and catalogue number. If plant specimens were collected from the field, describe the collection location, date and sampling procedures.* |
| Novel plant genotypes | *Describe the methods by which all novel plant genotypes were produced. This includes those generated by transgenic approaches, gene editing, chemical/radiation-based mutagenesis and hybridization. For transgenic lines, describe the transformation method, the number of independent lines analyzed and the generation upon which experiments were performed. For gene-edited lines, describe the editor used, the endogenous sequence targeted for editing, the targeting guide RNA sequence (if applicable) and how the editor was applied.* |
| Authentication | *Describe any authentication procedures for each seed stock used or novel genotype generated. Describe any experiments used to assess the effect of a mutation and, where applicable, how potential secondary effects (e.g. second site T-DNA insertions, mosiacism, off-target gene editing) were examined.* |

# ChIP-seq

## Data deposition

☐ Confirm that both raw and final processed data have been deposited in a public database such as GEO.

☐ Confirm that you have deposited or provided access to graph files (e.g. BED files) for the called peaks.

Data access links
*May remain private before publication.*
*For "Initial submission" or "Revised version" documents, provide reviewer access links. For your "Final submission" document, provide a link to the deposited data.*

Files in database submission
*Provide a list of all files available in the database submission.*

Genome browser session
(e.g. UCSC)
*Provide a link to an anonymized genome browser session for "Initial submission" and "Revised version" documents only, to enable peer review. Write "no longer applicable" for "Final submission" documents.*

## Methodology

Replicates
*Describe the experimental replicates, specifying number, type and replicate agreement.*

Sequencing depth
*Describe the sequencing depth for each experiment, providing the total number of reads, uniquely mapped reads, length of reads and whether they were paired- or single-end.*

Antibodies
*Describe the antibodies used for the ChIP-seq experiments; as applicable, provide supplier name, catalog number, clone name, and lot number.*

Peak calling parameters
*Specify the command line program and parameters used for read mapping and peak calling, including the ChIP, control and index files used.*

Data quality
*Describe the methods used to ensure data quality in full detail, including how many peaks are at FDR 5% and above 5-fold enrichment.*

Software
*Describe the software used to collect and analyze the ChIP-seq data. For custom code that has been deposited into a community repository, provide accession details.*

# Flow Cytometry

## Plots

Confirm that:

☐ The axis labels state the marker and fluorochrome used (e.g. CD4-FITC).

☐ The axis scales are clearly visible. Include numbers along axes only for bottom left plot of group (a 'group' is an analysis of identical markers).

☐ All plots are contour plots with outliers or pseudocolor plots.

☐ A numerical value for number of cells or percentage (with statistics) is provided.

## Methodology

Sample preparation
*Describe the sample preparation, detailing the biological source of the cells and any tissue processing steps used.*

Instrument
*Identify the instrument used for data collection, specifying make and model number.*

Software
*Describe the software used to collect and analyze the flow cytometry data. For custom code that has been deposited into a community repository, provide accession details.*

Cell population abundance
*Describe the abundance of the relevant cell populations within post-sort fractions, providing details on the purity of the samples and how it was determined.*

Gating strategy
*Describe the gating strategy used for all relevant experiments, specifying the preliminary FSC/SSC gates of the starting cell population, indicating where boundaries between "positive" and "negative" staining cell populations are defined.*

☐ Tick this box to confirm that a figure exemplifying the gating strategy is provided in the Supplementary Information.

# Magnetic resonance imaging

## Experimental design

Design type
*Indicate task or resting state; event-related or block design.*

| Design specifications | *Specify the number of blocks, trials or experimental units per session and/or subject, and specify the length of each trial or block (if trials are blocked) and interval between trials.* |
|---|---|
| Behavioral performance measures | *State number and/or type of variables recorded (e.g. correct button press, response time) and what statistics were used to establish that the subjects were performing the task as expected (e.g. mean, range, and/or standard deviation across subjects).* |

## Acquisition

| Imaging type(s) | *Specify: functional, structural, diffusion, perfusion.* |
|---|---|
| Field strength | *Specify in Tesla* |
| Sequence & imaging parameters | *Specify the pulse sequence type (gradient echo, spin echo, etc.), imaging type (EPI, spiral, etc.), field of view, matrix size, slice thickness, orientation and TE/TR/flip angle.* |
| Area of acquisition | *State whether a whole brain scan was used OR define the area of acquisition, describing how the region was determined.* |

Diffusion MRI  ☐ Used   ☐ Not used

## Preprocessing

| Preprocessing software | *Provide detail on software version and revision number and on specific parameters (model/functions, brain extraction, segmentation, smoothing kernel size, etc.).* |
|---|---|
| Normalization | *If data were normalized/standardized, describe the approach(es): specify linear or non-linear and define image types used for transformation OR indicate that data were not normalized and explain rationale for lack of normalization.* |
| Normalization template | *Describe the template used for normalization/transformation, specifying subject space or group standardized space (e.g. original Talairach, MNI305, ICBM152) OR indicate that the data were not normalized.* |
| Noise and artifact removal | *Describe your procedure(s) for artifact and structured noise removal, specifying motion parameters, tissue signals and physiological signals (heart rate, respiration).* |
| Volume censoring | *Define your software and/or method and criteria for volume censoring, and state the extent of such censoring.* |

## Statistical modeling & inference

| Model type and settings | *Specify type (mass univariate, multivariate, RSA, predictive, etc.) and describe essential details of the model at the first and second levels (e.g. fixed, random or mixed effects; drift or auto-correlation).* |
|---|---|
| Effect(s) tested | *Define precise effect in terms of the task or stimulus conditions instead of psychological concepts and indicate whether ANOVA or factorial designs were used.* |

Specify type of analysis:  ☐ Whole brain   ☐ ROI-based   ☐ Both

| Statistic type for inference<br>(See Eklund et al. 2016) | *Specify voxel-wise or cluster-wise and report all relevant parameters for cluster-wise methods.* |
|---|---|
| Correction | *Describe the type of correction and how it is obtained for multiple comparisons (e.g. FWE, FDR, permutation or Monte Carlo).* |

## Models & analysis

| n/a | Involved in the study |
|---|---|
| ☐ | ☐ Functional and/or effective connectivity |
| ☐ | ☐ Graph analysis |
| ☐ | ☐ Multivariate modeling or predictive analysis |

| Functional and/or effective connectivity | *Report the measures of dependence used and the model details (e.g. Pearson correlation, partial correlation, mutual information).* |
|---|---|
| Graph analysis | *Report the dependent variable and connectivity measure, specifying weighted graph or binarized graph, subject- or group-level, and the global and/or node summaries used (e.g. clustering coefficient, efficiency, etc.).* |
| Multivariate modeling and predictive analysis | *Specify independent variables, features extraction and dimension reduction, model, training and evaluation metrics.* |

