## [Peer Review File · Nature]

Non-antibiotics disrupt colonization resistance against enteropathogens

Corresponding Author: Professor Lisa Maier

Version 1:

Reviewer comments:

Referee #1

(Remarks to the Author)

In this intriguing study, Griebhammer, Cuesta-Zuluaga, Zahir, et al investigate how non-antibiotic drugs impact infections with enteric pathogens, in particular Salmonella. The overall idea for this study is that non-antibiotic drugs disproportionately impede physiological functions of commensal gut microbes, leading to a disruption of normal microbiota functions. In contrast, enteric pathogens appear to be fairly resistant to the negative effects of non-antibiotic drugs and the decreased colonization resistance provided by a disrupted microbiota allows for increased pathogen susceptibility and colonization. By comparison, commensal bacteria were inhibited more readily than pathogenic bacteria. Using a small, defined microbial community in vitro and in a gnotobiotic mouse model, authors identify a subset of a non-antibiotic drugs that perturb community structure and allow for pathogen outgrowth. A second subset of drugs decreased biomass of the community; the decrease in biomass itself appears to be sufficient to allow pathogen growth. The ability to exclude/suppress Salmonella growth in vitro could not be traced back to a single organism in the synthetic community. Addition of *E. coli* as a nutritional competitor decreased Salmonella growth; non-antibiotic drugs that negatively affected *E. coli* growth allowed for Salmonella expansion, suggesting that niche competition is driving the observed phenotypes. Similar observations are made for other enteric pathogens, and using complex microbial communities derived from human fecal samples. In mice colonized with a defined microbiota or conventionally SPF-raised animals, treatment with relevant doses of select non-antibiotic drugs increased Salmonella colonization by up to 100-fold.

The topic of how non-antibiotic drugs influence host-associated microbial communities is very timely; there is a great interest in understanding the molecular details of these interactions, and as such, this study seeks to address a significant gap in our understanding. The experiments presented here are quite elegant, and the writing is very clear.

Major concerns:

1. This work is based on a previously published study (Nature. 2018 Mar 29;555(7698):623-628) where a panel of FDA approved drugs is added to bacterial cultures of commensal gut bacteria to investigate the impact of these drugs on bacterial growth; here, this work is extended to five enteric pathogens. It is quite established that perturbations of the gut microbiota structure will lead to increased susceptibility to enteric infections (for example, Infect Immun. 2008 Oct;76(10):4726-36, Cell Host Microbe. 2007 Aug 16;2(2):119-29, PLoS Biol. 2007 Oct;5(10):2177-89). Given these prior observations, the novelty of the conceptual advance needs to be better established.

2. Along these lines, the authors argue that the mechanism at work here is non-antibiotic drugs affecting nutritional competition. The importance of nutritional competition in excluding pathogens is well established (for example, Cell Host Microbe. 2020 Oct 7;28(4):526-533.e5., Science. 2012 Jun 8;336(6086):1325-9), in particular commensal *E. coli* competing with pathogens for limited resources.

3. It is not clear if the findings from the Com20 community can be readily applied to a more complex ecosystem. The fact that this community is unable to control Salmonella colonization at all (Fig 7b) compared to a complex community, raises significant concerns. Furthermore, a key aspect of the in vitro work, i.e. the importance of *E. coli* as a nutritional competitor, is not recapitulated in this in vivo setting. This raises a significant concern in that the main conclusion (non-antibiotic drugs alter nutritional competition) may -or may not- hold up in a complex microbiota in a mammalian host.

4. The role of the host has not been explored. Prior work has shown that oral administration of antibiotics change microbial metabolism, in particular production of short chain fatty acids, which impact host physiology and metabolism. Absence of short chain fatty acids influences oxygen availability in the gut and supports growth of enteric pathogens (Science. 2017 Aug 11;357(6351):570-575.; Cell Host Microbe. 2019 Jan 9;25(1):128-139.e5.). It is conceivable that in a natural setting (complex microbiota inside a host), the action of non-antibiotic drugs act on the gut microbiota/enteric pathogens would involve the host (rather than nutritional competition).

Minor comments:

Line 248-264: The use of PICRUSt2 on such a small, define community to demonstrate that microbial metabolism changes when the community structure changes is not appropriate as it generates a very predictable outcome. This approach also appears to have generated little detailed insights. I would suggest removing this experiment. The subsequent modeling is far more powerful.

Referee #2

(Remarks to the Author)

In this paper, the authors thoroughly investigate the impact of non-antibiotic drugs on commensal microbe community and consequent influence on colonization resistance against enteric pathogens. The authors first examined the effects of 1,200 FDA-approved drugs on the growth of gamma-proteobacteria and commensal bacteria. They find that Proteobacteria strains exhibit sensitivity to fewer drugs than commensals, and the sensitivity of Proteobacteria is evident only at higher concentrations of drugs. Then, the authors examined the effects of selected 67 drugs on commensal community models consisting of 20/21 strains. Through in vitro and in vivo experiments, they demonstrate that treatment with certain non-antibiotic drugs compromises the commensal community-mediated colonization resistance against Salmonella. The influence of non-antibiotic drugs is shown to reduce the overall bacterial amounts, alter the composition, and/or presumably modify the function of the commensal community, facilitating the colonization and growth of Salmonella. The authors suggested that altering the metabolic potential of the commensal population might create an open niche for Salmonella. The impact of non-antibiotic drugs extended beyond Salmonella, potentially promoting colonization by other Proteobacteria, indicating the generalizability of the authors' findings. While certain disparities between in vitro and in vivo results, along with varied outcomes in the context of different commensal communities or host conditions, hinder the establishment of universal rules, I recognize the work as valuable and well-conducted. This paper offers highly valuable knowledge that must be considered in clinical practice, as well as fundamental insights into the complex interactions of the gut microbiota members. The importance of this research could be further enhanced by addressing the following points:

1. There are some concerns regarding the clarity of certain figure panels and their legends. Several of the figure panels are quite intricate, making them challenging to decipher. Indeed, Figures 1b, 2d, 3a, 3b, 4a, 4b, 4d, 5c, 6a, 6b, 7bcd, Supp Figures 1a, Supp 2a, 2e, 2f, 2l, Supp 3b, Supp 5a, Supp 7c, 7d are difficult to comprehend without a deep knowledge of informatics and perhaps mathematics. For instance, in lines 286-287, the authors assert that "E. coli, a member of the Enterobacteriaceae family with metabolic characteristics similar to S. Tm (Figure 4a)." However, grasping this claim proves challenging as Figure 4a does not explicitly illustrate the basic metabolic characteristics of each individual strain; instead, the authors present the metabolic competition index. I would recommend the authors to make the figures a bit more intuitively understandable.

In addition, figure legends are often too brief and may not be sufficient to assist readers in understanding the complexities of the figures. To make the findings more accessible to a broader audience, consider providing more detailed explanations and context for the key elements in each figure. I understand that space constraints can be a concern, but a more comprehensive legend would undoubtedly improve the overall readability of the paper.

2. The presented data compellingly unveil heightened resistance to non-antibiotic drugs in Proteobacteria pathogens—a discovery of great intrigue. Nonetheless, the mechanisms underlying the effects of these drugs on commensals and pathogens employed in this study remain elusive. A comprehensive examination of the shifts in transcriptome landscape induced by drugs, such as terfenadine, clomiphene, and floxuridine, in both the pathogenic and commensal strains would provide invaluable mechanistic insights. Furthermore, by harnessing the genetic manipulability of Salmonella or other Proteobacteria strains, delving into the genes associated with resistance to non-antibiotic drugs via a mutant library could prove instrumental in elucidating the intricate molecular pathways involved.

3. In this study, the authors investigated the impacts of non-antibiotic drugs on both pathogen and commensal strains primarily by investigating their growth or abundance. However, the authors have yet to explore the functional consequences of these strains. In this regard, transcriptome analyses of pathogen and commensal strains would also be highly valuable.

4. Are there any common characteristics in the chemical structure of drugs that facilitate the influence on commensal strains while exerting less influence on pathogens? Understanding the specific interactions between drug chemical structures and commensal microbiota, as opposed to pathogens, could provide crucial insights into designing drugs that minimize impact on commensal organisms. Exploring these nuances in drug interactions holds significant implications for developing targeted and more tailored therapeutic strategies.

5. To establish clinical relevance, it is imperative to verify the impact of non-antibiotic drugs within the complex human gut microbiota context in vivo. This can be achieved, for instance, by employing germ-free mice colonized with human stool samples containing Proteobacteria.

6. Regarding difference and overlap of metabolic capabilities of 20/21 commensal communities and Salmonella, the authors used PICRUSt2 to predict pathway abundance. I appreciate the utility of PICRUSt2 for rough predictive functional analysis at the species level; however, it cannot account for differences between strains. Metabolic pathways are known to vary

significantly between strains, and relying solely on PICRST analysis may be insufficient. I would recommend the author to consider validating the analysis through in vitro experiments. Alternatively, conducting whole genome sequencing of each strain followed by transcriptome analysis could offer a comprehensive understanding of the metabolic activities within the bacterial communities. Such efforts would significantly strengthen the reliability and impact of the conclusions.

7. Supplementary Fig 4 should examine and display the Com21 and Com20 communities side by side.

End.

Referee #3

(Remarks to the Author)

The manuscript aims to investigate the effect of non-antibiotic drugs on the microbiota and pathogenic gammaproteobacteria. The authors find that pathogens are, in general, more resistant than commensals to non-antibiotic drugs, and that some of these drugs reduce colonization resistance to the pathogen *Salmonella*. By testing synthetic and human stool-derived communities, as well gnotobiotic mice and SPF mice, they find that non-antibiotic drugs favored *Salmonella* expansion in vitro, and at day 1 post-infection.

The study has the merit of exploring the role of non-antibiotic drugs in altering the gut microbiome composition and promoting pathogen expansion, likely by reducing colonization resistance. These findings are important because there is little known about the possible consequences of non-antibiotic drugs in altering colonization resistance to pathogens in general. The finding that gammaproteobacteria are, in general, more resistant to non-antibiotic drugs also has important implications. The manuscript is also generally well-conducted and well-written.

1. My primary concern about the study is that there is no mechanism for the presented observations. The reason why gammaproteobacteria are more resistant to most non-antibiotic drugs is not investigated. Similarly, the effects of non-antibiotic drugs on the gut microbiota are unclear, and not all the in vitro phenotypes are recapitulated in vivo. I suggest that the authors choose one of the drugs and follow up their findings with mechanistic studies, at the very minimum, to identify which specific changes in the gut microbiota would enhance pathogen colonization.

2. Related to understanding possible mechanisms, the mouse studies in Figure 7 should be expanded to include an analysis of the host responses and possibly additional timepoints. The experiments here show a disruption of colonization resistance at early stages post-infection, but again the mechanism is not completely elucidated. Even though no inflammation was observed (which is expected at day 1 post-infection), it is possible that other more subtle changes in gene expression may have occurred. For example, induction of iNos, or increased epithelial oxygenation. Bottom line, it needs to be shown whether administration of the non-antibiotic drugs in vivo affects the host in addition to the microbiota, before and after infection.

3. The animal experiments in Fig. 7 show that the non-antibiotic drug treatment disrupts colonization resistance and results in a higher colonization level of *Salmonella* at day 1 post-infection. A question remains whether the non-antibiotic drugs also increase the severity of infection. This would require later time points.

Referee #4

(Remarks to the Author)

This is a fascinating study by Griebhammer and colleagues which builds on previous work from the same group showing that many non-antibiotic drugs can have inhibitory effects on human gut commensal bacteria. Here they show that some of these drugs can increase ability of *Salmonella Typhimurium* and other Gammaproteobacteria members to invade model microbial communities, ex-vivo microbial communities and in mouse models. These are important and original findings with wide reaching implications. I have some questions regarding the methodology and several suggestions that I think would help strengthen the findings. Please see below.

Line 91 and Fig. S1. 'these pathogens showed different sensitivity profiles compared to a panel of commensal gut bacteria'. To help determine how expected or unexpected this is it would be useful to show the genetic distance between the species tested. I.e. would the large difference in sensitivity profiles of the Gammaproteobacteria compared to other species be a consequence from their large evolutionary distance to the other species tested? There appear to be other clusters of similarity e.g. the *Bacteriodes* spp. At a minimum showing the phylogenetic tree of the bacteria tested would help, but a more rigorous approach could be to plot genetic distance against to drug sensitivity distance to show if these correlate and if Gammaproteobacteria are outliers or not. The comparison of gram positive to gram negative could also be informative in explaining the sensitivity profile of Gammaproteobacteria compared to other species.

Fig. 1 please define 'active'. I'm inferring that it is the number of drugs which were inhibitory to the species tested.

Line 111: 'while only 14% of the pathogens were affected'. This sentence is worded incorrectly. There are only 5 pathogens, so you can't get 14% of them. I believe you mean something to the effect of '14% of pathogen-drug combinations were inhibitory'. Please rephrase for correctness.

119. 'a synthetic community comprising the 20 gut commensals'. Fig. 1 legend says 19 commensals + the pathogen. Are there 20 commensals + pathogen, or 19 commensals + pathogen?

Line 122: It's unclear from Supplementary Figure 2a how the 61.3% overlap of metabolic pathways was calculated. It may be cleared to add the number of total metabolic pathways and number of pathways present Com20.

Line 124. 'and robustly colonized the gastrointestinal tract of mice for up to 57 days'. The figure shows that the mice are robustly colonized in the sense that the biological repeats show similar relative abundances. However, it doesn't look terribly robust temporally – many species appear to be lost or at undetectable levels by 6 days?

Line 130 and Methods. Is the drug washed out at the 24h timepoint before *S. Tm* challenge? If I have understood the protocol correctly then the drug is present, but there is a 10-fold dilution into fresh mGAM during the invasion step. This is important to make explicitly clear, since the presence of the drug may also be directly affecting the *S. Tm* growth, even if diluted. The authors note that they exclude drugs that directly inhibited *S. Tm* growth, I'm assuming this is because the drug remained present. What was the threshold used to determine *S. Tm* inhibition here- IC25? If so, this could mean some modestly *S. Tm* inhibiting drugs were still present, which may affect the results. Can this be ruled out by a control which includes a wash step?

Fig. 2b. OD could be a relative inaccurate measure of biomass in the complex community, especially if cidal drugs are used and when low or high ODs are reached. Some controls to get cell counts per ml, for example with flow cytometry, would be helpful here to verify that OD is a reasonable reporter of cells/ml.

line 142 'we tested 52 of the 67 drugs', line 296 'We challenged Com21 with 48 out of 52 drugs', line 318 '12 out of 52 drugs' etc. In general, I had a very difficult time working out the numbers of drugs used for different experiments and the inclusion/exclusion criteria. A different number seems to be used in almost every experiment. The methods do not adequately explain the selection criteria: e.g. Line 893 'we selected 30 drugs representing diverse inhibition spectra across Com20 members. Additionally, we incorporated 25 clinically relevant drugs, resulting in a final selection of 52 drugs' These numbers don't add up – I assume that this is due to exclusions mentioned later, but it is unclear. Drugs were taken out for *S. Tm*, are later introduced in other assays, so mentioning that they are still incorporated in other assays would be useful. Also, what do the authors mean by 25 clinically relevant drugs, and what are they – all the 1200 drugs originally screened are clinically relevant? A supplementary flow diagram or similar, with clear inclusion/exclusion criteria might help readers to understand.

Similarly, to the above, the number of bacteria tested varies by experiment and the inclusion criteria are unclear. A clear set of tables with the species present for each experiment would be helpful.

Line 201: 'we manually generated four communities whose composition resembled that of drug-treated communities'. If I have understood the protocol correctly then then a 10-fold dilution of the drug remains present during the *S. Tm* challenge in the drug-treated assay. Some inhibited species may remain inhibited during the 4.5 h *S. Tm* challenge under these conditions, whereas in the drug-mimicking communities they do not. Could this explain the mimicking conditions that did not fully phenocopy treatment? A drug wash step should answer this.

Line 225 please specify 16S rRNA amplicon sequencing

Line 226 mentions '*S. Tm*-favoring communities' then references figure 3a. but the legend of figure 3a. says '*S. Tm*-favoring treatments'. Since both treated and treatment-mimicking communities are used in the study this needs clarification.

Line 254: Please expand on what is meant by "metagenome potential"

Line 255 a brief explanation of what is meant by 'functional capacity' would be useful.

Line 324-331: "The effect of certain drugs was pathogen-specific.....zafirlukast restricted the growth of *S. flexneri* 24570, *Y. pseudotuberculosis* YPIII, and *Y. enterocolitica* WA-314 but favored the growth of *E. coli* CFT073". Does this pathogen-specific effect relate to the community-disrupting effect, or to the sensitivity of the pathobionts/pathogens to said drug? Do these drugs (e.g. clomiphene, chlorpromazine, zafirlukast) have an inhibiting effect on the pathobionts/pathogens in pure cultures?

Line 400-402: "changes in the abundance of 4 and 5 species.....Similarly, 31, 2 and 1 amplicon sequencing variants were significantly different..." Which species and sequencing variants were identified? Are these species/sequencing variants unique to their respective mice model?

Line 837-838: "monoculture were mixed in equal ratios". Was this based on volume, OD or cell number?

Line 948. Community assembly: in previous papers by the same group the community has been assembled based on equal CFU/ml rather than OD, presumably to obtain more equal distribution of each species. Why was this not done here? Are the Com20 members in stationary phase or exponential phase when added to the drug plates? The drugs may have quite different effects on the bacteria depending on growth phase.

Lines 989-990: Please expand on how the luminescence and OD values were normalized.

Lines 1213-1222: For the Germfree mouse both female and male mouse were included, for the SPF mice only male. Male and female mice have on occasion been shown to respond different to treatment, why was it decided to use only male SPF mice? Also please include the number of SPF mice used.

Minor points

- Title, Line 47, 79,85, 100, 102: Bacteria class should not be italicized
 - Line 516-517: Order name also doesn't need to be italicized
 - Line 88: "approximately 1200 FDA-approves drugs", please include the exact number.
 - Line 938: stating specifically the 5 concentrations would be more helpful.
 - Line 1027: OD578 format
 - Throughout paper: inconsistency of numerical formatting written vs numerals e.g Line 973, 1019, 1255, 1286.
 - Line 1048: please include the pooled concentration
-
- Figure 3b: the bacteria in grey are hard to read against the white background, it may be better to change to a different colour.
 - Figure 1b: I'm still not clear on the difference between >10 and 10-20 or >20 and 20-40 IC25?
 - In figure 7: where is part e?
 - Figure S1b: For better readability it may be good to include another colour strip to highlight antibiotic vs non-antibiotic drugs.

Version 2:

Reviewer comments:

Referee #1

(Remarks to the Author)

1. Additional experiments using humanized mice have been added to demonstrate the concern that the phenotypes observed here are specific to low-complexity microbiomes, and immune profiling is performed to investigate as to whether certain drugs affect host status. These experiments adequately address these major concerns, and nicely extend the current work.
2. I think it would be important to show that increased colonization with Salmonella during Terfenadine treatment has quantifiable consequences on the host. Therefore, I would suggest to move pertinent panels of Extended Data Figure 11 to the main figure, e.g. Figure 4.
3. There is still some lingering concerns that conceptually, the effect of non-antibiotic drugs on colonization resistance is very similar to the effect of antibiotics (which is well documented), i.e. decreased biomass, decreased microbiome diversity, and decreased nutritional competition, which should be addressed.

Referee #2

(Remarks to the Author)

In the revised manuscript, substantial modifications have been made to the main text, figures, and legends, improving readability and clarifying key points. However, I believe the analysis of the molecular mechanisms underlying the effects of drugs on commensal bacteria requires further depth. While these mechanisms are likely to be case-by-case, depending on the drug and specific context (as clearly demonstrated in the manuscript), without presenting at least one concrete molecular mechanism—ideally a novel one—the study risks being perceived as merely a collection of observations, which would be unfortunate.

For example, in Fig. 3, the manuscript concludes that the reduced fitness of *fliD*-deficient Salmonella under tiratricol treatment suggests a decrease in fumarate availability as a key factor. However, the precise molecular events underlying this effect remain unclear. As mentioned in the discussion, "Tiratricol could cause lysis of commensal microorganisms, altering the pool of available substrates such as microbiota-derived fumarate," but the specific commensal member(s) directly affected by tiratricol and the molecular mechanisms driving this effect should be further elucidated.

Additionally, in the final HUM mice experiment, terfenadine was used, yet its effects were not shown in Fig. 3. Given that terfenadine was consistently used throughout the study, further investigation into its molecular mechanisms of action seems warranted.

One final point: the manuscript categorizes three major mechanisms through which non-antibiotic drugs promote pathogen expansion. A summary figure illustrating which tested drugs correspond to each category would enhance clarity and improve the overall structure of the findings.

Referee #3

(Remarks to the Author)

The authors have addressed my prior concerns and have added new data and new analyses that further strengthen the manuscript's conclusions.

(Remarks on code availability)

I'm not the best person to review the code, sorry

Referee #4

(Remarks to the Author)

Grießhammer and colleagues have very comprehensively addressed the reviewer comments in this revised manuscript. In particular, the further analysis to help explain the different drug sensitivity observed in Gammaproteobacteria compared to other gut commensals has improved the manuscript. Similarly, the experiments with a wash step between treatment and pathogen is an important control which has now been added. I have no further suggestions.

Point by point response to reviewer comments

We thank the reviewers for their constructive and insightful feedback. Their comments, questions, and suggestions have greatly improved our manuscript and reinforced its core messages. Our major revisions focused on three key areas:

1. Explaining differential drug sensitivity: We now provide detailed explanations for the observed differences in drug sensitivity between pathogenic *Gammaproteobacteria* and gut commensals. In addition to the well-known higher resistance of Gram-negative bacteria due to their selective outer membrane, these differences are also linked to the abundance and efficiency of stress response mechanisms and efflux pumps (**ED Fig. 3**, section *Gut commensal bacteria are more sensitive to non-antibiotic drugs compared to Gammaproteobacteria species*).

2. Elucidating functional changes in the microbial communities: We expanded and refined our analyses of the composition of the microbial community after drug treatment and how it is associated with pathogen growth. We examined microbial diversity, performed a comprehensive analysis of changes in individual taxa, and interrogated alterations in gene expression in both pathogen and commensals. For the *E. coli*-promoting drug tiratricol, we show that fumarate respiration is an important driver of *S. Tm* expansion. (**Fig. 2 and 3, ED Fig. 6-9**, sections *Drug-induced changes in community structure can lead to altered pathogen growth*, *S. Tm modifies its transcriptional profile after drug treatment according to the community context* and *Presence of a phylogenetically related niche competitor in the community hampers S. Tm expansion after drug treatment*).

3. Extending *in vivo* findings: We now present evidence that our *in vivo* results with the antihistaminic drug terfenadine in the Com20 and SPF models are reproducible in a humanized microbiome mouse model. We also demonstrate that increased pathogen colonization leads to inflammation at earlier time points, with a faster progression of infection in drug-treated mice compared to vehicle-treated controls (**Fig. 4, ED Fig. 10 and 11**, section *Non-antibiotic drugs disrupt colonization resistance against S. Tm in mice*).

Addressing the reviewers' comments led to substantial revisions and additions throughout, which clarified the main message of the manuscript. In the updated manuscript, we now include four main figures and 11 extended data figures. As suggested, certain figure panels from the original submission (5b, Suppl. Fig. 1a, Suppl. Fig. 2f, h, Suppl. Fig. 3, Suppl. Fig. 7a,b) were either removed or replaced. Our revisions and the new data address all reviewer concerns comprehensively. Below, we provide detailed responses to each point.

Reviewer #1

In this intriguing study, Griebhammer, Cuesta-Zuluaga, Zahir, et al investigate how non-antibiotic drugs impact infections with enteric pathogens, in particular Salmonella. The overall idea for this study is that non-antibiotic drugs disproportionately impede physiological functions of commensal gut microbes, leading to a disruption of normal microbiota functions. In contrast, enteric pathogens appear to be fairly resistant to the negative effects of non-antibiotic drugs and the decreased colonization resistance provided by a disrupted microbiota allows for increased pathogen susceptibility and colonization. By comparison, commensal bacteria were inhibited more readily than pathogenic bacteria. Using a small, defined microbial community in vitro and in a gnotobiotic mouse model, authors identify a subset of a non-antibiotic drugs that perturb community structure and allow for pathogen outgrowth. A second subset of drugs decreased biomass of the community; the decrease in biomass itself appears to be sufficient to allow pathogen growth. The ability to exclude/suppress Salmonella growth in vitro could not be traced back to a single organism in the synthetic community. Addition of *E. coli* as a nutritional competitor decreased Salmonella growth; non-antibiotic drugs that negatively affected *E. coli* growth allowed for Salmonella expansion, suggesting that niche competition is driving the observed phenotypes. Similar observations are made for other enteric pathogens, and using complex microbial communities derived from human fecal samples. In mice colonized with a defined microbiota or conventionally SPF-raised animals, treatment with relevant doses of select non-antibiotic drugs increased Salmonella colonization by up to 100-fold.

The topic of how non-antibiotic drugs influence host-associated microbial communities is very timely; there is a great interest in understanding the molecular details of these interactions, and as such, this study seeks to address a significant gap in our understanding. The experiments presented here are quite elegant, and the writing is very clear.

We thank the reviewer for their support of our manuscript!

Major concerns

Comment 1.1

This work is based on a previously published study (Nature. 2018 Mar 29;555(7698):623-628) where a panel of FDA approved drugs is added to bacterial cultures of commensal gut bacteria to investigate the impact of these drugs on bacterial growth; here, this work is extended to five enteric pathogens. It is quite established that perturbations of the gut microbiota structure will lead to increased susceptibility to enteric infections (for example, Infect Immun. 2008 Oct;76(10):4726-36, Cell Host Microbe. 2007 Aug 16;2(2):119-29, PLoS Biol. 2007 Oct;5(10):2177-89). Given these prior observations, the novelty of the conceptual advance needs to be better established.

The reviewer is right in pointing out that it is well-established that perturbations caused by broad-spectrum antibiotics can reduce colonization resistance. Our work builds upon both the references cited by the reviewer and the study by Maier et al. (2018) by showing that even non-antibiotics, which typically exert a narrow spectrum of inhibition on bacteria, can alter the composition and function of the microbiome in ways that disrupt colonization resistance. This finding is important because it demonstrates that even subtle, targeted changes in microbiome composition can undermine colonization resistance.

For antibiotics, the reduction in colonization resistance can be largely attributed to an overall decrease in microbial biomass (**ED Fig. 1b**). However, with non-antibiotic drugs, other ecological dynamics come to the forefront, such as shifts in community composition

and function, including specific drug effects on direct niche competitors (**Fig. 2 and 3, ED Fig. 6-9**). These changes require *Salmonella* to employ different invasion strategies, as reflected in the distinct transcriptomic profiles of *S. Tm* in communities exposed to biomass-reducing drugs versus those that alter community composition without perturbing biomass (**Fig. 2e, f, g and ED Fig. 8**).

Finally, from a pharmacological perspective, it has previously been unclear why certain drugs are associated with higher infection risks or immune side effects. Our findings suggest that these effects may be microbiome-mediated.

Action taken: In the revised manuscript, we restructured the introduction and the discussion to make clear the relevance of our study and the novelty of our results. In addition, we have included new experiments and analyses focused on non-antibiotics and their impact on the composition of the microbial community.

Comment 1.2

Along these lines, the authors argue that the mechanism at work here is non-antibiotic drugs affecting nutritional competition. The importance of nutritional competition in excluding pathogens is well established (for example, *Cell Host Microbe*. 2020 Oct 7;28(4):526-533.e5., *Science*. 2012 Jun 8;336(6086):1325-9), in particular commensal *E. coli* competing with pathogens for limited resources.

Please note that we sought not to use the term ‘mechanism’ since it can be used to describe phenomena across very different scales, including ecological dynamics, cellular processes, and molecular interaction (or some combination thereof). A drug can interact with bacterial intracellular structures at a molecular level, resulting in a series of altered cellular processes that, depending on the severity of the alteration, can impact the bacterium’s growth or even lead to death, potentially altering the ecological dynamics of the microbial community.

The goal of our study was to assess the effects of a wide range of non-antibiotic drugs on the microbiome and their impact on colonization resistance. Drugs vary

substantially in terms of chemical composition, molecular targets, consumption patterns, dosages, and the potential effects of drug-drug interactions in polypharmacy. Likewise, there is substantial variation in gut microbiome composition across individuals. A similar disruption of the microbiome can be reached in multiple ways. Rather than identifying a single molecular interaction or cellular process that is being affected in one or a limited set of species, the novelty and significance of our study lies in understanding the broader influence of drugs on the ecology of the microbial community and its influence on colonization resistance.

Based on our initial findings, along with our new experiments in this revision and recent research from other groups (Spragge et al., 2023), we posit that drug-induced loss of colonization resistance can largely be understood as the result of three non-mutually exclusive ecological changes within the microbial community:

1. *Reduction of microbial biomass*, as evidenced by the correlation between *S. Tm* levels and the optical density of the community after treatment or dilution (**Fig. 1d**).

2. *Alterations in microbial community composition or diversity*, as demonstrated by the relationship between species richness and *S. Tm* growth, after accounting for community biomass. In addition, this association is supported by changes in abundance of specific commensals in treatments that favor *S. Tm*, as well as the pathogen's ability to grow on synthetic communities built to mimic the composition observed during drug treatments without biomass loss (**Fig. 2a-d, ED Fig. 6, 7**).

3. *Context-dependent competition for nutritional resources* with one or more resident microbes, as shown by changes in pathogen growth in a 20- or 21-member community after treatment with drugs targeting *E. coli* (**Fig. 3a,b, ED Fig. 9a,b,c**). In the revised manuscript, we now show that at the molecular level, fumarate respiration is a key factor underlying the competition between *S. Tm* and *E. coli* in drug-treated communities (**Fig. 3d,e**).

Action taken: We have rewritten and reorganized large portions of the text to emphasize these key points. In particular, we now emphasize in the introduction (lines 69 - 74) that it has been unclear to what extent drug-induced changes in the gut microbiome

may increase the risk of enteric pathogen infections, highlighting the need for systematic studies to address this knowledge gap. The results section has been reorganized to clearly convey the importance of ecological dynamics in influencing colonisation resistance following drug perturbations. In the discussion, we summarize the three key ecological changes described above that affect colonization resistance and provide examples of drugs for each case (lines 401 - 417).

Comment 1.3

It is not clear if the findings from the Com20 community can be readily applied to a more complex ecosystem. The fact that this community is unable to control *Salmonella* colonization at all (Fig 7b) compared to a complex community, raises significant concerns. Furthermore, a key aspect of the in vitro work, i.e. the importance of *E. coli* as a nutritional competitor, is not recapitulated in this in vivo setting. This raises a significant concern in that the main conclusion (non-antibiotic drugs alter nutritional competition) may -or may not- hold up in a complex microbiota in a mammalian host.

We agree with the reviewer that the complexity of the microbial community can play a key role in controlling the ability of pathogens to proliferate, as we highlight in **Fig. 4**. Indeed, we found that *S. Tm* CFU counts in SPF mice treated with the vehicle are lower than those in Com20-colonized gnotobiotic mice. However, we disagree with the statement that the community is unable to control colonization at all. Our Com20 model results in levels of colonization resistance that are expected for mice colonized with a low complex community. As a reference, Brugiroux and colleagues observed similar levels of *Salmonella* colonisation at 1 day post infection (d.p.i.) in OMM12-colonized mice as we did in this study using Com20-colonized mice (Brugiroux et al., 2016). Moreover, *S. Tm* colonisation at 1 d. p.i. in germ-free mice would be at least two orders of magnitude higher ($\sim 10^9$ CFUs/g) (Stecher et al., 2005).

Most importantly to the central message of our manuscript, in both Com20-colonized and SPF mouse models, we observed an increase in pathogen levels following drug treatment compared to control mice, demonstrating that even a simple community can significantly alter colonization dynamics. Starting with a simpler community and

progressing to more complex models enabled us to discriminate between the effects of closely related competitors from those mediated by other community members. Furthermore, in the revised manuscript, we include an experiment in which germ-free mice were colonized with a human donor microbiota containing *E. coli* and treated with the antihistamine terfenadine. Drug-treated mice exhibited increased *S. Tm* counts in both the feces and cecum at 4 d.p.i. compared to controls (**Fig. 4f, g**), yet we observed no differences in *E. coli* counts over the course of treatment (**ED Fig. 10h**). These new data indicate that the presence of a close competitor alone is not sufficient to prevent pathogen colonization if the broader community context is altered, a finding supported by recent studies relating colonisation resistance to microbiota diversity (Spragge et al., 2023).

Action taken: We have added the results of the experiment with humanized mice to the manuscript (**Fig. 4f,g; ED Fig. 10c,f,h**; lines 359 - 365). In the discussion, we emphasize that the reasons why a particular non-antibiotic drug affects colonization resistance can vary and must be investigated on a case-by-case basis (lines 431 - 435).

Comment 1.4

The role of the host has not been explored. Prior work has shown that oral administration of antibiotics change microbial metabolism, in particular production of short chain fatty acids, which impact host physiology and metabolism. Absence of short chain fatty acids influences oxygen availability in the gut and supports growth of enteric pathogens (Science. 2017 Aug 11;357(6351):570-575.; Cell Host Microbe. 2019 Jan 9;25(1):128-139.e5.). It is conceivable that in a natural setting (complex microbiota inside a host), the action of non-antibiotic drugs act on the gut microbiota/enteric pathogens would involve the host (rather than nutritional competition).

Indeed, we had omitted investigating the role of the host in the loss of colonization resistance caused by non-antibiotics. We agree with the reviewer that this is an important factor that should be investigated. Therefore, we have performed additional experiments to explore whether drug-induced changes in gut physiology prior to infection, such as the induction of inflammation, could create conditions that promote *S. Tm* colonisation.

Following the reviewer's suggestion, we performed these analyses using germ-free mice colonized with a complex human microbiota. We collected tissue sections from the cecum of these mice after 6 days of treatment with antihistamine terfenadine, and used immunohistochemistry with a panel of markers to assess whether drug treatment affects key cellular processes, including *immune cell infiltration* (CD8, CD11b, CD11c, F4/80), *apoptosis* (Cl. Casp3), *proliferation* (Ki67), *inflammation* (RelA), *hypoxic adaptation* (HIF-1 α), and *endothelial function* (CD31). We observed no significant effects in the host tissue at the site of initial *S. Tm* colonization and tissue infection (**ED Fig. 11**). We are aware that this is a single example and cannot exclude the possibility that host-mediated responses contribute to increased colonization for other drugs. In the case of terfenadine, our data indicate that they likely play a minor role. However, we discuss the potential contribution of the host in the discussion (lines 436 - 444).

Action taken: For terfenadine, we included a comprehensive analysis of potential drug effects on the host that could result in increased *S. Tm* colonization (**ED Fig. 11**) and have discussed the influence of the host response on drug-induced microbiome changes in the discussion.

Minor comments:

Comment 1.5

Line 248-264: The use of PICRUST2 on such a small, define community to demonstrate that microbial metabolism changes when the community structure changes is not appropriate as it generates a very predictable outcome. This approach also appears to have generated little detailed insights. I would suggest removing this experiment. The subsequent modeling is far more powerful.

We agree with the reviewer that the contributions of the PICRUST2 analyses are rather minor.

Action taken: We have removed this section and the corresponding figure panels from the manuscript accordingly.

Referee #2

In this paper, the authors thoroughly investigate the impact of non-antibiotic drugs on commensal microbe community and consequent influence on colonization resistance against enteric pathogens. The authors first examined the effects of 1,200 FDA-approved drugs on the growth of gamma-proteobacteria and commensal bacteria. They find that Proteobacteria strains exhibit sensitivity to fewer drugs than commensals, and the sensitivity of Proteobacteria is evident only at higher concentrations of drugs. Then, the authors examined the effects of selected 67 drugs on commensal community models consisting of 20/21 strains. Through in vitro and in vivo experiments, they demonstrate that treatment with certain non-antibiotic drugs compromises the commensal community-mediated colonization resistance against Salmonella. The influence of non-antibiotic drugs is shown to reduce the overall bacterial amounts, alter the composition, and/or presumably modify the function of the commensal community, facilitating the colonization and growth of Salmonella. The authors suggested that altering the metabolic potential of the commensal population might create an open niche for Salmonella. The impact of non-antibiotic drugs extended beyond Salmonella, potentially promoting colonization by other Proteobacteria, indicating the generalizability of the authors' findings. While certain disparities between in vitro and in vivo results, along with varied outcomes in the context of different commensal communities or host conditions, hinder the establishment of universal rules, I recognize the work as valuable and well-conducted. This paper offers highly valuable knowledge that must be considered in clinical practice, as well as fundamental insights into the complex interactions of the gut microbiota members. The importance of this research could be further enhanced by addressing the following points:

We appreciate the reviewer's support and helpful comments!

Comment 2.1

There are some concerns regarding the clarity of certain figure panels and their legends. Several of the figure panels are quite intricate, making them challenging to decipher. Indeed, Figures 1b, 2d, 3a, 3b, 4a, 4b, 4d, 5c, 6a, 6b, 7bcd, Supp Figures 1a, Supp 2a,

2e, 2f, 2l, Supp 3b, Supp 5a, Supp 7c, 7d are difficult to comprehend without a deep knowledge of informatics and perhaps mathematics. For instance, in lines 286-287, the authors assert that "E. coli, a member of the Enterobacteriaceae family with metabolic characteristics similar to S. Tm (Figure 4a)." However, grasping this claim proves challenging as Figure 4a does not explicitly illustrate the basic metabolic characteristics of each individual strain; instead, the authors present the metabolic competition index. I would recommend the authors to make the figures a bit more intuitively understandable.

In addition, figure legends are often too brief and may not be sufficient to assist readers in understanding the complexities of the figures. To make the findings more accessible to a broader audience, consider providing more detailed explanations and context for the key elements in each figure. I understand that space constraints can be a concern, but a more comprehensive legend would undoubtedly improve the overall readability of the paper.

We agree with the reviewer that some of our figures are complex, and the lack of sufficient explanation can make them difficult to interpret.

Action taken: In the revised manuscript, we have expanded all figure legends and referenced the relevant sections of the methods where more detailed explanations can be found. We believe that these changes should clarify the figures and their interpretation for the reader.

Comment 2.2

The presented data compellingly unveil heightened resistance to non-antibiotic drugs in Proteobacteria pathogens—a discovery of great intrigue. Nonetheless, the mechanisms underlying the effects of these drugs on commensals and pathogens employed in this study remain elusive. A comprehensive examination of the shifts in transcriptome landscape induced by drugs, such as terfenadine, clomiphene, and floxuridine, in both the pathogenic and commensal strains would provide invaluable mechanistic insights. Furthermore, by harnessing the genetic manipulability of Salmonella or other Proteobacteria strains, delving into the genes associated with resistance to non-antibiotic

drugs via a mutant library could prove instrumental in elucidating the intricate molecular pathways involved.

We are pleased to read that the reviewer finds this result as intriguing as we do, and we agree that further examination of the functional underpinnings of the differences in the response of commensals and pathogens would strengthen the manuscript. Therefore, we performed multiple additional experiments and analyses:

First, we evaluated whether differences in the genetic repertoire of pathogens and commensals could explain their differential response to drugs. We looked for the presence of genes involved in antimicrobial resistance, stress responses or efflux using the available genomes of the species tested. We found that pathogens encode a larger fraction of genes involved in efflux processes and tend to have more genes involved in resistance to antibiotics or stress responses. We posit that the expanded genetic repertoire for resistance and detoxification processes in pathogenic species constitute adaptations that allow them to survive and thrive in the hostile environments they encounter, such as those produced by the host immune system upon infection, conditions encountered less often by commensal microbes (**ED Fig. 3e,f**, lines 116 - 120).

Second, we examined the importance of drug efflux on the microbial response to drugs. After our initial submission, a comprehensive study on the transcriptional responses of gut bacteria to drugs was published (Ricaurte et al., 2024). Fortunately, the authors looked into 12 drugs and 13 species that we included in **Fig. 1a** of the revised manuscript. They showed that non-antibiotic drugs can induce efflux pumps, particularly in *Bacteroides/Phocaeicola* species. Four of the 19 drugs tested upregulated the expression of the RND-type efflux pump in *Phocaeicola vulgatus* (BVU_1673-1675) and one, simvastatin, induced BVU_0237-0244 (see Figure R1 below). Based on this information, we investigated whether the absence of these efflux pumps abrogates resistance to non-antibiotics. Per the reviewer's suggestion, we leveraged the genetic manipulability of *P. vulgatus* and knocked out the BVU_1672-1675 operon (transcriptional regulator + efflux pump). As a comparison, we knocked out the RND-type efflux pump key component TolC in *S. Tm*. We then assessed drug susceptibility of wild-type and mutant strains of both species across ~1270 compounds using an updated version of the

Prestwick Chemical Library (see *Methods*). Knocking out the RND-type efflux pump in *P. vulgatus* only resulted in changes in drug sensitivity for 4 compounds. In contrast, the *S. Tm toIC* mutant was sensitized against a large number of antibiotics and non-antibiotics. This exciting finding suggests that *Gammaproteobacteria* pathogens such as *S. Tm*, possess efflux pumps that are well adapted for the extrusion of xenobiotics, making them more resilient and resistant to drug treatments (**ED Fig. 3g**, lines 120 - 126), which is not the case for *P. vulgatus* (and perhaps other commensals more generally).

Figure R1: In *P. vulgatus*, human-targeted drugs at a concentration of 20 μ M induce the expression of genes encoding RND-type efflux pumps. The y-axis shows the log2 fold change in expression levels for each gene within two operons, comparing drug-treated samples to untreated controls. While genes in the operon BVU_0237-0244 were upregulated exclusively by simvastatin, genes in the operon BVU_1672-1675 were induced by multiple human-targeted drugs. These results represent a reanalysis of previously published data (Ricaurte et al., 2024) to justify our selection of BVU_1672-1675 (transcriptional regulator + pump) for further study.

Action taken: We have included the results of the assessment of the genes involved in antimicrobial resistance and the changes in drug sensitivity in *S. Tm* and *P. vulgaris* efflux mutants in the results as possible contributors to the differences observed between pathogens and commensals.

Comment 2.3

In this study, the authors investigated the impacts of non-antibiotic drugs on both pathogen and commensal strains primarily by investigating their growth or abundance. However, the authors have yet to explore the functional consequences of these strains. In this regard, transcriptome analyses of pathogen and commensal strains would also be highly valuable.

Following the reviewer's suggestion, we conducted a transcriptomic analysis of both *S. Tm* and Com20 after drug treatment in an adapted invasion assay. In this assay, we used transwell plates to physically separate the community from *S. Tm*, while allowing them to share the same growth medium (see *Methods*). For our analysis, we selected four drugs that interfered with colonisation resistance: two that primarily reduced community biomass (clomiphene and terfenadine), and two that mainly altered the overall community composition but had no strong effects on biomass (floxuridine and simvastatin). In the case of *S. Tm*, we also evaluated the transcriptome in pure culture in mGAM.

For *S. Tm*, we observed distinct transcriptomic profiles between the experimental groups. The two biomass-reducing drugs showed profiles similar to *S. Tm* in the absence of a microbial community. These data suggest that in the case of drugs that reduce biomass, *S. Tm* behaves as if no competitors were present. However, for the high-biomass drugs, we observed two distinct responses, which also differ from the *S. Tm* response to an untreated Com20, suggesting that the pathogen employs different functional strategies depending on the community it encounters (**Fig. 2e,f, ED Fig. 8a**). Thus, this response ultimately depends on the effect of the drug on the community.

For the high-biomass drugs, we analyzed the metatranscriptomes and mapped the individual responses of each community member (**Fig. 2g, ED Fig. 8b**). Notably, both

drugs elicited distinct responses of the individual members of Com20, which are also different from their transcriptional profiles in the untreated community. These findings are consistent with the corresponding differences in pathogen response. These profiles likely reflect mixed responses due to both direct drug effects and community-mediated effects on individual Com20 members.

These drug-specific responses highlight the complexity of the molecular interactions involved and support the notion that the precise underpinnings will vary depending on the drug. Thus, further investigation of these effects on a case-by-case basis for each drug will be essential in the future, and is well beyond the scope of this paper.

Action taken: In the revised manuscript, we included transcriptomic and metatranscriptomic analyses of *S. Tm* and Com20 for four drugs. To report these results, we added new figure panels (**Fig. 2e,f,g**) as well as a new extended data figure (**ED Fig. 8**). We discuss these new results on lines 249- 288.

Comment 2.4

Are there any common characteristics in the chemical structure of drugs that facilitate the influence on commensal strains while exerting less influence on pathogens? Understanding the specific interactions between drug chemical structures and commensal microbiota, as opposed to pathogens, could provide crucial insights into designing drugs that minimize impact on commensal organisms. Exploring these nuances in drug interactions holds significant implications for developing targeted and more tailored therapeutic strategies.

We agree with the reviewer that the properties of the compounds could provide valuable insights and represent an important first step toward developing tailored therapeutic strategies. Per the reviewer's suggestion, we examined the chemical properties of the non-antibiotics in the Prestwick library that inhibited at least five commensal species in our assay. We assessed the molecular weight, hydrophobicity (measured by XLogP), polar surface area (measured by TPSA), complexity, and volume of the compounds according to the number of *Gammaproteobacteria* pathogens or

commensals inhibited. This analysis showed that drugs that are more hydrophobic, have higher molecular weight, and are larger in volume tend to inhibit more commensals. Conversely, smaller, polar, hydrophilic drugs tend to inhibit more pathogens.

Action taken: We have included the results of these analyses in the text (lines 109 - 111) and in **ED Fig. 2b,c**.

Comment 2.5

To establish clinical relevance, it is imperative to verify the impact of non-antibiotic drugs within the complex human gut microbiota context *in vivo*. This can be achieved, for instance, by employing germ-free mice colonized with human stool samples containing Proteobacteria.

We agree with the reviewer that our results are based on models that, by their nature, have limitations. To address this concern and connect our findings with clinical scenarios, we evaluated the effect of the antihistamine terfenadine on *S. Tm* colonization in germ-free mice colonized with feces from a human donor that contains members of the family *Enterobacteriaceae*. Consistent with the results in our original submission, we found that drug-treated mice had higher pathogen counts 1 and 4 d.p.i. in both the feces and cecum compared to DMSO-treated controls. This finding demonstrates that our observations extend to the context of a complex human gut microbiome *in vivo*.

Action taken: We have added these results to the manuscript (lines 359 - 375) and in **Fig. 4f,g,h** and **ED Fig. 10h**.

Comment 2.6

Regarding difference and overlap of metabolic capabilities of 20/21 commensal communities and *Salmonella*, the authors used PICRUSt2 to predict pathway abundance. I appreciate the utility of PICRUSt2 for rough predictive functional analysis at the species level; however, it cannot account for differences between strains. Metabolic pathways are known to vary significantly between strains, and relying solely on PICRUSt analysis may be insufficient. I would recommend the author to consider validating the analysis through *in vitro* experiments. Alternatively, conducting whole genome sequencing of each strain

followed by transcriptome analysis could offer a comprehensive understanding of the metabolic activities within the bacterial communities. Such efforts would significantly strengthen the reliability and impact of the conclusions.

As reviewer #1 also pointed out, the contributions of the PICRUSt2 analyses are relatively minor (see comment 1.5). We agree with this assessment.

Action taken: We removed the PICRUSt2 analyses from our revised manuscript. In its place, we performed a series of transcriptomic and metatranscriptomic analyses to examine the expression profiles of *S. Tm* and Com20 following drug treatment; see Comment 2.3. These analyses were conducted using a transwell assay, which kept the pathogen and the community physically separated while sharing the same medium. This setup allowed us to obtain a metatranscriptomic profiling of both pathogen and commensals by separating Com20 members from *S. Tm*, thus preventing the predominance of *S. Tm* from affecting the measurement of Com20 cells. At the same time, it ensured an adequate amount of *S. Tm* cells in all treatments. For more details on these findings, please refer to our response to comment 2.3 above.

Comment 2.7

Supplementary Fig 4 should examine and display the Com21 and Com20 communities side by side.

Action taken: We agree with the reviewer. We now display the results of the growth of the pathogen on drug-treated Com20 and Com21 side-by-side (**ED. Fig. 9b**).

Referee #3

The manuscript aims to investigate the effect of non-antibiotic drugs on the microbiota and pathogenic gammaproteobacteria. The authors find that pathogens are, in general, more resistant than commensals to non-antibiotic drugs, and that some of these drugs reduce colonization resistance to the pathogen *Salmonella*. By testing synthetic and human stool-derived communities, as well gnotobiotic mice and SPF mice, they find that non-antibiotic

drugs favored *Salmonella* expansion in vitro, and at day 1 post-infection.

The study has the merit of exploring the role of non-antibiotic drugs in altering the gut microbiome composition and promoting pathogen expansion, likely by reducing colonization resistance. These findings are important because there is little known about the possible consequences of non-antibiotic drugs in altering colonization resistance to pathogens in general. The finding that gammaproteobacteria are, in general, more resistant to non-antibiotic drugs also has important implications. The manuscript is also generally well-conducted and well-written.

We appreciate the reviewer's support of our manuscript and their helpful comments and suggestions!

Comment 3.1

My primary concern about the study is that there is no mechanism for the presented observations. The reason why gammaproteobacteria are more resistant to most non-antibiotic drugs is not investigated. Similarly, the effects of non-antibiotic drugs on the gut microbiota are unclear, and not all the in vitro phenotypes are recapitulated in vivo. I suggest that the authors choose one of the drugs and follow up their findings with mechanistic studies, at the very minimum, to identify which specific changes in the gut microbiota would enhance pathogen colonization.

Similar concerns were raised by reviewer #1 in comment 1.2 and by reviewer #2 in comment 2.2. We would like to first clarify that we tried to refrain from using the broad term mechanism, in favour of focusing on the distinctions between molecular interactions, cellular processes, and ecological dynamics. In the context of our study, a drug can interact with a bacterial intracellular structure at a molecular level, resulting in a series of altered cellular processes that, depending on the severity of the alteration, can affect the growth of a bacterium or lead to its death, thereby changing the ecological dynamics of the microbial community. In our revision, we have tried to address the factors driving our observations at each of these scales.

We agree with the reviewer that our initial manuscript lacked an explanation of why *Gammaproteobacteria* pathogens are less sensitive to non-antibiotics compared to gut commensals. A detailed explanation of the analyses we performed can be found in comment 2.2. Briefly, we performed two new analyses and sets of experiments. First, we used whole genome sequences of the tested pathogens and commensals to explore their repertoire of genes involved in antimicrobial resistance, stress responses, and efflux. We found that a larger fraction of the genes in pathogens fall within those categories compared to commensals (**ED Fig. 3e,f**). Second, we assessed the changes in drug resistance in *S. Tm* and *P. vulgatus* mutants lacking a resistance nodulation-cell division (RND) multidrug efflux pump. We found that the pathogen (*S. Tm*) mutant became sensitive to 35 drugs to which the wild-type strain is resistant. In contrast, the sensitivity of the commensal (*P. vulgatus*) only increased by 4 drugs due to knocking out the efflux pump (**ED Fig. 3g**). We posit that the importance of antimicrobial resistance and stress-related genes differs between gut bacteria and *Gammaproteobacteria* pathogens; the detoxification response of the former is less effective and the removal of one such mode has little impact on overall susceptibility. Conversely, the adaptations of pathogens to the harsh environment generated by the host's immune system during infection makes them more resistant to non-antibiotics but also more reliant on detoxification and resistance processes.

As the reviewer suggested, we expanded our analyses *in vivo* and *in vitro* to explain which changes in the gut microbiota are linked to increased pathogen growth. We included analyses of changes in microbial diversity in drug-treated communities *in vitro* and their association with drug-induced changes in community biomass (lines 184 - 247). We also investigated the transcriptional response of both the pathogen and community upon treatment. Finally, we used a humanized mouse model to complement our initial experiments in gnotobiotic and SPF mice (see comment 3.2 below). Our *in vitro* observations are largely recapitulated *in vivo* in all our models for drugs such as clotrimazole, chlorpromazine, clomiphene, and terfenadine. As the reviewer points out, for zafirlukast, the effects we observed *in vitro* in Com20 did not extend to mice. This discrepancy is possibly due to community-specific factors or host contributions, which we touch upon in the result section and the discussion, as it points to an additional mode by

which the community is disrupted: indirect effects mediated by changes in host physiology (lines 376 - 380 & line 420 & lines 436 - 440).

In regard to mechanistic insights, we assume that the reviewer is suggesting we identify an individual drug-protein interaction that leads to alteration of a specific metabolic pathway. While we appreciate the motivation, this goal is well beyond the scope of this paper for two reasons. First, our focus is to assess the effects of non-antibiotics on the microbiome and their impact on colonization resistance across a wide range of drugs using multiple model systems. Second, our results indicate that there is no single molecular interaction or cellular process that explains the drug-induced loss of colonization resistance across all drugs or diverse microbiome compositions. Rather, all possible alterations to the individual members of the community or the host ultimately converge to shifts in the competition between commensals and pathogens. These shifts can impact the community's ability to effectively respond to the pathogen by altering the total community biomass, by changing the composition of the overall microbial community, and/or by targeting direct niche competitors of the pathogen. We expand on these ecological dynamics in the discussion (lines 401 - 417).

Action taken: We have included the new analyses of antimicrobial genes and the knockout strains of *S. Tm* and *P. vulgatus* at lines 116 - 131. We expanded our analyses of the microbial composition and diversity at lines 184 - 247. We included the results of the new humanized mouse experiments at lines 359 - 381. Of note, we conducted follow-up investigations on the drug terfenadine across all model systems used in this study.

Comment 3.2

Related to understanding possible mechanisms, the mouse studies in Figure 7 should be expanded to include an analysis of the host responses and possibly additional timepoints. The experiments here show a disruption of colonization resistance at early stages post-infection, but again the mechanism is not completely elucidated. Even though no inflammation was observed (which is expected at day 1 post-infection), it is possible that other more subtle changes in gene expression may have occurred. For example, induction

of iNos, or increased epithelial oxygenation. Bottom line, it needs to be shown whether administration of the non-antibiotic drugs in vivo affects the host in addition to the microbiota, before and after infection.

We are largely in agreement with the reviewer, although we note that comprehensive analyses of host responses are quite labour intensive. Because of this, we focused on a single drug and microbiota composition. Specifically, we used a humanized mouse model, which is arguably a closer representation of conditions in human patients. We investigated the antihistamine drug terfenadine, since it resulted in the highest pathogen growth in the Com20-colonized and SPF mouse models.

After drug administration and before infection, we sacrificed five humanized mice per group. From these mice we collected cecal tissue, and prepared sections for immunohistochemistry to assess whether the treatment itself induced any host responses at the site of *S. Tm* colonization and infection. The markers we examined included *immune cell infiltration* (CD8, CD11b, CD11c, F4/80), *apoptosis* (Cl. Casp3), *proliferation* (Ki67), *inflammation* (RelA), *hypoxic adaptation* (HIF-1 α), and *endothelial function* (CD31). We observed no significant influence of terfenadine on these host markers (**ED Fig. 11**), suggesting that the disruption of colonization resistance was likely microbiome mediated.

After infection, terfenadine-treated mice showed higher *S. Tm* colonization in the cecum and higher fecal pathogen loads at 1 and 4 d.p.i., compared to DMSO-treated controls (**Fig. 4f, g**). Inflammation onset occurred earlier in the treated mice, as evidenced by an earlier increase in fecal lipocalin-2 levels (**Fig. 4h**). At 4 d.p.i., pathoscores in HE-stained cecal sections were higher (**ED Fig. 11a**), as were the numbers of CD4 and CD8 positive infiltrated immune cells (**ED Fig. 11**). In summary, terfenadine-treated mice exhibited faster onset and progression of *S. Tm* infection compared to the DMSO-treated controls.

Action taken: We conducted a thorough analysis of host responses before and after infection in terfenadine-treated mice. The results are now included in **Fig. 4f,g,h**, and **ED Fig. 11** and the modified text (lines 371 - 375).

Comment 3.3

The animal experiments in Fig. 7 show that the non-antibiotic drug treatment disrupts colonization resistance and results in a higher colonization level of Salmonella at day 1 post-infection. A question remains whether the non-antibiotic drugs also increase the severity of infection. This would require later time points.

We agree with the reviewer that it is important to determine whether higher pathogen loads at 1 d.p.i. are associated with more severe infection at later time points. In the case of terfenadine treatment in the humanized mouse model, higher *S. Tm* loads at 1 d.p.i. indeed led to faster disease progression. This is evidenced by an earlier onset of inflammation (as indicated by the rise in fecal lipocalin-2, **Fig. 4h**), a higher pathoscore at 4 d.p.i. (**ED Fig. 11a**), and an increased number of infiltrated CD4 and CD8 positive immune cells in cecal tissue sections compared to controls (**ED Fig. 11f,g**).

Action taken: The new results have been incorporated into **Fig. 4f, g, h, and ED Fig. 11**, and the text has been updated accordingly (lines 359 - 381).

Referee #4:

This is a fascinating study by Griebshammer and colleagues which builds on previous work from the same group showing that many non-antibiotic drugs can have inhibitory effects on human gut commensal bacteria. Here they show that some of these drugs can increase the ability of *Salmonella Typhimurium* and other Gammaproteobacteria members to invade model microbial communities, ex-vivo microbial communities and in mouse models. These are important and original findings with wide reaching implications. I have some questions regarding the methodology and several suggestions that I think would help strengthen the findings. Please see below.

We thank the reviewer for their support and helpful comments!

Comment 4.1

Line 91 and Fig. S1. 'these pathogens showed different sensitivity profiles compared to a panel of commensal gut bacteria'. To help determine how expected or unexpected this is it would be useful to show the genetic distance between the species tested. I.e. would the large difference in sensitivity profiles of the Gammaproteobacteria compared to other species be a consequence from their large evolutionary distance to the other species tested? There appear to be other clusters of similarity e.g. the Bacteriodes spp. At a minimum showing the phylogenetic tree of the bacteria tested would help, but a more rigorous approach could be to plot genetic distance against drug sensitivity distance to show if these correlate and if Gammaproteobacteria are outliers or not. The comparison of gram positive to gram negative could also be informative in explaining the sensitivity profile of Gammaproteobacteria compared to other species.

As per the reviewer's suggestion, we assessed the association between the response of microbes to antibacterials and non-antibacterials in an evolutionary context. We used matrices of Euclidean distances calculated from bacterial AUCs upon drug treatment to measure the similarity of responses between species. The phylogenetic distance between species was measured using the cophenetic distance; this distance was calculated from a phylogenetic tree reconstructed using a multilocus alignment obtained from whole bacterial genomes.

Overall, we found that differences in the responses of pathogens and gut commensals to antibacterial and non-antibacterial exposure are modestly explained by phylogeny alone. First, we generated principal component plots of the response profile to antibacterials and non-antibacterials. For both groups of drugs, the overall response of pathogens is distinguishable from that of commensals, however, there is large variation in the response of commensals, with some commensals showing a similar response profile to *Gammaproteobacteria* pathogens. Moreover, when coloured by phylum, there is some overlap between clades (**ED Fig. 3a,b**). Next, we tested the association between phylogeny and response profile using the Mantel test to correlate both AUC and cophenetic distance matrices. In the case of both antibacterials and non-antibacterials, we found a positive but weak correlation, that was only significant for antibacterials

(Mantel's correlation for antibacterials: 0.08, p value = 0.04; Mantel's correlation for non-antibacterials: 0.03, p value = 0.18). We further assessed the change in the strength of the association across evolutionary distances using a phylocorrelogram, which shows the change in the average correlation in the response to the drugs between species as the phylogenetic distance increases. In both cases, we observed higher correlations at smaller phylogenetic distances, which indicates that closely related species tend to respond similarly to the evaluated drugs. However, even at very short distances, the overall correlations were moderate (**ED Fig. 3c,d**).

In summary, these results indicate that factors other than phylogenetic relatedness fully account for the distinct responses of pathogens and gut commensals to antibacterial and non-antibacterial treatments.

Action taken: We have added these analyses and figures to the manuscript (**ED Fig. 3a-d**, lines 111 - 115).

Comment 4.2

Fig. 1 please define 'active'. I'm inferring that it is the number of drugs which were inhibitory to the species tested.

Action taken: Indeed, we refer to the number of drugs with an inhibitory effect. We have added this information to the legend.

Comment 4.3

Line 111: 'while only 14% of the pathogens were affected'. This sentence is worded incorrectly. There are only 5 pathogens, so you can't get 14% of them. I believe you mean something to the effect of '14% of pathogen-drug combinations were inhibitory'. Please rephrase for correctness.

Action taken: The reviewer is correct. However, the sentence was removed from the new manuscript after restructuring the result section.

Comment 4.4

Line 119. 'a synthetic community comprising the 20 gut commensals'. Fig. 1 legend says 19 commensals + the pathogen. Are there 20 commensals + pathogen, or 19 commensals + pathogen?

Action taken: It is 20 commensals plus pathogens. We have corrected the text accordingly.

Comment 4.5

Line 122: It's unclear from Supplementary Figure 2a how the 61.3% overlap of metabolic pathways was calculated. It may be cleared to add the number of total metabolic pathways and number of pathways present Com20.

Action taken: Following the reviewer's suggestion, we have revised this section to specify the number of MetaCyc metabolic pathways detected in both Com20 and the HMP samples (lines 138 - 141).

Comment 4.6

Line 124. 'and robustly colonized the gastrointestinal tract of mice for up to 57 days'. The figure shows that the mice are robustly colonized in the sense that the biological repeats show similar relative abundances. However, it doesn't look terribly robust temporally – many species appear to be lost or at undetectable levels by 6 days?

The reviewer is right in pointing out that there is an initial adaptation phase of the community to the host, which is expected. Indeed, a compositional change between days 2 and 6 is clear in the figure. In contrast, the changes between days 6 and 57 are smaller. We agree with the reviewer that the bar plot we used to show the composition of the community *in vitro* and *in vivo* does not allow the reader to easily visualise the levels of low abundance species. This makes it hard to appreciate the actual changes across the three time points and which species are lost in the transition to a host-associated environment, which we have now addressed.

Action taken: We made a new figure that makes it easier to appreciate the changes in the relative abundance of each species and that clarifies which species are not detected at a given time point (**Fig. 1b**), as well as making it clear that the robust colonization comes after a rearrangement of the community (line 144).

Comment 4.7

Line 130 and Methods. Is the drug washed out at the 24h timepoint before *S. Tm* challenge? If I have understood the protocol correctly then the drug is present, but there is a 10-fold dilution into fresh mGAM during the invasion step. This is important to make explicitly clear, since the presence of the drug may also be directly affecting the *S. Tm* growth, even if diluted. The authors note that they exclude drugs that directly inhibited *S. Tm* growth, I'm assuming this is because the drug remained present. What was the threshold used to determine *S. Tm* inhibition here- IC25? If so, this could mean some modestly *S. Tm* inhibiting drugs were still present, which may affect the results. Can this be ruled out by a control which includes a wash step?

The reviewer is correct. The drug is expected to be diluted by 10-fold or more due to factors such as drug metabolism, bioaccumulation, and chemical instability. We do not consider this a shortcoming of the method since one would not expect a complete washout of drugs from the gastrointestinal tract of patients or mice after treatment. Importantly, non-antibiotic drugs are generally less potent than antibiotics, which typically require stronger dilutions for their effects to disappear.

Per the reviewer's suggestion, we performed an experiment that included a washing step after treatment, before the inclusion of the pathogen. For most drugs, washing did not lead to an increase in *S. Tm* levels (**ED Fig. 4f**). Surprisingly, washing led to a decrease in *S. Tm* growth in clomiphene- or sertindole-treated Com20, which suggests that the pathogen may be using compounds liberated by dead commensal bacteria as nutrients, compounds that are removed by the washing step. From an operational perspective, the washing step significantly reduced the throughput of our screen since it requires centrifugation, which is challenging to perform with plates inside the anaerobic chambers.

Action taken: We present and discuss the results of the washing experiments on lines 153 - 155, in **ED Fig. 4f** and in the Methods section (lines 1211 - 1226). In addition, we included additional sentences in the methods section for clarity:

- Line 1084: We excluded drugs that directly inhibited *S. Tm* growth with IC₂₅ values < 5 μ M (nalidixic acid and norfloxacin, **Supplementary table 2**).
- Line 1191: Of note, this protocol resulted in the transfer of residual amounts of the drug, up to 10% of the original concentration.

Comment 4.8

Fig. 2b. OD could be a relative inaccurate measure of biomass in the complex community, especially if cidal drugs are used and when low or high ODs are reached. Some controls to get cell counts per ml, for example with flow cytometry, would be helpful here to verify that OD is a reasonable reporter of cells/ml.

The reviewer is right in pointing out that, under certain conditions, the optical density of a community fails to accurately measure its biomass. However, we think that our use of this measure is adequate for methodological and practical reasons. First, OD is a reasonable, even if imperfect, approximation of biomass. In our previous work, we found a significant positive correlation between the optical density of a community and the levels of a DNA spike-in of a particular species using 96 *in vitro* communities of different compositions (Celis et al., 2022). Similarly, we found that results obtained from analyses based on normalizing 16S rRNA abundances by OD also held true when the abundances were standardized using robust upper-quartile normalization (Garcia-Santamarina et al., 2024). Second, alternative methods to quantify community biomass can also lead to inaccurate measurements. For example, CFU quantification is highly variable due to factors such as clumping or cell division, which can result in a 2-fold difference between measurements. Similarly, determining the true gating criteria for identifying cells using FACS can introduce variability. Third, measuring the OD of thousands of samples is simple and fast, which is a critical requirement for the high-throughput of our assay.

Action taken: Based on our previous results and the technical considerations of our assay, we are confident in the results obtained with OD and do not consider it necessary to perform additional validations.

Comment 4.9

line 142 'we tested 52 of the 67 drugs', line 296 'We challenged Com21 with 48 out of 52 drugs', line 318 '12 out of 52 drugs' etc. In general, I had a very difficult time working out the numbers of drugs used for different experiments and the inclusion/ exclusion criteria. A different number seems to be used in almost every experiment. The methods do not adequately explain the selection criteria: e.g. Line 893' we selected 30 drugs representing diverse inhibition spectra across Com20 members. Additionally, we incorporated 25 clinically relevant drugs, resulting in a final selection of 52 drugs' These numbers don't add up – I assume that this is due to exclusions mentioned later, but it is unclear. Drugs were taken out for S. Tm, are later introduced in other assays, so mentioning that they are still incorporated in other assays would be useful. Also, what do the authors mean by 25 clinically relevant drugs, and what are they –all the 1200 drugs originally screened are clinically relevant? A supplementary flow diagram or similar, with clear inclusion/exclusion criteria might help readers to understand.

We agree with the reviewer that a better explanation of the drugs used in each experiment would clarify the manuscript. We performed additional experiments to ensure that the same compounds were evaluated in both Com20 and Com21. Regarding "clinical relevance", we meant drugs that are currently used in clinical practice but that are not part of the Prestwick library.

Action taken: We rephrased the explanation of how the list of 52 drugs (now 53) was obtained (lines 1072 - 1088). In addition, we added a new figure in which we specify the drugs tested in each assay, whether they were tested *in vivo* or *in vitro*, and the criteria for their inclusion in that particular assay (**ED Fig. 1a**).

Comment 4.10

Similarly, to the above, the number of bacteria tested varies by experiment and the inclusion criteria are unclear. A clear set of tables with the species present for each experiment would be helpful.

Action taken: We agree with the reviewer, and in line with the comment above, we now include the community used to evaluate each of the sets of drugs (i.e., Com20, Com21, human donor-derived) and whether the experiment was done *in vitro* or in a murine model (**ED Fig. 1a**).

Comment 4.11

Line 201: 'we manually generated four communities whose composition resembled that of drug-treated communities'. If I have understood the protocol correctly then then a 10-fold dilution of the drug remains present during the S Tm challenge in the drug-treated assay. Some inhibited species may remain inhibited during the 4.5 h S. Tm challenge under these conditions, whereas in the drug-mimicking communities they do not. Could this explain the mimicking conditions that did not fully phenocopy treatment? A drug wash step should answer this.

This is a point similar to comment 4.7. As the reviewer suggested, we performed the S. Tm invasion assay with an additional washing step. As described above and shown in **ED Fig. 4f**, growth of the pathogen in communities with fresh medium after drug treatment was largely unchanged. Moreover, in certain cases, the washing step led to decreased growth of the pathogen, suggesting that certain compounds, likely coming from dead bacteria, are used by the pathogen as nutrients. Therefore, we do not think that residual drug in the medium can explain the mimicking conditions that did not fully phenocopy treatment.

Comment 4.12

Line 225 please specify 16S rRNA amplicon sequencing

Action taken: We have revised the text accordingly. The specific sentence has been removed in the updated manuscript. However, we have ensured that throughout the revised manuscript, we clearly specify that we are referring to 16S rRNA sequencing.

Comment 4.13

Line 226 mentions 'S. Tm-favoring communities' then references figure 3a. but the legend of figure 3a. says 'S. Tm-favoring treatments'. Since both treated and treatment-mimicking communities are used in the study this need clarification.

Action taken: We agree with the reviewer that this is confusing. We have changed the text in line 1853 to 'communities resulting from S. Tm-favoring treatments'.

Comment 4.14

Line 254: Please expand on what is meant by "metagenome potential"

Action taken: This line is part of a section of the text that was removed after revisions due to comments from other reviewers (see comments 1.5 and 2.6).

Comment 4.15

Line 255 a brief explanation of what is meant by 'functional capacity' would be useful.

Action taken: This line is part of a section of the text that was removed after revisions due to comments from other reviewers (see comments 1.5 and 2.6).

Comment 4.16

Line 324-331: "The effect of certain drugs was pathogen-specific.....zafirlukast restricted the growth of S. flexneri 24570, Y. pseudotuberculosis YPIII, and Y. enterocolitica WA-314 but favored the growth of E. coli CFT073". Does this pathogen-specific effect relate to the community-disrupting effect, or to the sensitivity of the pathobionts/pathogens to said drug? Do these drugs (e.g. clomiphene, chlorpromazine, zafirlukast) have an inhibiting effect on the pathobionts/pathogens in pure cultures?

This is a very interesting point, which falls within the broad concern raised by the reviewer about the potential effect of drug residues on the pathogen. We think it is unlikely that the direct effect of the drugs is a major contributor to the variation in pathogen growth we observed across *Gammaproteobacteria* species for several reasons. First, the expected residual concentration after the 10-fold dilution following the inclusion of the pathogen in the invasion assay is below the IC₂₅ of the pathogens in most cases. Even when the IC₂₅ of the pathogens was low, such as in the case of floxuridine (**Supplementary table 2**, Figure R2 below), community treatment at the highest concentrations of the compound led to an increase in pathogen growth (**Supplementary table 5**), despite the potentially disrupting effect of any residual drug. Second, as explained in comment 4.7, including a washing step had either no effect or reduced the growth of *S. Tm*; we expect something similar to occur with the other pathogens. Third, the community context plays a role in the ability of the pathogens to proliferate in the absence of drug disruptions, as evidenced by the variation in growth across the *Gammaproteobacteria* species in 19-member communities (**ED Fig. 7a**). This variation in growth might be responsible for the pathogen-specific effect of certain drugs. We speculate that some of the pathogens are better at utilizing certain commensal-derived metabolites or the open niches that become available after drug disruption, explaining the observed variation.

Figure R2; IC₂₅ values, plotted from **Supplementary table 2** to illustrate the direct effect of these drugs on pathogens. Please note that the final drug concentration in the in vitro challenge assay is at least 10-fold lower at the time point when the pathogen is added.

Action taken: We added to the methods a mention of our reasoning behind not carrying an additional experiment such as the one performed for S. Tm.

Comment 4.17

Line 400-402: “changes in the abundance of 4 and 5 species.....Similarly, 31, 2 and 1 amplicon sequencing variants were significantly different...” Which species and sequencing variants were identified? Are these species/sequencing variants unique to their respective mice model?

The reviewer is correct to point out that these questions could be of interest to certain readers, although due to the space restrictions, we chose not to include the names of the species on the main text, but list them in the supplement (**Supplementary tables 10, 11 and 12**).

Regarding the uniqueness of an ASV to a given model, we expect each set of ASVs to be distinct in each model given that the microbiota of each is, by design, very different. Com20-colonized mice are colonized with type strains of prevalent and abundant members of the human microbiome; SPF mice contain the native microbiota of specific pathogen free mice; Hum mice are germ-free mice colonized with a community derived from a human donor.

Action taken: We added tables containing the names, effect size of the linear models and adjusted p values of the species/ASVs evaluated in all mouse models (**Supplementary tables 10, 11 and 12**).

Comment 4.18

Line 837-838: “monoculture were mixed in equal ratios”. Was this based on volume, OD or cell number?

Action taken: We changed the text to clarify that cultures were mixed at equal OD values (line 994).

Comment 4.19

Line 948. Community assembly: in previous papers by the same group the community has been assembled based on equal CFU/ml rather than OD, presumably to obtain more equal distribution of each species. Why was this not done here? Are the Com20 members in stationary phase or exponential phase when added to the drug plates? The drugs may have quite different effects on the bacteria depending on growth phase.

The reviewer is correct to point out that different optical densities (OD) may not correspond precisely to the same CFU counts across the various species tested. We have tested both approaches for assembling the community based on equal CFUs and OD. In this study, we opted to use OD for assembly because measuring this value is more straightforward and time is a critical factor when assembling a community from multiple species so as to not result in outgrowth that disrupts the initial abundances. In our experience, small variations in the initial volumes do not significantly affect the community composition at the relevant time points for our study, and we did not observe significant differences in community structure when assembled based on OD or CFU in the context of drug treatments.

It is also true that drug effects can vary depending on the bacterial growth phase within a community. In our study, the Com20 is not in the stationary phase, as the starting OD is 0.5 when the drugs are applied (line 1191). The OD of Com20 in the stationary phase is ~ 5 - 6.

Comment 4.20

Lines 989-990: Please expand on how the luminescence and OD values were normalized.

Action taken: In the revised manuscript, we provided an explanation of how the values were normalized (lines 1205 - 1209).

Comment 4.21

Lines 1213-1222: For the Germfree mouse both female and male mouse were included, for the SPF mice only male. Male and female mice have on occasion been shown to

respond different to treatment, why was it decided to use only male SPF mice? Also please include the number of SPF mice used.

We agree with the reviewer that treatment effects can indeed differ between male and female mice. To minimize confounding factors, we initially worked only with male mice, as the focus was on simplifying the experimental design to identify basic pharmacological effects without introducing the complexity of sex-related variables. However, for the later models (Com20 and humanized mice), we included both sexes to better reflect real-world conditions. In these models, we did not observe any detectable differences between the sexes.

Action taken: We acknowledge that sex differences will be an important consideration for future studies, and we agree that it is crucial to report these parameters clearly in our work. Therefore, we have now specified the number of male and female mice tested in each model (line 1579), as well as the number of SPF mice used (line 1585).

Minor points

Comment 4.22

Title, Line 47, 79,85, 100, 102: Bacteria class should not be italicized. Line 516-517: Order name also doesn't need to be italicized

We recognize that there are a range of italicization conventions in the literature, and we did not find any specific instructions about the use of italics for the names of bacterial taxa in journals from the Nature Publishing Group. Therefore, we adhered to the guidelines of the journals of the American Society for Microbiology, which state: "Names of all bacterial taxa (kingdoms, phyla, classes, orders, families, genera, species, and subspecies) are printed in italics; strain designations and numbers are not." (<https://journals.asm.org/writing-your-paper>). As an aside, this syntax has been shown to facilitate its recognition in scientific texts (Thines et al., 2020).

Comment 4.23

Line 88: “approximately 1200 FDA-approves drugs”, please include the exact number.

Action taken: We added the exact number (1197, line 94).

Comment 4.24

Line 938: stating specifically the 5 concentrations would be more helpful.

Action taken: In the revised manuscript, we added the specific concentrations used in each of the gradients (line 1153).

Comment 4.25

Line 1027: OD578 format

Action taken: We fixed the format across the manuscript accordingly.

Comment 4.26

Throughout paper: inconsistency of numerical formatting written vs numerals e.g Line 973, 1019, 1255, 1286.

We apologise for the confusion. This is not an inconsistency; in those lines, we follow the convention of rewording a sentence to avoid beginning with a numeral, as recommended by many English language style guides (e.g., see <https://apastyle.apa.org/learn/faqs/numbers-as-words>).

Comment 4.27

Line 1048: please include the pooled concentration

Action taken: We added the pooled concentration to the methods (4 nM, line 1422).

Comment 4.28

Figure 3b: the bacteria in grey are hard to read against the white background, it may be better to change to a different colour.

Action taken: We changed the figure to improve the readability of species names (ED Fig. 6h).

Comment 4.29

Figure 1b: I'm still not clear on the difference between >10 and 10-20 or >20 and 20-40 IC25?

Action taken: We added the following explanation to the figure legend: "The size of the circles indicates the percentage of species inhibited for a given concentration range. For example, clarithromycin inhibits 20% of pathogens at concentrations between 2.5 and 5 μM . For concentrations labelled as greater than (e.g. >20 μM), at the indicated value, the inhibition threshold was not reached and higher values (e.g., 40 μM) were not evaluated."

Comment 4.30

In figure 7: where is part e?

Action taken: Apologies for the oversight—the figure mistakenly did not include panel e. In the revised manuscript, we have updated the content and rearranged all figures accordingly.

Comment 4.31

Figure S1b: For better readability it may be good to include another colour strip to highlight antibiotic vs non-antibiotic drugs.

Action taken: We added a colour key to differentiate between antibiotics and non-antibiotics.

References

- Brugiroux, S., Beutler, M., Pfann, C., Garzetti, D., Ruscheweyh, H.-J., Ring, D., Diehl, M., Herp, S., Lötscher, Y., Hussain, S., Bunk, B., Pukall, R., Huson, D.H., Münch, P.C., McHardy, A.C., McCoy, K.D., Macpherson, A.J., Loy, A., Clavel, T., Berry, D., Stecher, B., 2016. Genome-guided design of a defined mouse microbiota that confers colonization resistance against *Salmonella enterica* serovar Typhimurium. *Nat Microbiol* 2, 16215. <https://doi.org/10.1038/nmicrobiol.2016.215>
- Celis, A.I., Aranda-Díaz, A., Culver, R., Xue, K., Relman, D., Shi, H., Huang, K.C., 2022. Optimization of the 16S rRNA sequencing analysis pipeline for studying in vitro communities of gut commensals. *iScience* 25, 103907. <https://doi.org/10.1016/j.isci.2022.103907>
- García-Santamarina, S., Kuhn, M., Devendran, S., Maier, L., Driessen, M., Mateus, A., Mastrorilli, E., Brochado, A.R., Savitski, M.M., Patil, K.R., Zimmermann, M., Bork, P., Typas, A., 2024. Emergence of community behaviors in the gut microbiota upon drug treatment. *Cell* S0092867424009668. <https://doi.org/10.1016/j.cell.2024.08.037>
- Ricaurte, D., Huang, Y., Sheth, R.U., Gelsinger, D.R., Kaufman, A., Wang, H.H., 2024. High-throughput transcriptomics of 409 bacteria–drug pairs reveals drivers of gut microbiota perturbation. *Nat Microbiol* 9, 561–575. <https://doi.org/10.1038/s41564-023-01581-x>
- Spragge, F., Bakkeren, E., Jahn, M.T., B. N. Araujo, E., Pearson, C.F., Wang, X., Pankhurst, L., Cunrath, O., Foster, K.R., 2023. Microbiome diversity protects against pathogens by nutrient blocking. *Science* 382, eadj3502. <https://doi.org/10.1126/science.adj3502>
- Stecher, B., Macpherson, A.J., Hapfelmeier, S., Kremer, M., Stallmach, T., Hardt, W.-D., 2005. Comparison of *Salmonella enterica* Serovar Typhimurium Colitis in Germfree Mice and Mice Pretreated with Streptomycin. *Infect Immun* 73, 3228–3241. <https://doi.org/10.1128/IAI.73.6.3228-3241.2005>
- Thines, M., Aoki, T., Crous, P.W., Hyde, K.D., Lücking, R., Malosso, E., May, T.W., Miller, A.N., Redhead, S.A., Yurkov, A.M., Hawksworth, D.L., 2020. Setting scientific names at all taxonomic ranks in italics facilitates their quick recognition in scientific papers. *IMA Fungus* 11, 25. <https://doi.org/10.1186/s43008-020-00048-6>

1 **Point-by-point response to the referee's comments**

2
3 We thank the referees for the time and effort they dedicated to evaluating our
4 manuscript and for the constructive comments provided. We appreciate the
5 opportunity to revise our work and have carefully addressed each of the points raised.
6 Below, we provide a point-by-point response, with our revisions clearly marked in the
7 manuscript.

8 9 **Referee #1**

10
11 1. Additional experiments using humanized mice have been added to demonstrate the
12 concern that the phenotypes observed here are specific to low-complexity
13 microbiomes, and immune profiling is performed to investigate as to whether certain
14 drugs affect host status. These experiments adequately address these major
15 concerns, and nicely extend the current work.

16
17 We thank Referee #1 for recognizing the strengths of our work.

18
19 2. I think it would be important to show that increased colonization with Salmonella
20 during Terfenadine treatment has quantifiable consequences on the host. Therefore,
21 I would suggest to move pertinent panels of Extended Data Figure 11 to the main
22 figure, e.g. Figure 4.

23
24 We agree with this suggestion and have moved the pathoscore panel
25 (previously Extended Data Figure 11a) to panel i in the main Figure 4. While the
26 pathoscore is less quantitative than other measures, we believe it effectively
27 summarizes multiple phenotypes and have therefore chosen it as the representative
28 panel.

29
30 3. There is still some lingering concerns that conceptually, the effect of non-antibiotic
31 drugs on colonization resistance is very similar to the effect of antibiotics (which is well
32 documented), i.e. decreased biomass, decreased microbiome diversity, and
33 decreased nutritional competition, which should be addressed.

35 We agree with the referee that there are conceptual similarities between the
36 effects of non-antibiotic and antibiotic drugs on colonization resistance. However,
37 there are key differences, such as the likely need for higher concentrations of non-
38 antibiotics to promote pathogen colonization. Additionally, non-antibiotics typically
39 have a narrower target spectrum, meaning that only specific non-antibiotics—rather
40 than all—will disrupt colonization resistance. This distinction is important, as it has
41 significant implications for both risk assessment and drug development strategies.

42

43 To clarify that the underlying principles by which both antibiotic and non-
44 antibiotic drugs disrupt colonization resistance are similar, we have expanded this in
45 the discussion (lines 441–449).

46

47 **Referee #2**

48

49 In the revised manuscript, substantial modifications have been made to the main text,
50 figures, and legends, improving readability and clarifying key points.

51

52 We thank the referee for their positive assessment of the revised manuscript
53 and their helpful feedback, which contributed to further improving the clarity and
54 presentation of our work.

55

56 However, I believe the analysis of the molecular mechanisms underlying the effects of
57 drugs on commensal bacteria requires further depth. While these mechanisms are
58 likely to be case-by-case, depending on the drug and specific context (as clearly
59 demonstrated in the manuscript), without presenting at least one concrete molecular
60 mechanism—ideally a novel one—the study risks being perceived as merely a
61 collection of observations, which would be unfortunate.

62

63 We share the referee's curiosity about the underlying molecular mechanisms
64 and fully intend to explore some of them in greater depth in future studies. We also
65 hope to spark interest in other laboratories—especially those with expertise in specific
66 drug classes or disease models and access to the appropriate experimental
67 systems—to pursue these questions further.

68 Here, we go beyond a collection of observations to propose and discuss an
69 ecological framework explaining how non-antibiotic drugs can impair the microbiome's
70 ability to resist pathogen colonization. In this context, focusing on the precise mode of
71 action of a single compound would narrow the scope and detract from the broader,
72 cross-compound perspective we aim to present. Therefore, such analysis is beyond
73 the scope of this work.

74
75

76 For example, in Fig. 3, the manuscript concludes that the reduced fitness of *fldD*-
77 deficient *Salmonella* under tiratricol treatment suggests a decrease in fumarate
78 availability as a key factor. However, the precise molecular events underlying this
79 effect remain unclear. As mentioned in the discussion, "Tiratricol could cause lysis of
80 commensal microorganisms, altering the pool of available substrates such as
81 microbiota-derived fumarate," but the specific commensal member(s) directly affected
82 by tiratricol and the molecular mechanisms driving this effect should be further
83 elucidated.

84

85 We agree with the referee that the precise molecular dynamics by which
86 tiratricole affects fumarate availability merit further investigation; for example, by
87 directly measuring metabolite concentrations following treatment. However, we would
88 like to highlight that the manuscript already includes data on the specific commensal
89 members directly affected by tiratricole, both in monocultures (ED2a, Suppl. Table 2)
90 and in the Com20 and Com21 communities (ED Fig. 9b). We are currently
91 investigating potential molecular targets of tiratricole in these commensals, but as
92 outlined above, we believe that such in-depth mechanistic analysis goes beyond the
93 scope of this manuscript.

94

95 Additionally, in the final HUM mice experiment, terfenadine was used, yet its effects
96 were not shown in Fig. 3. Given that terfenadine was consistently used throughout the
97 study, further investigation into its molecular mechanisms of action seems warranted.

98 Terfenadine belongs to the category of biomass-lowering drugs that do not
99 specifically target *E. coli* and is included in Fig. 3a. However, we did not observe
100 significant differences in *S. Tm* growth between Com20 and Com21 following

101 treatment. Consequently, terfenadine appears as one of the unlabeled data points (for
102 exact values, see Supplementary Tables 4 and 8). This is not an inconsistency, but
103 rather reflects the absence of *E. coli*-dependent effects.

104 We are intrigued by these results, which merit further investigation to elucidate
105 the underlying cellular and molecular dynamics. Indeed, we are actively pursuing such
106 analyses, which we believe warrant their own, separate research article.

107
108 One final point: the manuscript categorizes three major mechanisms through which
109 non-antibiotic drugs promote pathogen expansion. A summary figure illustrating which
110 tested drugs correspond to each category would enhance clarity and improve the
111 overall structure of the findings.

112 We think this is a very good suggestion. In fact, we had similar intentions when
113 providing examples for each of the three major modes discussed in lines 396–412.
114 However, since colonization outcomes are context-dependent (e.g., see Zafirlukast),
115 mechanisms vary with drug concentration, and multiple mechanisms can overlap, we
116 found it challenging to create a summary figure that would be both accurate and easy
117 to interpret.

118
119 Instead, we revised subpanels of Extended Data Figures 5a, 5e, and 9a to
120 include normalized OD values. This allows readers to more readily distinguish
121 between biomass reduction and community composition shifts across drug
122 concentrations. We understand this may not fully align with the summary figure the
123 referee proposed, but we believe this is a precise and informative way to present these
124 findings.

125

126 **Referee #3**

127

128 The authors have addressed my prior concerns and have added new data and new
129 analyses that further strengthen the manuscript's conclusions.

130

131 Referee #3 (Remarks on code availability):

132 I'm not the best person to review the code, sorry

133

134 We appreciate Referee #3's positive assessment and are pleased that the
135 revisions strengthened the manuscript.

136

137 **Referee #4**

138

139 Griebhammer and colleagues have very comprehensively addressed the reviewer
140 comments in this revised manuscript. In particular, the further analysis to help explain
141 the different drug sensitivity observed in Gammaproteobacteria compared to other gut
142 commensals has improved the manuscript. Similarly, the experiments with a wash
143 step between treatment and pathogen is an important control which has now been
144 added. I have no further suggestions.

145

146 We thank Referee #4 for their kind comments and are happy they appreciated
147 the improvement of our manuscript.

148